# CDK phosphorylation of Sfr1 downregulates Rad51 function in late-meiotic homolog invasions

Inés Palacios-Blanco[1], Lucía Gómez[1], María Bort[1], Nina Mayerová [2], Silvia Bágeľová Poláková [2,3] & Cristina Martín-Castellanos [1✉]

## Abstract

**Meiosis is the developmental program that generates gametes. To produce healthy gametes, meiotic recombination creates reciprocal exchanges between each pair of homologous chromosomes that facilitate faithful chromosome segregation. Using fission yeast and biochemical, genetic, and cytological approaches, we have studied the role of CDK (cyclin-dependent kinase) in the control of Swi5–Sfr1, a Rad51-recombinase auxiliary factor involved in homolog invasion during recombination. We show that Sfr1 is a CDK target, and its phosphorylation downregulates Swi5–Sfr1 function in the meiotic prophase. Expression of a phospho-mimetic *sfr1-7D* mutant inhibits Rad51 binding, its robust chromosome loading, and subsequently decreases interhomolog recombination. On the other hand, the non-phosphorylatable *sfr1-7A* mutant alters Rad51 dynamics at late prophase, and exacerbates chromatin segregation defects and Rad51 retention observed in *dbl2* deletion mutants when combined with them. We propose Sfr1 phospho-inhibition as a novel cell-cycle-dependent mechanism, which ensures timely resolution of recombination intermediates and successful chromosome distribution into the gametes. Furthermore, the N-terminal disordered part of Sfr1, an evolutionarily conserved feature, serves as a regulatory platform coordinating this phospho-regulation, protein localization and stability, with several CDK sites and regulatory sequences being conserved.**

**Keywords** CDK; Meiosis; Rad51; Recombination; Swi5-Sfr1
**Subject Categories** Cell Cycle; DNA Replication, Recombination & Repair

## Introduction

Meiosis is the developmental program that ensures the genome transmission from one generation to the next in sexually reproducing organisms. In sharp contrast to mitotic dividing cells, meiosis is linked to the generation of genetic variability which is the driving force of evolution. This is achieved by the assortment of the parental chromosomes in the meiotic products (gametes), as well as by recombination between them (physical exchange of genetic material between each pair of homologous chromosomes). Recombination is remarkably important, since it also imposes a stable architecture on the pair of homologs that ensures their correct orientation and faithful segregation to opposite poles of the cell during the first meiotic division (reductional chromosome segregation) (Marston and Amon, 2004; Petronczki et al, 2003). Chromosomes without reciprocal or with misplaced exchanges missegregate and produce gametes with an abnormal number of chromosomes. This underlies the high frequency of spontaneous miscarriages in human conceptions as well as the severe congenital defects associated to genetic syndromes (Hassold and Hunt, 2021). As a key feature of meiosis, recombination is a complex process under a very tight regulation (Hunter, 2015; Keeney et al, 2014; Phadnis et al, 2011; Yadav and Claeys Bouuaert, 2021).

Meiotic recombination is the outcome of the repair of programmed Double-Strand Breaks (DSBs), which are introduced in the DNA at early meiosis (prophase) by the conserved meiosis-specific Spo11 transesterase and its accessory proteins (Arter and Keeney, 2023; Bergerat et al, 1997; Hunter, 2015; Keeney et al, 1997; Yadav and Claeys Bouuaert, 2021). After Spo11 removal from break sites, single-stranded DNA (ssDNA) nucleofilaments are generated by endonucleolytic resection of the break ends. Coated with the strand exchange proteins Rad51 and Dmc1, these nucleofilaments invade the homologous chromosome searching for a repair template (Brown and Bishop, 2014). The fate of this invasion is an important point of regulation. If the invasion is stabilized, intermediates mature into interhomolog joint molecules (IH JM; Holliday junctions) that when resolved by structure-selective endonucleases (SSE) can generate crossovers (COs, reciprocal exchange between the pair of homologs). The physical links provided by COs ensure the reductional chromosome segregation at the first meiotic division. On the other hand, the counteraction of helicases dissolves the invasion, which facilitates the reannealing with the sister chromatid in the original chromosome (Synthesis-Dependent Strand Annealing pathway, SDSA) and produces non-reciprocal exchanges between the parental chromosomes (non-crossovers, NCOs) (Hunter, 2015; Lorenz, 2017; Lorenz et al, 2014; Lorenz et al, 2012). In addition, template choice is also important for the recombination outcome; invasion can be established with the homologous chromosome (interhomolog, IH) or with the sister chromatid (intersister, IS), and the preferential template choice varies locally and between species (Hyppa and Smith, 2010).

[1]Instituto de Biología Funcional y Genómica (IBFG), CSIC-USAL, Salamanca 37007, Spain. [2]Department of Genetics, Faculty of Natural Sciences, Comenius University in Bratislava, Bratislava 841 04, Slovakia. [3]Centre of Biosciences SAS, Institute of Animal Biochemistry and Genetics, Bratislava 840 05, Slovakia. ✉E-mail: cmartin@usal.es

The loading to ssDNA and the activity of Rad51 and Dmc1 recombinases require a number of conserved accessory proteins (Brown and Bishop, 2014; Tsubouchi et al, 2021). In fission yeast, it has been proposed that Rad51/Dmc1 accessory-complexes Swi5–Sfr1, Rad55–Rad57, and Rlp1-Rdl1-Sws1 protect invasion intermediates from the unwinding action of FANCM-type Fml1 and RecQ-type Rqh1 helicases, promoting Holliday junctions and COs (Lorenz et al, 2014; Tsutsui et al, 2014). In particular, the Swi5–Sfr1 complex is one of the most relevant Rad51/Dmc1 mediators, well conserved through evolution from yeasts to humans, and extensively molecularly studied in fission yeast (Argunhan et al, 2017). In vitro assays with purified proteins have shown that the complex binds to Rad51 (and Dmc1) stabilizing the nucleofilaments, enhances ssDNA binding activity of Dmc1, stimulates ATPase activity of Rad51, and promotes Dmc1 and Rad51-mediated strand exchange reactions (Haruta et al, 2006; Lee et al, 2023; Murayama et al, 2013). In addition, the complex has been structurally characterized (Kokabu et al, 2011; Kuwabara et al, 2012; Saikusa et al, 2013). Sfr1 protein is composed of two clear independent domains. The N-terminal domain is intrinsically disordered and mediates the interaction with Rad51 anchoring the complex to the nucleofilament; meanwhile, the conserved and structured C-terminal domain binds to the Swi5 partner generating an elongated alpha-helix structure that inserts into the groove of the nucleofilament and stimulates the strand exchange reaction. More recently, positively charged regions in the N-terminal have been shown to mediate Rad51 cooperative interactions (Argunhan et al, 2020). The long N-terminal disordered domain is a highlighted feature in Sfr1 homologs (Argunhan et al, 2020). Moreover, functions of this region are conserved as it is the case for the binding to Rad51 and Dmc1 recombinases of the budding yeast ortholog Mei5 (Hayase et al, 2004; Say et al, 2011). The important role of Swi5–Sfr1 in meiotic recombination is shown by the reduction in recombination of swi5 and sfr1 deletion mutants (Ellermeier et al, 2004; Hyppa and Smith, 2010; Lorenz et al, 2014; Young et al, 2004). Swi5–Sfr1 is specifically required for IH recombination, and a reduction in Holliday junctions is specifically observed in IH, but not IS, JMs at DSB hotspots (Hyppa and Smith, 2010).

Proper resolution of the recombination intermediates is crucial to maintain genome stability during meiosis. The self-inflicted DSBs have to be repaired on time before the first meiotic division, when homologous chromosomes segregate apart (meiosis I). In budding yeast, several cell-cycle-dependent mechanisms have been described to ensure timely resolution of recombination intermediates (Blanco and Matos, 2015; Grigaitis et al, 2020; San-Segundo and Clemente-Blanco, 2020). One of them is the phosphorylation of the conserved Mus81-Mms4 SSE (Matos et al, 2011); the hyperphosphorylation of Mms4 boosts the complex nuclease activity and promotes resolution of JMs prior to anaphase I. Similarly, phosphorylation of the conserved RecQ-family DNA helicase Sgs1 enhances its unwinding activity to avoid an abnormal accumulation of aberrant JMs (Grigaitis et al, 2020). Furthermore, Yen1 SSE is activated in meiosis II and acts as a backup system to resolve late persistent JMs (Alonso-Ramos et al, 2021; Matos et al, 2011). The importance of this JM clearance has been also reported in fission yeast where the UvrD-type Fbh1 helicase and its loader (Dbl2) have been shown to participate in JM resolution. In the absence of these proteins, cells enter meiosis I with an abnormal number of Rad51 foci and recombination intermediates (Polakova et al, 2016; Sun et al, 2011).

In this report, we show that fission yeast Sfr1 is highly phosphorylated at the end of meiotic prophase in its N-terminal disordered domain, and that Sfr1 is a substrate for cyclin-dependent kinase (CDK) phosphorylation. Sfr1 phosphorylation impairs the binding to Rad51 and its loading onto chromosomes, reducing meiotic recombination. This inhibition is particularly important before the first meiotic division and sfr1 phospho-null mutants enhance the segregation defects observed in dbl2 deletion mutants. Our data support a model where CDK-dependent Sfr1 phosphorylation at the end of prophase downregulates Swi5–Sfr1 function by inhibiting the interaction with Rad51, and thus, it helps to avert IH JMs prior to chromosome segregations. Based on our genetic data, the aborted JMs would be repaired with the sister chromatid. This is a novel regulatory mechanism, directly impinging on a broadly conserved Rad51-accessory protein, to prevent the formation of late IH intermediates. If not repaired on time, these intermediates would interfere with the segregation of chromosomes and, therefore, hamper the viability of gametes. Furthermore, we describe the disordered N-terminal part of Sfr1 as a hub for protein regulation, required not only for CDK-modulated Rad51 binding but also for protein stability and nuclear localization. This study will help to broadly understand homologous recombination in other eukaryotes and how genome integrity is maintained during sexual reproduction. Furthermore, except in budding yeast, the Swi5–Sfr1 complex is not meiosis-specific, and the results will also be useful for understanding genome stability in response to DNA damage.

## Results

### CDK phospho-regulation of Sfr1

Using the fission yeast S. pombe as a model organism, we previously described that CDK (Cdc2) activity is required to form DSBs as well as to maintain the nuclear architecture during meiotic prophase (Bustamante-Jaramillo et al, 2019; Bustamante-Jaramillo et al, 2021). In addition, based on the comparison of DSB and CO levels, we proposed CDK might have an additional function, acting downstream of break formation in CO inhibition (Bustamante-Jaramillo et al, 2019). One possibility is that it negatively controls the formation or stability of the displacement loop (D-loop) generated during homolog invasion. In fission yeast it has been proposed that Rad51/Dmc1 accessory-complexes Swi5–Sfr1, Rad55–Rad57, and Rlp1-Rdl1-Sws1 protect the D-loop from the unwinding action of helicases (FANCM-type Fml1 and RecQ-type Rqh1), promoting in this way the formation of Holliday junctions and COs (Lorenz et al, 2014). In this view, CDK (Cdc2) could phosphorylate and inhibit some of the accessory proteins required for nucleoprotein filament stabilization and strand exchange activity. Based on the presence of CDK signatures on the proteins and the available data generated in different proteomic approaches, we selected Sfr1 as a good candidate (https://www.pombase.org/gene/SPBC28F2.07). Over the years, five Ser/Thr phosphorylated residues have been identified in Sfr1; four of them are putative CDK phosphorylation sites, and at least one of them (Ser165) is phosphorylated in a Cdc2-dependent manner during mitotic

M-phase (Carpy et al, 2014; Cipak et al, 2009; Kettenbach et al, 2015; Koch et al, 2011; Swaffer et al, 2016). Except for Cipak et al, 2009 where a Sfr1-TAP purification was analyzed, these studies were not focused on Sfr1. Furthermore, all of them were conducted in vegetative cells.

To evaluate the phosphorylation of Sfr1 protein during meiotic prophase, we profited of an EGFP-Sfr1 version (Akamatsu et al, 2007). This tagged protein is proficient in meiotic recombination since CO levels are not affected in cells harboring this EGFP-Sfr1 version (Appendix Fig. S1). We have used a temperature-sensitive allele of the inhibitor of meiosis Pat1 (*pat1-114*) to induce synchronous meiotic entry of diploid cells previously arrested in G1 by nitrogen depletion (Iino and Yamamoto, 1985). Cells collected at different time points of the kinetics were used for different analyses. Counting of nuclei per cell and flow cytometry were used to follow the synchrony of the culture, and protein samples were analyzed by western blot. EGFP-Sfr1 protein was clearly detected from the end of S-phase (2.5 h after meiotic induction) to the entry into the first chromosome segregation (meiosis I, 4.5 h), with the highest levels during meiotic prophase (Fig. 1A). When protein samples were analyzed in Phos-tag gels, a clear reduction in the mobility of the protein was also observed during meiotic prophase, with the strongest mobility shift at later time points (4.5 h) (Fig. 1B). These data suggest that EGFP-Sfr1 protein is heavily phosphorylated at the end of meiotic prophase. Moreover, time of Sfr1 phosphorylation was concomitant with the accumulation of Cdc13 cyclin, the main cyclin required for chromosome segregation in fission yeast meiosis (Borgne et al, 2002) (Fig. 1B). As mentioned above, four phosphorylated residues in putative CDK sites have been identified in vegetative cells; besides, three additional putative sites are present in the protein which could be phosphorylated in meiosis. All seven sites show ≥0.833 score in NetPhosYeast 1.0 prediction server. Thus, to further explore this result, we generated a mutant version in which all the putative CDK phosphorylation sites (7 sites in the protein, see Fig. 1C for a scheme) were changed to non-phosphorylatable alanine residues. The mobility shift observed in Phos-tag gels was almost completely abolished in this EGFP-Sfr1-7A mutant protein, indicating that it was related to modifications in these residues of the protein (Fig. 1D). Furthermore, during the course of this study, Sevcovicova and colleagues described that six out the seven CDK sites in Sfr1 are indeed in vivo phosphorylated during meiotic prophase (Sfr1-TAP purification analysis) (Sevcovicova et al, 2021). Notice also that the EGFP-Sfr1-7A protein is present at meiosis I compared to the EGFP-Sfr1 wild-type protein whose levels are reduced upon entry into meiosis I (Fig. 1D).

To address the Cdc2 dependency of these modifications, we used an ATP-analog-sensitive *cdc2-asM17* allele to chemically downregulate Cdc2 activity (Aoi et al, 2014). The ATP analog 1-NM-PP1 was added just after meiotic DNA replication (2.5 h after meiotic induction) in synchronized diploid *pat1-114* cells, and EGFP-Sfr1 mobility was analyzed as above (Fig. 2A). In control cells treated with DMSO solvent, a clear reduction in the mobility of the protein was observed when cells were going through prophase. This mobility shift was diminished in cells of the same culture treated in parallel with 1-NM-PP1, showing a 50% reduction in the slow-migrating band compared to DMSO-treated cells at 4 h after meiotic induction ($n = 4$, $P$ value = 0.0337). This result suggests that Cdc2 is responsible for the phosphorylation of Sfr1 during meiotic prophase.

To further support this observation, we used GFP-Trap to enrich EGFP-Sfr1 protein from cells at late meiotic prophase (4 h after meiotic induction), and the immunoprecipitates (IPs) were analyzed by western blot using anti-phospho (Ser) CDK-substrate antibodies. EGFP-Sfr1 was clearly detected, and the antibody recognition was abolished when the IP was treated with λ-phosphatase (Fig. 2B, notice also the collapse of the EGFP-Sfr1 band when treated with phosphatase in the anti-GFP control blot). In addition, we performed in vitro kinase assays. EGFP-Sfr1 was enriched by GFP-trap IP from cells at early prophase (3 h after meiotic induction), treated with λ-phosphatase to remove any phosphorylation, and incubated in kinase reactions with Cdc13 IPs obtained from cells at meiosis I (5 h) where its associated Cdc2-kinase activity is maximal (Borgne et al, 2002). As shown in Fig. 2C, EGFP-Sfr1 was detected with anti-phospho (Ser) CDK-substrate antibodies only when mixed with Cdc13 IPs, and not when mixed with control IPs where anti-Cdc13 antibodies were not added to the extract. These results indicate that Sfr1 is most likely a direct target of Cdc2.

## Sfr1 phospho-regulation modulates Rad51 recombinase binding and the recombination proficiency

Argunhan and colleagues have identified the regions on Sfr1 required for efficient interaction with the recombinase Rad51 (Argunhan et al, 2020). Using physical methods to analyze energy changes in the disordered N-terminal part of Sfr1 upon interaction with Rad51, they identified two regions enriched in basic residues (positive charged) (see Fig. 1C for a scheme). Neutralization of positive charges in these regions replacing 7 arginine and lysine residues by non-charged alanine residues, almost abolishes Rad51 binding in vitro (Argunhan et al, 2020). The CDK sites in Sfr1 are located at the N-terminal part of the protein, within or in close proximity to the described positively charged regions (Fig. 1C). Thus, we decided to address whether the phosphorylation of the CDK sites modulates the interaction with Rad51. For doing so, we generated GST-Sfr1-7A and GST-Sfr1-7D versions, where the serines and threonines in these sites were replaced by non-phosphorylatable residues (alanine) or phospho-mimetic residues (aspartic acid). These Sfr1 protein versions, control GST-Sfr1, as well as untagged Rad51, were expressed in bacteria and Sfr1-Rad51 interaction was addressed by GST pulldowns. As a control, we used a GST-Sfr1-WI version (Weak Interaction), the mutant protein reported to strongly impair Rad51 interaction (Argunhan et al, 2020); on sake of clarity we have renamed this mutant to avoid confusion with the Sfr1-7A version generated in our study. Pulldown assays showed that GST-Sfr1-7A protein significantly binds to Rad51 with an 81% efficiency compared to control GST-Sfr1 protein ($n = 3$, $P$ value = 0.5709) (Fig. 3A). By contrast, the phospho-mimetic GST-Sfr1-7D protein was unable to bind to Rad51 ($n = 3$, $P$ value = 0.0080), and this defect, though not statistically different, was even more severe than the one observed with the GST-Sfr1-WI protein which retained an 17% efficiency ($n = 3$, $P$ value = 0.0208). The interaction of phospho-mutant Sfr1 proteins and Rad51 was also addressed by Yeast Two-Hybrid (Y2H) assays. As shown in Fig. EV1A, Sfr1-7A protein showed a robust Rad51 interaction; meanwhile, Sfr1-7D protein did not interact with Rad51. In addition, wild-type Sfr1 and Sfr1-7A interacted with the same efficiency in plates containing 3-amino-

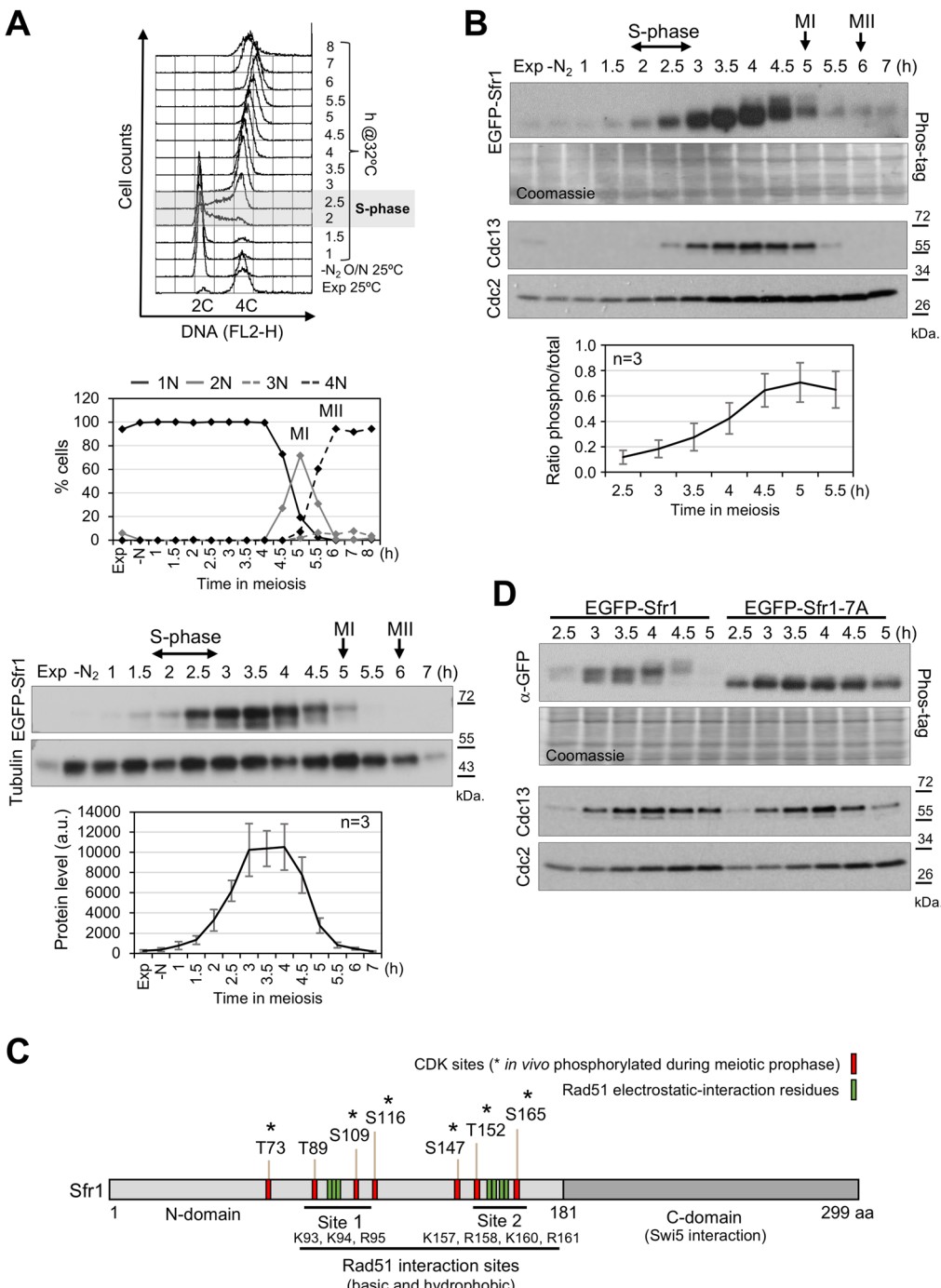

**Figure 1. Sfr1 expression and phosphorylation during meiosis.**

(A) EGFP-Sfr1 expression during diploid *pat1-114* synchronous meiosis (CMC1649 strain). Upper, FACS analysis; timing of DNA synthesis is highlighted. Below, meiotic progression measured as the number of nuclei per cell; timing of meiosis I (MI) and meiosis II (MII) is indicated. Lower, western blot detection of EGFP-Sfr1 protein; tubulin detection was used as loading control. Below is the quantification of the EGFP-Sfr1 signal in three independent kinetics (mean; error bars: standard error of the mean (SEM)). (B) Upper, western blot detection of EGFP-Sfr1 protein in Phos-tag gels (20 µM) and Coomassie staining of the membrane as loading control. Below, western blot detection of Cdc13 cyclin and Cdc2 in the same kinetics. Lower, quantification of EGFP-Sfr1 mobility shift in three independent kinetics (mean; error bars: SEM). (C) Scheme of Sfr1 protein. Protein domains, CDK sites, and Rad51 interaction sites are highlighted. In vivo phosphorylated CDK sites (*) during meiotic prophase were identified using a Sfr1-TAP purification in (Sevcovicova et al, 2021). Rad51 interaction sites were identified by structural approaches in (Argunhan et al, 2020). (D) Upper, western blot detection of EGFP-Sfr1 (CMC1649 strain) and EGFP-Sfr1-7A (CMC1733 strain) proteins during diploid *pat1-114* synchronous meiosis in Phos-tag gels (20 µM) and Coomassie staining of the membrane as loading control. All the samples were loaded in the same gel. Two independent meiotic kinetics were performed with similar reduction in the mobility of the EGFP-Sfr1-7A protein. Lower, western blot detection of Cdc13 cyclin and Cdc2 in the same kinetics. Source data are available online for this figure.

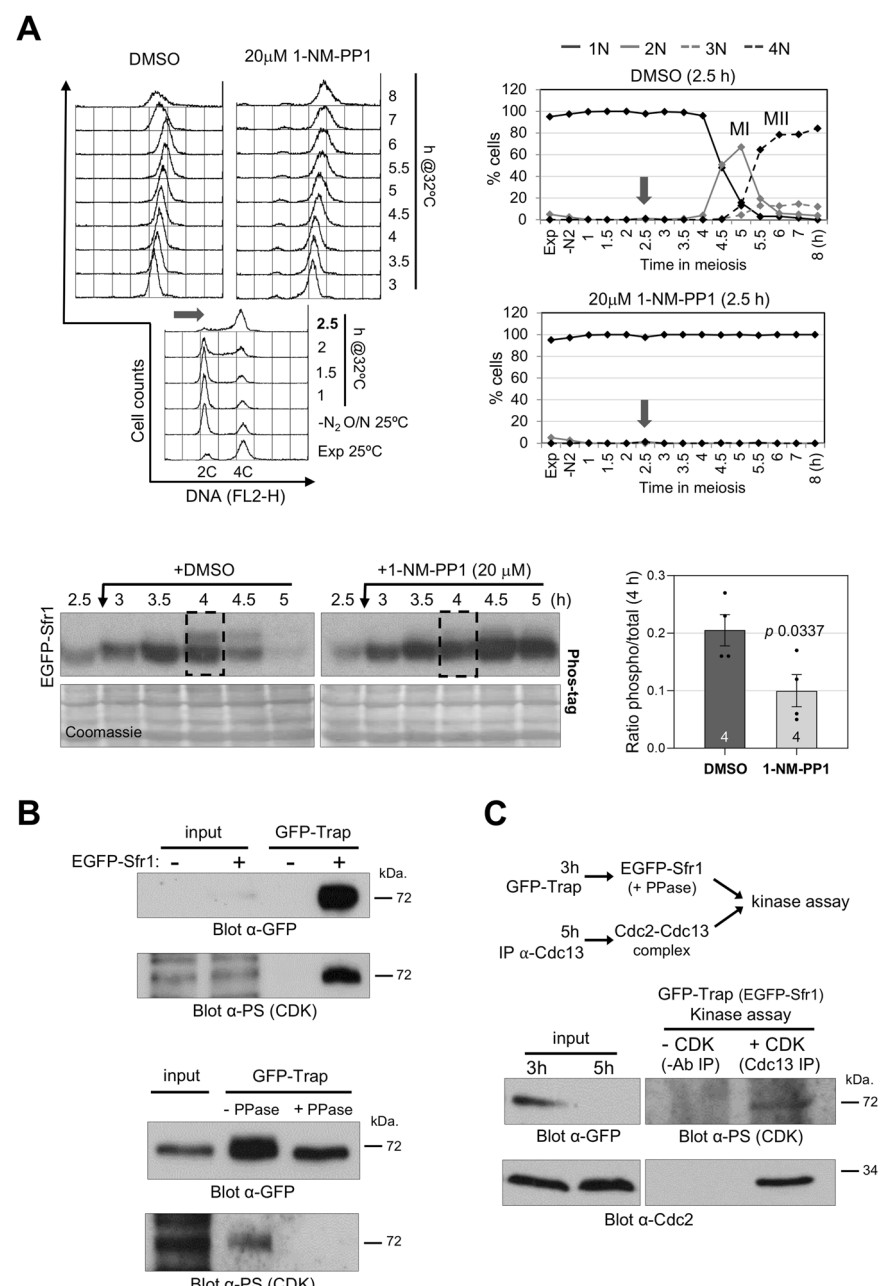

**Figure 2. Sfr1 phosphorylation depends on Cdc2.**

(**A**) Upper left, FACS analysis of *pat1-114 EGFP-sfr1 cdc2-asM17* diploid synchronous meiosis (CMC1660 strain). After 2.5 h of meiotic induction, culture was split in two and 1-NM-PP1 (20 μM) or same volume of DMSO added to the cultures. Upper right, meiotic progression measured as the number of nuclei per cell; timing of meiosis I (MI) and meiosis II (MII) is indicated. Time point of 1-NM-PP1 (or DMSO) addition is indicated. Lower left, western blot detection of EGFP-Sfr1 protein in Phos-tag gels (20 μM) and Coomassie staining of the membranes as loading control. Lower right, quantification of mobility shift in four independent kinetics (mean; error bars: SEM); *P* value is indicated (Student's *t* test; unpaired, two tails). (**B**) *pat1-114 EGFP-sfr1* diploid cells (CMC1649) were induced to enter meiosis and collected for protein extraction at late prophase (4 h after meiotic induction). Upper, western blot detection of EGFP-Sfr1 protein after GFP-Trap immunoprecipitation (IP). Input extracts are shown. A parallel culture of cells with untagged Sfr1 version (CMC1074) was used as control. Samples were loaded in duplicate in the same gel (100 μg inputs and 50% IPs), and half of the membrane incubated with anti-GFP antibodies and the other half incubated with anti-phospho (Ser) CDK-substrate antibodies. Two replicates were done with samples collected in the same kinetics and independently analyzed weeks apart; similar results were obtained. Lower, western blot detection of EGFP-Sfr1 protein after GFP-Trap IP and λ-phosphatase treatment; parallel reaction without phosphatase was used as control. Input extract is shown. Samples were loaded in duplicate in the same gel (100 μg input and 50% −/+ phosphatase IPs), and analyzed as in (**B**). Three independent experiments were done; two with samples of the same meiotic kinetics independently analyzed weeks apart, and a third experiment with samples of an independent meiotic kinetics; similar results were obtained. (**C**) In vitro kinase assays. Scheme of the experiment is shown. *pat1-114 EGFP-sfr1* diploid cells (CMC1649) were induced to enter meiosis and collected for protein extraction at early prophase (3 h after meiotic induction) and meiosis I (5 h). Extracts from cells in prophase were used for GFP-Trap IP, treated with λ-phosphatase, and split in two. Extracts from cells in meiosis I were used for Cdc13 IP (+ CDK); parallel IP without antibodies was used as a negative control (−CDK). Each IP was mixed with half of the GFP-trap IP, and incubated for kinase reaction. The whole reaction (50% GFP-trap) was loaded in a single gel along with input extracts (100 μg), and western blot detection of EGFP-Sfr1 (anti-GFP), Cdc2 (anti-Cdc2, as a proxy of Cdc13 IP since direct detection was masked by immunoglobulins), and anti-phospho (Ser) CDK-substrate (anti-PS CDK) were performed. Source data are available online for this figure.

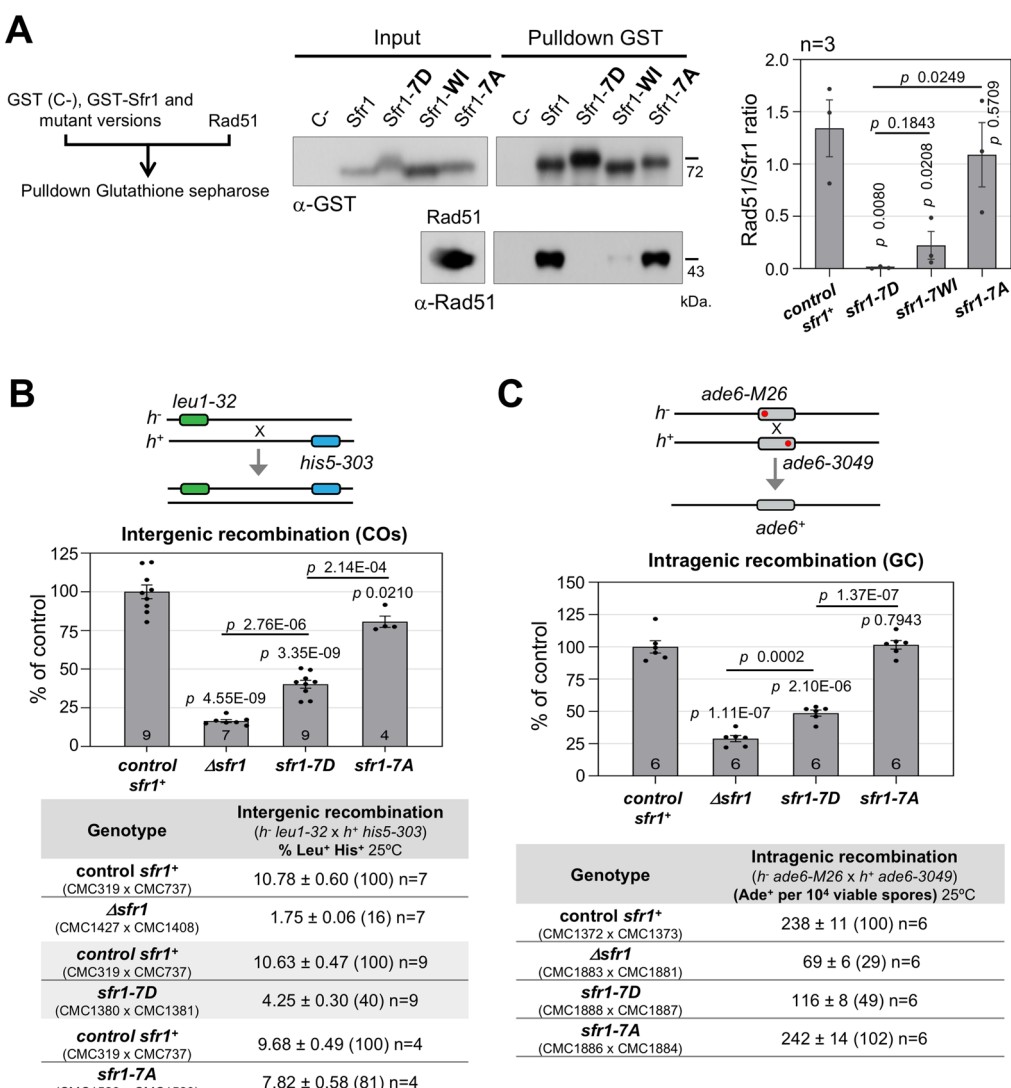

**Figure 3. Sfr1 phosphorylation impairs Rad51 recombinase binding and meiotic recombination.**

(A) Sfr1 phosphorylation impairs binding to recombinase Rad51 in pulldown in vitro assays. Scheme of the experiment is shown. Bacteria containing pGEX-2T (empty vector, CMC93), pGEX-4T-1 sfr1 (CMC96), pGEX-4T-1 sfr1-7D (CMC97), pGEX-4T-1 sfr1-WI (CMC98), pGEXT-4T-1 sfr1-7A (CMC102) or pET-11d rad51 (CMC99) plasmids were induced for expression and collected for protein extraction. Extract of GST (empty vector, C -) and GST-Sfr1 versions were mixed with Rad51 extracts and pulled down. Pulldowns were loaded in duplicate in the same gel, half of the membrane was incubated with anti-GST antibodies and the other half with anti-Rad51 antibodies; input extracts (40 μg) loaded in parallel in a different gel are shown. The experiment was performed four times in independent inductions of protein expression with similar results. Quantification of the Rad51/GST ratio is shown on the right. Data are the mean $+/-$ SEM of three independent experiments. P values were calculated based on Student's t test (unpaired, two tails). (B, C) Recombination assays. Schemes of the different assays are shown. Crosses were plated for recombination analysis at least twice and the results are based on the cumulative numbers in each cross. Tables show recombination frequencies $+/-$ SEM of n independent crosses, with numbers in parentheses showing the percentages relative to wild-type controls (sfr1+). Each mutant was analyzed only with crosses in the same experiment. Strains used in the crosses are indicated. Graphs show recombination expressed as the mean of the percentage relative to the control cross $+/-$ SEM of n independent crosses. P values were calculated based on Student's t test (unpaired, two tails). (B) Intergenic recombination (COs) in sfr1 phospho-mutants. Crosses of $h^-$ leu1-32 × $h^+$ his5-303 strains were performed in MEA at 25 °C. CO levels are expressed as the percentage of Leu+ His+ recombinants per haploid spore colonies; 28–246 haploid Leu+ His+ recombinant colonies scored in each independent cross, and 238–1812 total haploid Leu+ His+ recombinant colonies scored per genotype. (C) Intragenic recombination (GC) in sfr1 phospho-mutants. Crosses of $h^-$ ade6-M26 × $h^+$ ade6-3049 strains were performed in SPA at 25 °C. GC levels are expressed as the mean of Ade+ colonies per $10^4$ spore colonies; 45–413 Ade+ recombinant colonies scored in each independent cross, and 466–1722 total Ade+ recombinant colonies scored per genotype. Source data are available online for this figure.

1,2,4-triazol, a competitive inhibitor of *HIS3* gene product which helps to discriminate differences in the interaction strength. These results indicate that Sfr1 binding to the recombinase Rad51 is modulated by CDK phosphorylation. To discard structural alterations in the mutant proteins, Swi5–Sfr1 complexes with wild-type, Sfr1-7A and Sfr1-7D proteins were modeled by ColabFold. The large N-terminal domain of the phospho-mutant proteins remains disorganized as the wild-type and it does not interfere with the core structured complex (Swi5/Sfr1-C-terminal) (Fig. EV1B).

Next, we studied the in vivo impact of this regulation. We analyzed the IH invasion efficiency of the *sfr1-7A* and *sfr1-7D* alleles addressing both intergenic and intragenic recombination (see schemes of the assays in Fig. 3B,C). As shown in Fig. 3B, intergenic recombination, measured as COs between *leu1-32* and *his5-303* genetic markers on chromosome II, was strongly reduced in *sfr1-7D* mutants to a 40% of the CO levels observed in control crosses ($n = 9$, $P$ value = 3.35E-09). This effect was statistically different from the levels observed in *sfr1* deletion mutants where COs were severely reduced to 16% ($n = 7$, $P$ value = 4.55E-09), and *sfr1-7A* mutants where CO levels were modestly reduced to 81% ($n = 4$, $P$ value = 0.0210). The statistical difference between *sfr1-7D* and *sfr1-7A* alleles ($n = 4$, $P$ value = 2.14E-04) indicates that the changes introduced in the protein were not substantially altering the structure as it is also supported by the ability of the Sfr1-7A protein to maintain the interaction with Rad51 and by the ColabFold modeling (Figs. 3A and EV1). Homologous recombination was also assayed as intragenic recombination, measured as gene conversion (GC) between *ade6-M26* and *ade6-3049* alleles on chromosome III (Fig. 3C). As for COs, GC in *sfr1-7D* mutants was similarly reduced to 49% of the GC levels observed in control crosses ($n = 6$, $P$ value = 2.10E-06), and this was statistically different from the reduction in the *sfr1* deletion mutants where GC was further diminished to 29% of the control levels ($n = 6$, $P$ value = 0.0002). In contrast, the *sfr1-7A* allele showed wild-type levels of GC ($n = 6$, $P$ value = 0.7943). The similar reduction in CO and GC levels of the *sfr1-7D* mutant suggests a primary defect in the invasion of the homologous chromosome. These genetic data indicate that CDK phosphorylation of Sfr1 reduces the efficiency of the Swi5–Sfr1 complex to promote the invasion of the homologous chromosome and, therefore, IH meiotic recombination.

## Sfr1 phospho-regulation modulates Rad51 and Sfr1 chromatin loading

The in vivo recombination results correlate with the in vitro Rad51 binding of the phospho-Sfr1 mutant proteins; therefore, they prompted us to analyze the chromatin loading of Rad51 during meiotic prophase using *pat1-114* synchronous meiosis. G1-arrested diploid cells with *sfr1-7D* or *sfr1-7A* alleles were induced to enter meiosis, and cells collected for chromosome spread preparation at different time points during meiotic prophase (3, 3.5, 4, and 4.5 h after meiotic induction) (Figs. 4A and EV2). *sfr1-7D* and *sfr1-7A* alleles did not alter meiotic progression and the strains showed the peak of meiosis I at 5 h after meiotic induction as *sfr1*⁺ cells (Appendix Fig. S2). In control and *sfr1-7A* kinetics, the number of nuclei showing Rad51 foci increased during prophase progression with the highest percentage at 3.5 h (86 and 87%, respectively), and then decreased to 36 and 45%, respectively, at 4.5 h (Fig. 4B). This pattern was extremely affected in *sfr1-7D* mutants where a strong reduction in Rad51 positive nuclei was observed, with the highest percentage at 4 h after meiotic induction (21%). In addition to Rad51, we profited to study EGFP-Sfr1 (and mutant versions) localization in the same spreads (Figs. 4A and EV2). The kinetics of EGFP-Sfr1 proteins paralleled that of Rad51 (Fig. EV3A); the percentage of nuclei showing EGFP-Sfr1 and EGFP-Sfr1-7A foci were maximal at 3.5 h after meiotic induction (90 and 95%, respectively), and a severe reduction was observed for EGFP-Sfr1-7D with only 30% of the nuclei showing foci.

We also analyzed the number, intensity, and colocalization of Rad51 and EGFP-Sfr1 foci (Figs. 4C,D and EV3B,C). In the control kinetics, the number of Rad51 and EGFP-Sfr1 foci per nucleus were maximal at 3.5 h after meiotic induction (Figs. 4C and EV3B). In addition, foci of both proteins showed a boosted signal and higher-intensity foci appeared (Figs. 3D and EV3C). At this time point, 78% of Rad51 foci co-localized with EGFP-Sfr1 foci ($n = 219$ foci; Pearson coefficient 0.7707 +/− 0.01, $n = 43$ nuclei). Similarly, in *sfr1-7A* kinetics the number of Rad51 and EGFP-Sfr1-7A foci were also maximal at 3.5 h after meiotic induction, when foci of both proteins also showed high intensity and 74% of Rad51 foci co-localized with EGFP-Sfr1-7A foci ($n = 330$ foci; Pearson coefficient 0.7142 +/− 0.01, $n = 54$ nuclei) (Figs. 4C,D and EV3B,C). Thus, *sfr1-7A* mutant behaves very similar to the control. By contrast, in the *sfr1-7D* mutant the number of Rad51 and EGFP-Sfr1-7D foci per nucleus were severely reduced at 3.5 h (Figs. 4C and EV3B). Moreover, these foci did not show the boosted intensity observed in control and *sfr1-7A* kinetics, and Rad51 and EGFP-Sfr1-7D focus intensity range were very narrow with the weakest and the brightest foci per nucleus showing similar intensity (Figs. 4D and EV3C). In addition, only 40% of Rad51 foci co-localized with EGFP-Sfr1-7D foci ($n = 20$ foci; Pearson coefficient 0.5532 +/− 0.06, $n = 6$ nuclei). Similar results were obtained analyzing Rad51 and EGFP-Sfr1 signal at 4 h after meiotic inductions; a statistically significant reduction in both the number and intensity of Rad51 and Sfr1-7D foci were observed (Figs. 4C,D and EV3B,C). *sfr1-WI* mutant behaved as the phospho-mimetic *sfr1-7D* mutant, and a dramatic reduction in the proportion of nuclei with Rad51 or EGFP-Sfr1-WI foci was also observed. In this case, the effect was more severe and nuclei with signals were barely detected at 3.5 h and 4 h after meiotic induction (Figs. 4A,B, EV2, and EV3A). Therefore, *sfr1-7D* and *sfr1-WI* mutants strongly affect the dynamics of Rad51 and Sfr1 proteins.

An increase in Rad51 foci intensity was specifically observed in the non-phosphorylatable *sfr1-7A* mutant at late prophase (4.5 h after meiotic induction) ($P$ value = 0.0071) (Fig. 4D). Only 4% of foci in the control compared to 22% of the foci in the mutant showed an intensity higher than 7000 (a.u.); this difference, though less pronounced, was also observed at 3.5 h and 4 h (Fig. 4D). By contrast, and accordingly to the data above, the intensity of Rad51 foci was clearly reduced in the *sfr1-7D* phospho-mimetic mutant from the beginning of prophase (Fig. 4D).

These results indicate that Sfr1 and Rad51 interact in vivo during meiotic prophase, and that phospho-regulation of Sfr1 is important to load or to stabilize Rad51 recombinase onto chromatin. In addition, since the chromatin accumulation of Sfr1-7D and Sfr1-WI proteins is also defective, the interaction between Sfr1 and Rad51 is required for the loading of the Swi5–Sfr1 complex.

## In vivo localization of Sfr1 and phospho-mutant proteins

To further study the phospho-regulation of Sfr1, and since Sfr1 localization was not previously reported in fission yeast meiotic cells, we analyzed in vivo its localization as well as the localization of the phospho-mutant versions. Using time-lapse microscopy, we followed the expression of EGFP-Sfr1 proteins in live cells (Fig. 5; Tables EV1 and EV2; Movie EV1–EV4). We detected EGFP-Sfr1 induction in early meiotic cells, 120 min before nuclei stopped the

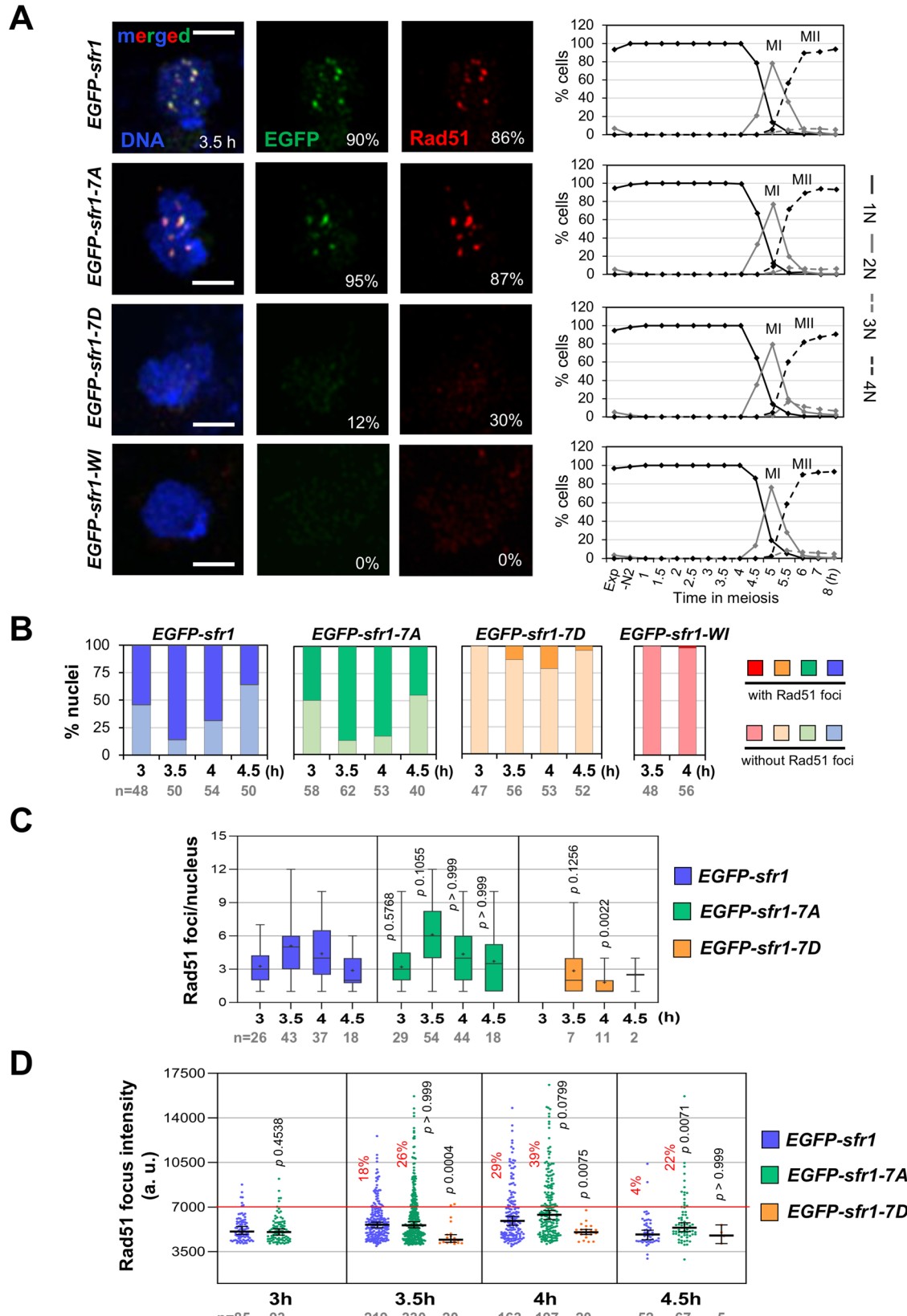

**Figure 4. Sfr1 phosphorylation impairs Rad51 recombinase loading onto chromatin.**

(A) *pat1-114 EGFP-sfr1* (CMC1649), *EGFP-sfr1-7A* (CMC1733), *EGFP-sfr1-7D* (CMC1756), and *EGFP-sfr1-WI* (CMC1769) diploid cells were induced to enter meiosis and collected at different time points during prophase for nuclear spread preparation. Spreads were stained with anti-GFP antibodies for the visualization of EGFP-Sfr1 proteins (in green) and with anti-Rad51 antibodies (in red). DAPI staining to visualize DNA is shown in blue. Representative images of nuclei at 3.5 h after meiotic induction are shown (maximum Z projections) on the left. Scale bars correspond to 2 μm. The percentages of nuclei showing foci for the corresponding proteins are indicated. Meiotic progression measured as the number of nuclei per cell is shown on the right; timing of meiosis I (MI) and meiosis II (MII) is indicated. (B–D) Dynamics of Rad51 signal during prophase in *sfr1* mutants. (B) The percentage of nuclei with (dark colors) and without (light colors) Rad51 foci is represented. (C) Quantification of the number of Rad51 foci per nucleus in the nuclei with signal. Data are represented by box-and-whisker plots where boxes extend from the 25th to 75th percentiles, and bars within the boxes represent the medians and black crosses the means; the whiskers represent the minimum and the maximal range. (D) Representation of the intensity of individual Rad51 foci. The median $+/-$ 95% confidence interval is shown. The threshold corresponding to 7000 a.u. is highlighted in red. At 3.5 h, 4 h and 4.5 h, the percentage of foci above the threshold is indicated. For all the graphs in the figure, the number of analyzed nuclei or foci (*n*) is indicated. Comparisons were done with the *EGFP-sfr1* control experiment, and *P* values were calculated based on Mann–Whitney test to compare results at 3 h (2 groups) and Kruskal–Wallis test (one-way nonparametric ANOVA) with Dunn's correction at the rest of the time points. Source data are available online for this figure.

characteristic meiotic *horsetail* movement (set as time 0 in the time lapses) (Ding et al, 2004), when signal was clearly reduced (Fig. 5A,B; Table EV1). EGFP-Sfr1 protein showed a pannuclear signal where a grainy pattern with clear accumulations was distinguished (Fig. 5A). We quantified this pattern (roughness, granularity) using the skewness parameter, as a measure of the asymmetry from the normal distribution (Fig. 5C; Table EV2). The skewness of EGFP-Sfr1 signal was positive, specifically at early prophase, indicating the presence of areas of higher intensity; and it was reduced as cells advanced in prophase and nuclei stopped moving (Fig. 5C). To explore the nature of this grainy signal, we analyzed EGFP-Sfr1 pattern in the absence of DSB formation, using a *rec12* deletion mutant (fission yeast *spo11* homolog, (Cervantes et al, 2000; Young et al, 2002)). In delta *rec12* cells, the skewness dynamics was lost, and EGFP-Sfr1 signal, although detected in the nuclei with similar total intensity, did not produce the grainy pattern neither the positive deviation from the normal distribution (Fig. 5A–C). These data suggest that EGFP-Sfr1 accumulation in some areas of the nuclei depends on DSB formation and they might represent sites for DSB repair. The non-phosphorylatable EGFP-Sfr1-7A protein showed also the grainy pattern, and a similar skewness dynamics to the wild-type protein with positive values at early prophase. In sharp contrast, the phospho-mimetic EGFP-Sfr1-7D protein behaved as the wild-type protein in the *rec12* deletion mutant showing not clear spots, and the skewness dynamics was completely lost (Fig. 5A,C). Importantly, EGFP-Sfr1-7A and EGFP-Sfr1-7D proteins were similarly expressed during meiotic prophase analyzed by signal intensity in time-lapse experiments as well as by western blot in *pat1-114* synchronous meiosis (Figs. 5B and EV4A; Table EV1). These cytological data in live cells, complement the above data using nuclear spreads, and point to an important role of CDK in the modulation of Sfr1 chromatin binding.

## Impact of Sfr1 phospho-regulation on Rad51 in vivo localization

We also analyzed Rad51 in live cells using a Rad51-ECFP version and time-lapse microscopy (Akamatsu et al, 2007) (Appendix Fig. S3A,B and Tables EV3 and EV4; Movies EV5–EV9). Rad51-ECFP was detected in early prophase ($-140$ min), and total intensity decreased as cells advanced and *horsetail* movement ended (time 0 in the time lapses), though reduction in total intensity was less dramatic than the one observed for EGFP-Sfr1 signal (Appendix Fig. S3B and Table EV3). As for EGFP-Sfr1, a

grainy pattern on top of a pannuclear signal was also distinguished (Appendix Fig. S3A; Yang et al, 2015). The Rad51-ECFP grainy pattern was also captured by the skewness parameter with a positive value; however, skewness was maintained along prophase compared to the skewness dynamics observed for EGFP-Sfr1 (Appendix Fig. S3B and Table EV4). As described in Yang et al 2015, this granny distribution was dependent on DSBs since, despite differences were less pronounced than in EGFP-Sfr1, it was statistically reduced in a *rec12* deletion mutant without affecting total intensity (Appendix Fig. S3A,B and Tables EV3 and EV4). Total intensity was similar in the *sfr1-7A* and *sfr1-7D* phospho-mutants compared to control zygotes (Appendix Fig. S3B and Table EV3). Levels of Rad51 were also analyzed by western blot in *pat1-114* synchronous meiosis. In control cells, Rad51 protein was clearly detected from the end of S-phase (2.5 h after meiotic induction) to the entry into the first chromosome segregation (meiosis I, 4.5–5 h), and similar pattern and levels were observed in *sfr1-7A* and *sfr1-7D* phospho-mutants (Appendix Fig. S3C). However, meanwhile skewness was very similar in control and *sfr1-7A* zygotes, this parameter was statistically higher in the *sfr1-7D* mutant (Appendix Fig. S3B and Table EV4). These data might indicate that Rad51 is loaded onto chromatin by alternative pathways in the phospho-mimetic *sfr1-7D* mutant, and that this loading is less resistant to the spreading protocol (Fig. 4A) (see "Discussion"). Rad51-ECFP showed a similar behavior in the *sfr1-WI* mutant (Appendix Fig. S3 and Table EV4).

## CDK modulates template choice for repair

So far, our data indicate that Sfr1 phosphorylation by CDK impairs the binding to Rad51, weakening its chromosome loading and, therefore, the formation of competent nucleofilaments for homolog invasion and repair. Since Sfr1 phosphorylation is higher at late prophase (Fig. 1), we envisioned that the inhibition of Sfr1 function would be more critical at this time of meiosis. One possibility is that homolog invasion is inhibited at this moment to prevent the formation of late IH JMs, that if not repaired on time would interfere with homolog disjunction in meiosis I. To explore this idea, we have modulated CDK levels to address the impact on partner choice for repair. We have used strains carrying an extra genomic copy of *cdc2*, an extra genomic copy of *cdc13* cyclin, or two extra genomic copies of *cdc13*, and the VL1 system to address recombination with the sister chromatid (Bustamante-Jaramillo et al, 2019; Latypov et al, 2010). This system consists of two truncated in tandem *ade6* fragments that generate a complete *ade6*

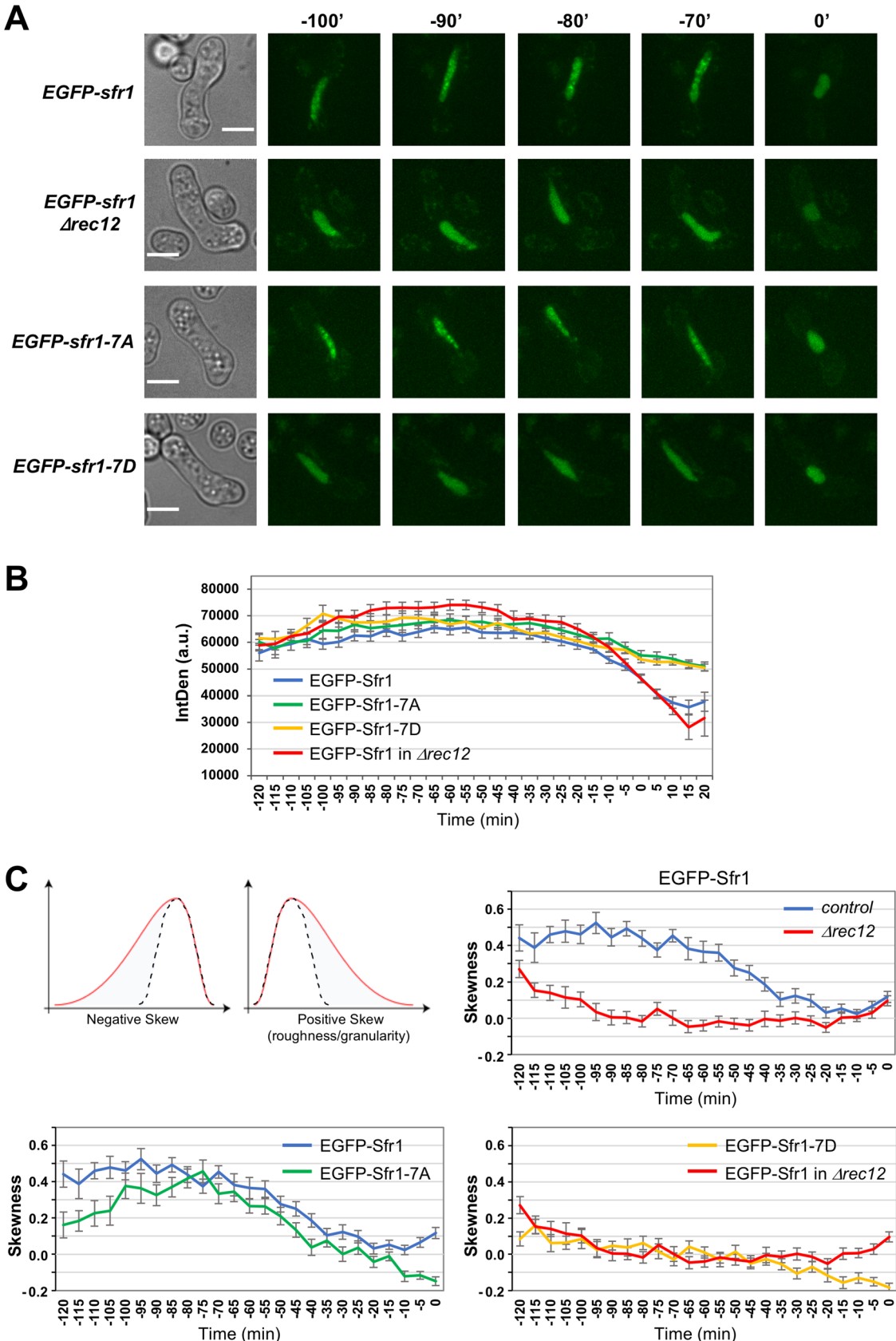

**Figure 5.  In vivo localization of EGFP-Sfr1 and phospho-mutant proteins.**

(A) Time-lapse microscopy of zygotes expressing tagged EGFP-Sfr1, EGFP-Sfr1-7A or EGFP-Sfr1-7D phospho-mutant versions in heterozygosis. Zygotes were obtained in crosses of *h⁻ EGFP-sfr1* (or phospho-mutant versions) × *h⁺* (or phospho-mutants) strains: control *EGFP-sfr1* (CMC1788 × CMC2), *EGFP-sfr1 Δrec12* (CMC1809 × CMC1808), *EGFP-sfr1-7A* (CMC1704 × CMC1791) and *EGFP-sfr1-7D* (CMC1712 × CMC1790). Time point 0 was set as the frame when nucleus stopped *horsetail* movement at the end of meiotic prophase. Representative images at time points in early prophase when roughness/granularity is observed for the EGFP-Sfr1 protein are shown (maximum Z projections). Scale bars correspond to 5 μm. Complete time lapses are shown in Movies EV1–EV4. (B) Quantification of EGFP-Sfr1, EGFP-Sfr1-7A and EGFP-Sfr1-7D total intensity in the same zygotes as in (A). Data are the mean $+/-$ SEM of n independent zygotes based on the cumulative numbers of several time-lapse experiments. In total, 6–36 zygotes analyzed at each time point, and at least 14 zygotes analyzed from time point −100 min to +15 min; n and P values are presented in Table EV1. (C) Upper left, representation of the skewness parameter (asymmetry from normal distribution). Skewness can be positive or negative, depending on the deviation to higher values (right part) or to lower values (left part) of the normal distribution, respectively. Upper right, representation of the skewness for the EGFP-Sfr1 signal in control (wild-type) and Δrec12 zygotes. Lower left, representation of the skewness for the EGFP-Sfr1 and EGFP-Sfr1-7A signals. Lower right, representation of the skewness for the EGFP-Sfr1 signal in Δrec12 zygotes and the EGFP-Sfr1-7D signal. Data are the mean $+/-$ SEM of n independent zygotes based on the cumulative numbers of several time-lapse experiments. In all, 9–36 zygotes analyzed at each time point, and at least 15 zygotes analyzed from time point −100 min; n and P values are presented in Table EV2. Source data are available online for this figure.

gene when intrachromosomal (IS) recombination occurs. As shown in Fig. 6A, the levels of IS repair were increased in cells carrying extra copies of *cdc2* or *cdc13*. One extra genomic copy of *cdc2* produced an increase of 49% ($n = 12$, $P$ value = 0.0029), one extra genomic copy of *cdc13* produced an increase of 38% ($n = 12$, $P$ value = 0.0327), and two extra genomic copies of *cdc13* produced an increase of 69% ($n = 7$, $P$ value = 0.0055) in intrachromosomal recombination. This statistically significant increment cannot be explained by a general increase in the levels of recombination, rising for example DSB levels, since the levels of GC were not correspondingly elevated by the extra genomic copies of *cdc2* or *cdc13* (Fig. 6B). These genetic data indicate that CDK modules partner choice for repair, favoring the repair with the sister chromatid.

## Sfr1 phosphorylation contributes to the timely removal of Rad51 nucleofilaments at the end of the meiotic prophase

If our model is correct, we would expect non-phosphorylatable Sfr1-7A protein to maintain more stable interactions with Rad51 at late prophase that would interfere with proper chromosome disjunction. To study this, we analyzed chromosome segregations and Rad51 retention in anaphase I and II. $h^{90}$ control *sfr1⁺*, *sfr1-7A*, *sfr1-7D*, and *sfr1* deletion cells were induced to enter meiosis and collected for immunostaining. Zygotes with anaphase I and anaphase II spindles were selected, and chromatin distribution was followed by DAPI staining. As shown in Fig. 7A, only in the *sfr1* deletion mutant a defect in chromosome segregation was observed with 46% of zygotes correctly segregating two equal DNA masses to opposite poles of the anaphase I spindle compared to 81% in control zygotes. Given the defect of the *sfr1-7A* mutant in Rad51 dynamics at the end of prophase (Fig. 4D), we decided to address segregation defects in combination with the *dbl2* deletion mutant, which represents a compromised situation with Rad51 retention and segregation defects at meiosis I (Polakova et al, 2016). As previously described, *dbl2* mutants showed segregation defects at anaphase I with 64% of zygotes showing lagging chromatin, fragmented DNA masses or unequal segregations compared to 19% of these type of errors in $h^{90}$ control zygotes. This was aggravated in *dbl2 sfr1-7A* mutants where even 45% of zygotes showed a single DNA mass, a phenotype rarely observed in *dbl2* single mutants (2.4%) and absent in $h^{90}$ control zygotes (Fig. 7A). The phenotype was even more dramatic in anaphase II where no zygotes with properly segregated DNA masses were observed. Indeed, 89% of

*dbl2 sfr1-7A* zygotes showed two spindles with a single DNA mass at anaphase II (Fig. 7B), a phenotype associated to segregation problems in anaphase I (Polakova et al, 2016). By contrast, *sfr1* deletion and *sfr1-7D* mutants alleviated the segregation problems of the *dbl2* mutant. The percentage of zygotes with proper chromosome segregation at anaphase I rose from 33% in *dbl2* to 53% and 57%, respectively, in the double mutants; and segregation problems were also improved in anaphase II (Fig. 7A). The segregation defects in the *dbl2 sfr1-7A* mutant correlated with the retention of Rad51 at anaphase I (Fig. 7C). In total, 78% of *dbl2 sfr1-7A* zygotes showed Rad51 foci (associated to the missegregated DNA) compared to 18% of *dbl2* single mutants; moreover, the *sfr1-7A* single mutant also showed a 12% of zygotes with Rad51 focus persistence. These results indicate that CDK downregulation of Swi5–Sfr1 function is important to unload Rad51 from chromatin, likely to dismantle IH invasions, facilitating chromosome segregation.

Finally, we studied the impact of these alterations on the final meiotic products by addressing the spore viability. For this, we plated spores in rich solid media and scored for microcolonies with at least five cells after 26 h of incubation. Spores from *dbl2* mutant cells showed 42% microcolonies compared to the 94% observed in control *sfr1⁺* ($n = 3$, $P$ value = 0.0003), and 46% of the spores did not germinate (Fig. 7D). This was specifically worsened in the double *dlb2 sfr1-7A* mutant with only 15% microcolonies ($n = 3$, $P$ value = 0.0033) and up to 73% of ungerminated spores. Similarly, as for the segregation problems described above, *sfr1* deletion and *sfr1-7D* mutants alleviated the phenotype of the *dbl2* deletion with 75% ($n = 3$, $P$ value = 0.0016) and 61% ($n = 3$, $P$ value = 0.0256), respectively, of microcolonies (Fig. 7D). Single *sfr1-7A* and *sfr1-7D* mutants were indistinguishable from the *sfr1⁺* control, and *srf1* deletion mutants showed a moderate reduction in spore viability (73% of microcolonies; $n = 3$, $P$ value = 0.0033).

## The disordered N-terminal part of Sfr1 is a hub for protein regulation

When studying the in vivo localization of EGFP-Sfr1 proteins, we realized that both EGFP-Sfr1-7A and EGFP-Sfr1-7D proteins were misregulated. EGFP-Sfr1 total intensity decreased at the end of the meiotic prophase; however, this was not the case for EGFP-Sfr1-7A and EGFP-Sfr1-7D proteins (Fig. 5B), which remained clearly visible in cells in meiosis I and meiosis II (Fig. EV4B; Movies EV1–EV4). This was confirmed by western blot analysis

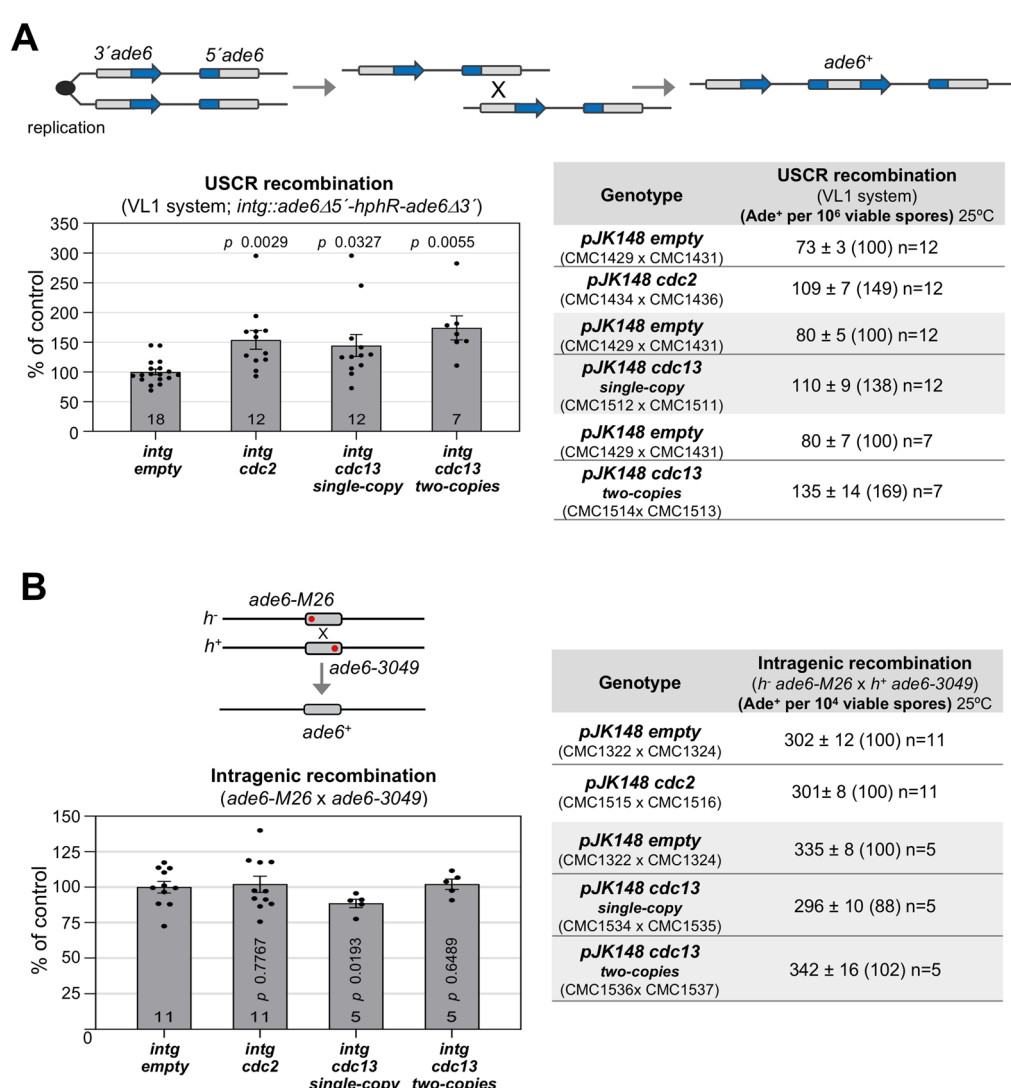

**Figure 6. CDK modulates intrachromosomal recombination.**

Schemes of the recombination assays are shown. Crosses were performed in MEA at 25 °C and plated for recombination analysis twice; the results are based on the cumulative numbers in each cross. Tables show recombination frequencies $+/-$ SEM of n independent crosses, with numbers in parentheses showing the percentages relative to controls (*pJK148 empty* integrants). Each integrant was analyzed only with control crosses in the same experiment. Strains used in the crosses are indicated. Graphs show recombination expressed as the mean of the percentage relative to the control cross $+/-$ SEM of n independent crosses. *P* values were calculated based on Student's *t* test (unpaired, two tails). (**A**) Intrachromosomal recombination assay using the VL1 system (Latypov et al, 2010). Crosses of $h^+$ *ade6-D20* $\times h^-$ *ade6-D20 VL1* stains carrying extra genomic copies of *cdc2* or *cdc13* (*pJK148* integrants). Unequal sister chromatid recombination (USCR) levels are expressed as the mean of Ade$^+$ colonies per $10^6$ viable plated spores; 22–132 Ade$^+$ recombinant colonies scored in each independent cross, and 336–876 total Ade$^+$ recombinant colonies scored per genotype. (**B**) Intragenic recombination (GC). Crosses of $h^-$ *ade6-M26* $\times h^+$ *ade6-3049* stains carrying extra genomic copies of *cdc2* or *cdc13* (*pJK148* integrants). GC levels are expressed as the mean of Ade$^+$ colonies per $10^4$ spore colonies; 70–313 Ade$^+$ recombinant colonies scored in each independent cross, and 1160–2333 total Ade$^+$ recombinant colonies scored per genotype. Source data are available online for this figure.

of *pat1-114* synchronous meiosis. Meanwhile EGFP-Sfr1 protein levels decreased when cells were entering meiosis I (4.5–5 h after meiotic induction), the expression of both phospho-mutant proteins was maintained at later time points until the end of the time courses (Fig. EV4A). These data suggest that a region required for Sfr1 stability is somehow altered in the phospho-mutants.

In our experiments, we did also study the localization of the EGFP-Sfr1-WI protein, which in vitro is strongly impaired in Rad51 binding (Fig. 3A) (Argunhan et al, 2020). We expected this protein to be a good control for our phospho-mutant proteins.

Sfr1-WI is proficient for DNA repair upon UV irradiation in vegetative growth unless the Rad55–Rad57 pathway for Rad51 loading is also compromised (Argunhan et al, 2020). However, the meiotic proficiency of the protein had not been explored. The *sfr1-WI* mutant was strongly impaired in meiotic homologous recombination, reducing COs to 27% of the control levels ($n = 6$, *P* value $= 7.38E-08$) (Appendix Fig. S4A). This was significantly lower than the reduction observed in the phospho-mimetic *sfr1-7D* allele ($n = 6$, *P* value $= 0.0002$), and similar to the reduction in the *sfr1* deletion mutant ($n = 3$, *P* value $= 0.1130$). This anticorrelates

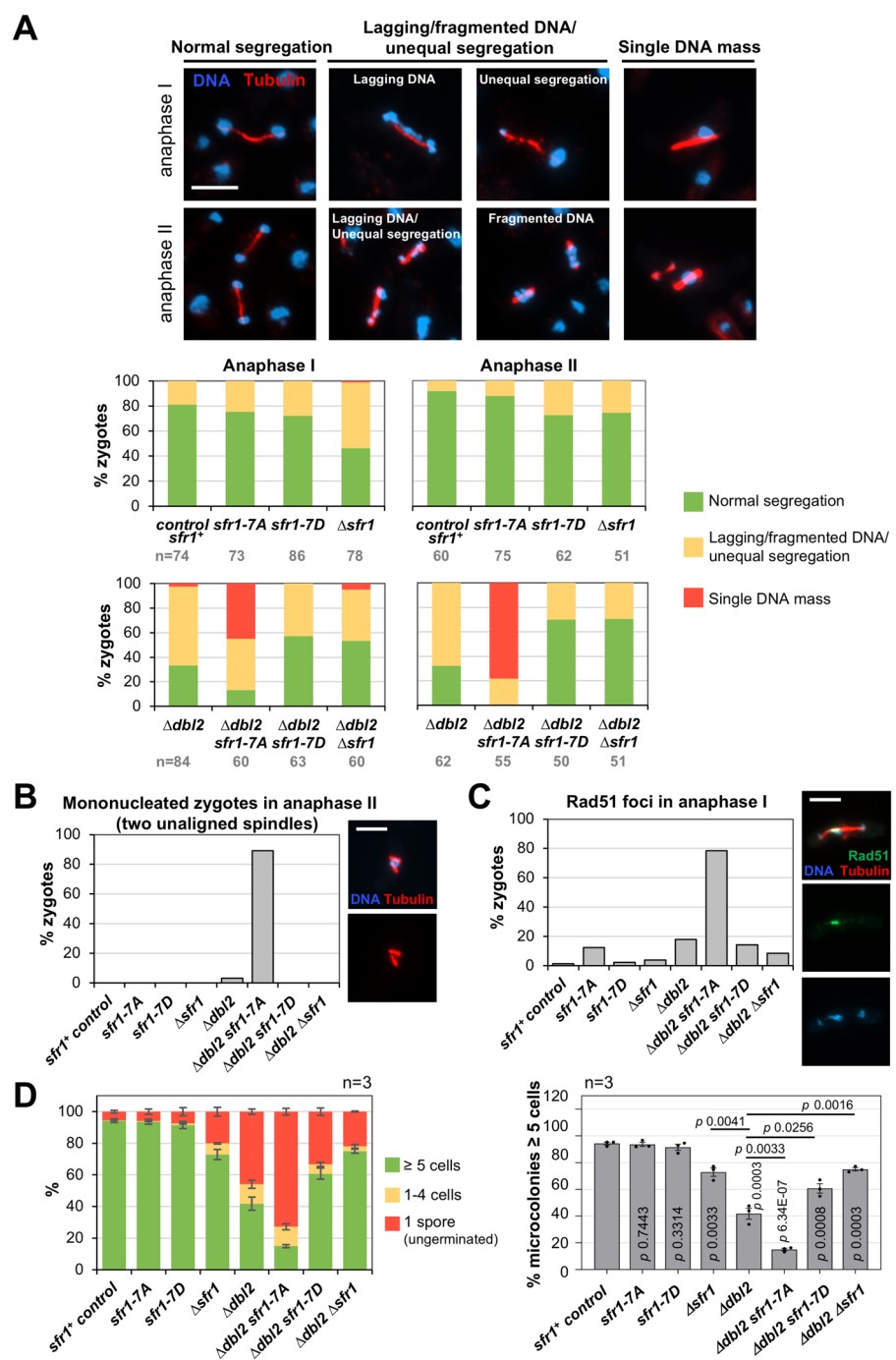

**Figure 7. Sfr1 phosphorylation affects Rad51 disassembly and DNA segregation.**

(A–C) Homothallic $h^{90}$ $sfr1^+$ (CMC1843), $sfr1$-7A (CMC1847), $sfr1$-7D (CMC1848), $sfr1$ deletion (CMC1845) and $dbl2$ deletion (CMC1860) cells, and double $dbl2$ $sfr1$-7A (CMC1862), $dbl2$ $sfr1$-7D (CMC1859) and $dbl2$ $sfr1$ (CMC1861) cells were grown in PMG-NH$_4$Cl plates to induce meiosis (asynchronous meiosis). Fixed zygotes were immunostained with anti-tubulin antibodies for spindle visualization (red) and anti-Rad51 antibodies (green), and DAPI stained for chromatin visualization (blue). For all the images in the figure, scale bars correspond to 5 µm. (A) Representative images of zygotes with normal segregations and the scored defective segregations are shown in the upper part. The percentage of zygotes with normal and defective segregations at anaphase I and II is represented in the lower part (left graphs and rights graph, respectively). The numbers of total analyzed zygotes are indicated (*n*). (B) Representation of the percentage of mononucleated zygotes containing two spindles (unaligned) at anaphase II. A representative image of a zygote showing this phenotype is shown on the right. (C) Representation of the percentage of zygotes at anaphase I with Rad51 foci. A representative image of a zygote with Rad51 retention is shown on the right. (D) Spore viability of the same homothallic $h^{90}$ strains as above. Cells were grown at 25 °C on SPA plates to induce sporulation. Spores were plated in YES solid media and scored for colony formation under the microscope after 26 h at 32 °C. Graph on the left shows the percentage of spore colonies in the indicated categories. Data are the mean +/− SEM of three independent experiments. In total, 150–182 spore colonies scored in each experiment, and 466–493 total spore colonies scored per genotype. Graph on the right shows the percentage of spores that generated microcolonies with at least five cells. *P* values were calculated based on Student's *t* test (unpaired, two tails). Source data are available online for this figure.

with the pulldown assays where, although strongly affected, the Rad51 interaction was detectable compared to the Sfr1-7D protein (Fig. 3A) (Argunhan et al, 2020). Indeed, this anticorrelation is also observed when analyzing Rad51 chromatin binding (Fig. 4). We would not expect a worse recombination rate and Rad51 chromatin binding for the *sfr1-WI* allele compared to the *sfr1-7D* mutant. This prompted us to check for the localization of the protein by time-lapse microscopy. As shown in Appendix Fig. S4B, EGFP-Sfr1-WI protein barely entered the nucleus. However, we confirmed by western blot analysis of *pat1-114* synchronous meiosis that the protein was normally expressed (Appendix Fig. S4C). These results indicate that Sfr1 has a more prominent role in the repair of meiotic DSBs than in the repair of DNA damage in vegetative growth; and more importantly, that the first in vivo defect of the Sfr1-WI protein is a failure to enter the nucleus.

Collectively, these data indicate that the disorganized N-terminal part of Sfr1 functions as a hub for the regulation of the protein, controlling both the stability of the protein as well as its nuclear localization, and as described above, the modulation by CDK of the binding to Rad51 recombinase.

# Discussion

## CDK phosphorylation of Sfr1 downregulates Rad51 function and interhomolog meiotic recombination

Meiotic recombination must be coordinated with cell cycle progression to ensure that recombination intermediates are accurately resolved before homologs segregate at meiosis I. This coordination guarantees the production of healthy gametes with a normal chromosome content and high fertility potential. In this report, we describe a novel mechanism involved in this essential control. We show that during meiotic prophase, a key accessory factor for the formation and activity of Rad51 nucleofilaments is phospho-regulated by CDK activity, the master driver of cell cycle progression.

The Sfr1 protein, a component of the conserved Swi5–Sfr1 complex, is highly phosphorylated during meiotic prophase in a Cdc2-dependent manner and, indeed, Sfr1 is an in vitro substrate for Cdc2-Cdc13 complexes (Figs. 1 and 2). This phosphorylation impacts on Rad51 recombinase binding since the phospho-mimetic Sfr1-7D protein losses its binding capability compared to the efficient interaction of a non-phosphorylatable Sfr1-7A version (Figs. 3A and EV1A). As a consequence, Rad51 stable loading onto meiotic chromosomes is severely impaired as shown by the dramatic reduction in the number and intensity of Rad51 foci in nuclear spreads (Figs. 4 and EV2); and the phospho-mimetic *sfr1-7D* mutant reduces recombination with the homologous chromosome, measured genetically as GC and CO levels (Fig. 3B,C). These results indicate that Sfr1 phosphorylation has an inhibitory effect on IH meiotic recombination.

In meiosis, 13 residues have been found phosphorylated in Sfr1 protein during prophase (Sfr1-TAP purification) (Sevcovicova et al, 2021). Therefore, we expected Sfr1-7A mutant protein (containing mutations in 6 of these sites) to retain some retardation in Phos-tag gels. However, this was not the case and mobility shift of the mutant protein was almost abolished (Fig. 1D). Thus, it is possible that CDK is priming for other phosphorylations. It is known that

CDK sites prime for DDK (Dbf4-dependent kinase) and Polo kinases. This has been well established for proteins involved in meiotic DSB formation (Mer2; Sasanuma et al, 2008; Wan et al, 2008) and DNA damage repair (Mms4; Gallo-Fernandez et al, 2012). Three of the seven residues mutated in the *sfr1-7A* allele are Thr-phospho Thr/Ser-Pro sites (T-pT/pS-P), and therefore, a priming role for additional phosphorylation can be recalled. These sites are Thr89, Ser147, and Thr152; and adjacent Thr146 and Thr151 residues could be also in vivo phosphorylated (Ser147 (or Thr146) and Thr152 (or Thr151) could not be assigned to an individual amino acid, Sevcovicova et al, 2021).

The non-phosphorylatable Sfr1-7A protein maintains substantial Rad51 interaction in pulldown and Y2H assays (Figs. 3A and EV1A), and exhibits high GC and CO levels (Fig. 3B,C); however, some alterations are observed in this mutant. Although Rad51 loading onto chromatin is mainly conserved in early prophase considering the number and intensity range of Rad51 foci, the intensity is particularly increased at late prophase due to an accumulation of brighter Rad51 foci (Fig. 4). These results indicate that Sfr1 phospho-regulation is dynamic and that a non-phosphorylatable protein impairs Rad51 chromosome-loading dynamics at the end of prophase. Supporting this, *sfr1-7A* enhances the segregation defects of the *dbl2* deletion mutant, which is required for chromosome removal of Rad51 prior to chromosome segregations (Polakova et al, 2016). Notably, most of double *dbl2 sfr1-7A* zygotes show an abnormal DNA distribution in anaphase I (Fig. 7A). Indeed, in anaphase II *dbl2 sfr1-7A* zygotes present two spindles and a single DNA mass, indicative of problems carried from meiosis I (Fig. 7B). Moreover, Rad51 persistence is also enhanced in the *dbl2 sfr1-7A* mutant where an extremely high number of zygotes retain Rad51 foci after prophase in cells with anaphase I spindles (Fig. 7C). This severely impacts on the spore viability (Fig. 7D).

We have also described the in vivo localization of Sfr1 during meiosis (Fig. 5; Tables EV1 and EV2; Movies EV1–EV4) and found that the EGFP-Sfr1 protein forms aggregates during prophase in live cells; these aggregates are captured by the skewness parameter (deviation from normal intensity distribution), and they depend on DSB formation since they are lost in a *rec12* mutant. Furthermore, there is a good correlation between the formation of these aggregates, Sfr1 phosphorylation status, and Rad51 binding (Figs. 3A, 5, and EV1A), as also shown for Sfr1 foci formation in nuclear spreads (Figs. 4A and EV3). This indicates that binding of the unphosphorylated Sfr1 protein to Rad51 is required for the chromosome loading of the Swi5–Sfr1 complex. Thus, Sfr1 and Rad51 are mutually interdependent for loading during meiotic prophase, and this is modulated by the phosphorylation status of Sfr1. The unphosphorylated Sfr1 protein, in complex with Swi5, would efficiently bind to Rad51 stabilizing the invasion nucleofilament as well as promoting its strand exchange activity, and Rad51 binding would allow Sfr1 chromosome loading. The interdependency of Sfr1 and Rad51 for foci formation was previously reported in mitotic cells upon DNA damage using deletion mutants but the regulation of this interaction was unknown (Akamatsu et al, 2007).

Collectively, our data support a model whereby Sfr1 N-terminal phospho-regulation is critical for the activity of the Swi5–Sfr1 mediator and the physiology of the Rad51 nucleofilament (Fig. 8). The fact that the Sfr1 N-terminal domain remains outside of the nucleofilament groove, together with the flexibility conferred by the

lack of secondary structure (Argunhan et al, 2020; Kokabu et al, 2011; Kuwabara et al, 2012), makes this domain accessible to the action of proteins, as CDK complexes. Since the phosphorylation is maximal at the end of prophase, and performed at least by Cdc2-Cdc13 complexes whose activity is maximal just prior to the first meiotic chromosome segregation, we propose that CDK phosphorylation downregulates late IH invasions. This mechanism would facilitate the repair of late recombination intermediates by other faster/simpler repair pathways (SDSA) or more compatible with the segregation of homologous chromosomes (sister chromatid recombination). The repair by alternative pathways is supported by the fact that spore viability is maintained in the *sfr1-7D* mutant, and only moderately reduced in *sfr1* deletion mutants despite the strong reduction in COs (Figs. 3B and 7D; Appendix Fig. S4A) (Lorenz et al, 2014). Regarding this, CDK activity modulates partner choice for repair since a moderate increase in Cdc2 or Cdc13 raises recombination levels with the sister chromatid (Fig. 6A). We also propose that Sfr1 phospho-regulation collaborates with UvrD-type DNA Fbh1 helicase, whose loading is mediated by Dbl2, in dismantling Rad51 recombination intermediates (Polakova et al, 2016; Sun et al, 2011; Tsutsui et al, 2014). In fact, both mechanisms might be CDK regulated since Dbl2 is potentially a good candidate as a Cdc2 substrate; Dbl2 harbors 14 putative CDK sites and 3 of them are phosphorylated in vegetative growth (https://www.pombase.org/gene/SPCC553.01c). Related to this, the budding yeast UvrD-type DNA Srs2 helicase, which promotes SDSA repair by D-loop disassembly, is regulated by CDK (Saponaro et al, 2010). Finally, it is also possible that Sfr1 phospho-regulation plays a role in the normal repair process earlier in prophase as a necessary step to remove Rad51 from the recombination intermediates facilitating downstream events such as DNA polymerases, ligases, translocases or resolvases entry. In addition, Rad51 interaction with the Swi5–Sfr1 complex is weak and proposed to reflect a dynamic binding to confer flexibility to the interaction during nucleofilament elongation (Kuwabara et al, 2012; Argunhan et al, 2020). This dynamics might be also phospho-regulated and altered in the *sfr1-7A* mutant. This could explain the small reduction in CO formation of the *sfr1-7A* allele (Fig. 3B).

It is worth noting the different behavior of Rad51 in *sfr1-7D* phospho-mimetic mutants in time-lapse experiments and in the analysis of nuclear spreads (Figs. 4A–C and EV2; Appendix Fig. S3). Meanwhile foci of Rad51 are mostly lost when analyzed by nuclear spreading, in time-lapse experiments the granularity captured by the skewness parameter is even higher than in the control. This suggests that Rad51 foci in this mutant are greatly lost during the spreading preparation and, given the defective binding of Sfr1-7D protein to Rad51 in the pulldown and Y2H assays (Figs. 3A and EV1A), that the recombinase is primarily loaded onto chromatin by alternative pathways. This unstable Rad51 loading in the phospho-mutant might be the one involved in SDSA or IS recombination. Meanwhile, the remaining Rad51 foci, which resist the spreading (stable loading), could explain the COs still present in the mutant.

In this study, we have used *sfr1* mutants harboring changes in all the putative CDK sites (Thr73, Thr89, Ser109, Ser116, Ser147, Thr152, and Ser165), four of them previously identified in different phosphoproteomic studies conducted in vegetative cells (Carpy et al, 2014; Cipak et al, 2009; Kettenbach et al, 2015; Koch et al, 2011; Swaffer et al, 2016) (Fig. 1A). During the course of the

project, except Thr89, they were all found in vivo phosphorylated during meiotic prophase (Sfr1-TAP purification analysis) (Sevcovicova et al, 2021). These residues are located within or in closed proximity to Site 1 and Site 2, two short protein stretches described by structural analysis to mediate Rad51 binding (Argunhan et al, 2020). These sites contain positively charged amino acids (Arg and Lys) that mediate electrostatic interactions with Rad51 and, when mutated, the in vitro interaction with the recombinase is strongly reduced. Based on the nature of Sfr1 and Rad51 interaction, we propose that Sfr1 phosphorylation counterbalances the positive charges involved in Rad51 binding, providing a molecular switch for Sfr1 function in homologous recombination repair (Fig. 8). This has been also recently proposed by Liang and colleagues (Liang et al, 2023). They have reported that the efficiency of the binding to Rad51, the stability of the Rad51 nucleofilament, and the strand exchange activity are modulated by the phosphorylation status of Sfr1. Using in vitro assays with bacterial purified proteins, they have shown that a phospho-mimetic Sfr1 protein, in 5 reported phosphorylated sites, impairs all these functions of Sfr1. This mutant contains four of the residues we have studied (Thr73, Ser109, Ser116, and Ser165) in addition to a non-CDK site. These molecular data complement our in vivo data that demonstrate the impact of Sfr1 phosphorylation on Rad51 dynamics during meiosis and its role in coordinating meiotic progression with efficient meiotic recombination. Based on this recent work and the high conservation of several CDK sites in the genus *Schizosaccharomyces* (Appendix Fig. S5), it is formally possible than a 5A/5D mutant (Thr73, Ser109, Ser116, Ser147 and Ser165) could recapitulate the phenotypes of our 7A/7D mutants. Finally, CDK phospho-regulation of the Swi5–Sfr1 complex is likely exerted exclusively through Sfr1, since only two phosphorylated sites have been reported in Swi5 (Ser72 and Ser84; Cipak et al, 2009; Sevcovicova et al, 2021) and these are not CDK sites. It is currently unknown the function of this phosphorylation. Furthermore, Swi5 protein does not harbor putative CDK sites.

## The disordered N-terminal domain of Sfr1 is a key platform to coordinate different aspects of Sfr1 regulation and function

The *sfr1-WI* mutant shows a defect in homologous recombination nearly as strong as the deletion mutant that correlates with a defect in the normal nuclear localization of the protein during meiotic prophase (Appendix Fig. S4). The defect in meiotic recombination was striking given that this mutant does not show sensitivity to DNA damage during vegetative growth unless additional factors for Rad51 recombinase loading were downregulated (Argunhan et al, 2020). This is also the case for the new *sfr1* phospho-mutant alleles generated by Liang and colleagues (Liang et al, 2023). These results pinpoint the prominent role of Sfr1 in meiosis *versus* the mitotic cycle. Furthermore, it suggests a differential regulation of the nuclear localization of the protein; in sharp contrast to the *sfr1-WI* mutant, the deletion of *sfr1* confers sensitivity to DNA damage in vegetative cells, indicating that Sfr1-WI protein enters the nucleus.

Nuclear localization signals (NLS) are short basic regions, a shared feature with Site 1 and Site 2. Indeed, NLS servers predict an NLS at Site 1 in the N-terminal part of Sfr1 protein, containing mutated residues in the *sfr1-WI* mutant (92S**KKR**AR97). Thus, this region seems to have a dual function. It primarily controls Sfr1

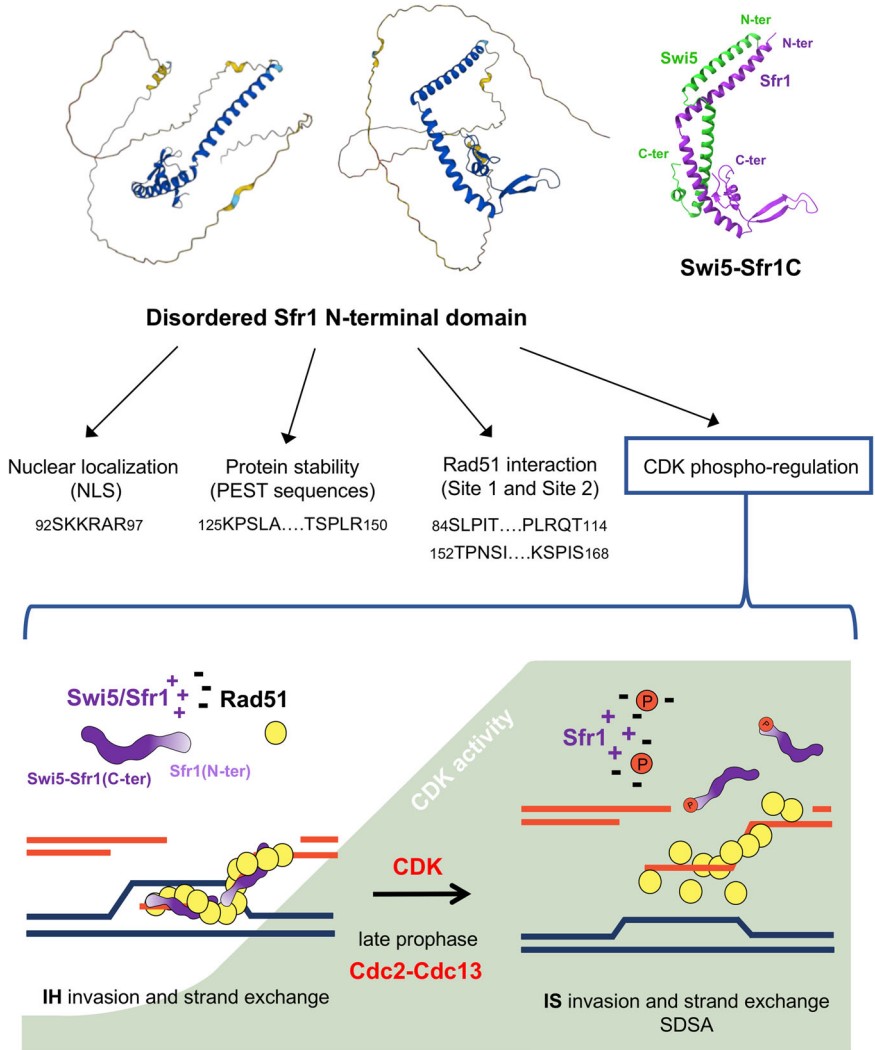

**Figure 8. Model of Sfr1 phospho-regulation.**

Top: The resolved structure of the Swi5–Sfr1 complex without the N-terminal disordered domain of Sfr1 (Swi5-Sfr1C) is shown on the right (Swi5 in green and Sfr1 in purple; Protein Data Bank PDB 3viq). Different views of Sfr1 showing the large disordered N-terminal part of the protein (AlphaFold modeling, color bfactor palette) are shown on the left (https://alphafold.ebi.ac.uk/entry/Q9USV1). This domain controls several aspects of the protein regulation: nuclear localization, protein stability, Rad51 recombinase interaction and CDK phospho-regulation. Bottom: Model of CDK downregulation of Sfr1 function. During prophase, repair of DSBs with the homologous chromosome requires that the Swi5–Sfr1 complex promote the stability as well as the strand exchange activity of the Rad51 nucleofilament. The complex interacts with Rad51 through positive charges in the disordered N-terminal part of Sfr1. At late prophase, accumulation of CDK activity (Cdc2–Cdc13) phosphorylates Sfr1 at sites close to the positive charges involved in the interaction. As a consequence, the stability of the invasion nucleofilament decreases, hampering the interaction with the homologous chromosome and favoring the repair by alternative pathways (SDSA or sister chromatid repair). This mechanism would prevent the formation of IH recombination intermediates at the end of prophase that, if not repaired on time, would interfere with the segregation of the homologs.

nuclear localization and, once in the nucleus, controls the formation and activity of the Rad51 nucleofilament. Site 1 and 2, and the positively charged residues within, are conserved among species of the genus *Schizosaccharomyces* (Argunhan et al, 2020); therefore, control of the nuclear localization of the protein might be also a conserved feature. Accordingly, NLSs are also predicted in these species at similar position (Appendix Fig. S5).

Our work with the *sfr1* phospho-mutants shows an unpredicted result. Although the *sfr1-7A* and *sfr1-7D* mutants behave differently in terms of homologous recombination and chromatin binding (Figs. 3B,C, 4A, 5, EV2, and EV3), both Sfr1-7A and Sfr1-7D proteins are present later in meiosis compared with the wild-type

protein that disappears during the first chromosome segregation at meiosis I (Fig. EV4; Movies EV1–EV4). Thus, N-terminal part of Sfr1 controls stabilization of the protein, and the changes introduced in the phospho-mutants might alter this regulation. Indeed, the region harbors predicted PEST sequences (Pro, Glu, Ser and Thr enriched tracks) that are also present at the same position in *S. japonicus, S octosporus, S. cryophilus* and *S. osmophilus*, and profusely detected in the N-terminal part of the mouse ortholog (EMBOSS explorer) (Appendix Figs. S5 and S6) (Akamatsu and Jasin, 2010). Regulation of Sfr1 protein stability at the end of meiotic prophase might represent an additional mechanism to inhibit late IH invasions.

The E3-Ubiquitin ligase SCF is involved in the signaling for ubiquitylation and degradation of proteins containing PEST sequences, and it plays a role in meiotic recombination from yeast to higher eukaryotes (Guan et al, 2022; Okamoto et al, 2012). In fission yeast, a temperature-sensitive mutant in the conserved SCF-component Skp1 shows segregation defects at meiosis I that are corrected when DSB formation is abolished, and moreover, *skp1* defective cells enter meiosis I with persistent Rad51 foci (Okamoto et al, 2012). The misregulated SCF targets involved in this phenotype are currently unknown and Sfr1 would be a candidate. Since nuclear localization and protein stability are regulated by phosphorylation, and in some cases in a CDK-dependent manner (Benito et al, 1998; Faustova et al, 2022; Lanker et al, 1996; Sutani et al, 1999; Yaglom et al, 1995), it would be interesting to explore CDK modulation of these aspects of Sfr1 regulation, particularly studying the contribution of individual conserved phosphorylation sites as Ser109 (adjacent to the predicted NLS sequence) and Ser147 (within the predicted PEST sequence).

Rad51 interaction, nuclear localization, and protein stability are regulated through the intrinsically disorganized N-terminal part of Sfr1, where Site 1 and 2, CDK phosphorylated residues, NLS and PEST sequences are located (Fig. 8). Disordered N-terminal regions are also present in Sfr1 orthologs from yeast to human (Argunhan et al, 2020). Since some functions of this domain are conserved, as the binding to Rad51 and Dmc1 of the budding yeast ortholog Mei5 (Hayase et al, 2004; Say et al, 2011), this might be also the case for other functions. Thus, the N-terminal domain in Sfr1 proteins might serve as a platform for the binding to different regulatory factors. Indeed, the flexibility of disordered regions is proposed as a key feature to provide plasticity/versatility to the proteins, facilitating the interaction with high specificity and very dynamically with different partners; in addition, they are enriched in post-translational modifications that modulate the interactions (Gsponer and Babu, 2009; Oldfield and Dunker, 2014; Wright and Dyson, 2015). Moreover, NLSs and determinants for protein degradation are frequently associated to these regions (Guharoy et al, 2016; Wubben et al, 2020; Yamagishi et al, 2015). There is a growing list of proteins regulated by disordered domains and involved in a wide variety of biological processes. In meiosis, it has been recently reported that the disordered C-terminal domain of the repair MSH-5 protein is phosphorylated by CDK-2, and this phosphorylation is important for the accumulation of pro-CO factors at the recombination sites (Haversat et al, 2022). Sfr1 represents a paradigm of proteins containing structurally disorganized domains, where several of the features associated to these domains are present.

## Concluding remarks

As a key factor for the regulation of Rad51 recombinase and homologous recombination, we have shown that Sfr1 is regulated at different levels during meiosis, and that this regulation is exerted through its N-terminal disordered domain. It is heavily phosphorylated at the end of prophase by CDK to restrict Swi5–Sfr1 function and to dismantle the stable loading of Rad51 onto chromatin. This is important to avoid the entrance into meiosis I with IH recombination intermediates that could hamper chromosome segregation and compromise gamete viability. In addition, we

report that nuclear localization and protein stability are also important points for protein regulation. Based on sequence comparisons and the presence of motifs, these features might be conserved in other eukaryotes apart from our working model organism, the fission yeast *S. pombe*. Collectively, our study expands the current knowledge on the regulation of conserved Rad51-accessory factors involved in recombinational repair. Furthermore, since in most species the Swi5–Sfr1 complex is not meiosis-specific, the results not only help to understand how genome integrity is maintained during meiosis to ensure the generation of healthy gametes, but more broadly in response to DNA damage.

# Methods

## Yeast manipulation, mutant construction, and general methods

Experimentally required strains were obtained by meiotic crosses. Strains used in this study are listed in Appendix Table S1. Cells were grown in yeast extract medium with supplements (YES) or Edinburgh minimal medium (MM) with supplements at 32 °C or 25 °C (for temperature-sensitive mutants). Normal supplements were Adenine, Leucine, Uracil, Lysine, and Histidine (225 mg/l). YES supplemented with 0.1 mg/ml Hygromycin B, G-418 or Nourseothricin was used to follow gene deletions and insertions. Genetic crosses were done in malt extract plates with supplements (MEA-4S) or sporulation agar (SPA) at 25 °C for 3 days. Spore viability was scored by plating $2 \times 10^4$ spores in YES media, incubation at 32 °C for 26 h, and microscopic examination to score for microcolony formation. *sfr1* mutant sequences and their *EGFP* fused versions (in frame before the ATG) were ordered to Integrated DNA Technologies (IDT) as synthetic DNA fragments. In the case of the N-terminal *EGFP-sfr1* wild-type version, the strain was obtained from Dr. Hiroshi Iwasaki (Tokyo Institute of Technology, Japan). The mutants were constructed by transformation of cassettes containing the ORF plus 118 bp upstream and 134 bp downstream sequences (IDT fragments PCR amplified with *sfr1-1* CAACTATTACACTCAACGCG/*sfr1-2* GATTAAGG TACCTCTGTACG primers) into the *sfr1::ura5-lys7* deletion strain (CMC1343) and 5-FOA selection. Gene replacements were confirmed by PCR amplification and sequencing with external primers (*sfr1-C1* TGGAACCACCAGTCATGTGC/*sfr1-C2* CATAAACTCAGCCTG-GAACG). For *sfr1-7D* and *sfr1-7A* design, codons for Thr73, Thr89, Ser109, Ser116, Ser147, Thr152, Ser165 were changed to GAT (Asp) or GCG (Ala), and the strains were followed by PCR amplification (*sfr1-1*/*sfr1-6* TTTGTTACAGCTAACGATGG primers) and Sau3AI (GATC) or Hin6I (GCGC) digestion, respectively. For *sfr1-WI*, Lys93, Lys94, Arg95, Lys157, Arg158, Lys160, Arg161 residues were mutated to Ala (GCG); and the mutant was also followed by PCR amplification (*sfr1-9* GACTCACAGGGTCAGTTAGG/*sfr1-2* primers) and Hin6I digestion. CMC1343 strain was made by complete ORF replacement with the ura5-lys7 cassette amplified from pUL57 plasmid with primers *sfr1-D1(ura5-lys7)* GCGCTTCATTGCATCTT TGTTATAATTTAACATTCACTCTTTCATCTTCGAATTAGTAC TTTAAAACGATTTAACTGAAACTCATTTGGCTTGGTACTGCT G/*sfr1-D2(ura5-lys7)* AGCTAAACATATTTAAGTTCGTACTAAAG GTTTTCTAACTAGTACGAACGAAAATTTTTTTAACAATAGTA AGGAGTCCTATTACAAGTCGTTCAATGTCTCCC.

## Synchronous meiosis and western blots

Diploid *pat1-114 leu1-32* strains were obtained by protoplast fusion and selection for complementation of *ade6-M210* and *ade6-M216* alleles (Sipiczki and Ferenczy, 1977). Synchronous meiosis by thermal inactivation at 32 °C of the *pat1-114* temperature-sensitive allele were done as previously described (Davis et al, 2008). Azygotic *pat1⁻* meiosis differs in some aspects from wild-type zygotic meiosis (Yamamoto and Hiraoka, 2003; Cipak et al, 2012). They were induced at lower than standard temperature (32 °C instead of 34 °C) and used only when high synchrony of the culture was required. During kinetics, $0.8 \times 10^7$ cells (1 ml O.D. 0.8) were fixed with cold 70% ethanol for flow cytometry analysis as described in (Tormos-Perez et al, 2016). A Becton Dickinson FACSCalibur and CellQuest software were used for cell acquisition and data analysis. Chromosome segregations were followed by DAPI staining of ethanol-fixed cells and counting the number of nuclei; 300 cells were scored at each time point. When experimentally required, 20 µM 1-NM-PP1 (Toronto Research Chemicals Inc.) or equal DMSO volume was added to the cultures. In total, $1–1.5 \times 10^8$ cells (10–15 ml O.D. 1) were collected at different time points during the kinetics for protein extract preparation in trichloroacetic acid (TCA) (Foiani et al, 1994); final pellets were resuspended in 200 µl 300 mM Tris base 2% SDS, boiled, cleared by centrifugation, and quantified (BCA, Pierce). Overall, 25–50 µg protein extract were separated by SDS-PAGE and semi-dry transferred to nitrocellulose membrane for western blot analysis. In the case of Sfr1 phosphorylation studies, Phos-tag (20–25 µM final concentration) and $MnCl_2$ (40–50 µM final concentration) were added to the resolving gels, and PVDF membranes were used in semi-dry transfers. Antibodies used for detection were: monoclonal anti-GFP JL-8 (1:3000, Living Colors, Clontech), monoclonal anti-Rad51 (3C10) (1:1000, Invitrogen), monoclonal anti-Cdc13 antibodies (1:1000, 6F11/2, Abcam), monoclonal anti-Cdc2 Y100.4 (1:3000, Santa Cruz), monoclonal anti-tubulin B-5-1-2 (1:5000, Sigma-Aldrich), anti-mouse HRP-conjugated (whole molecule 1:3000, Sigma A4416) and anti-mouse HRP-conjugated (Light Chain Specific, 1:3000, Jackson ImmunoResearch). SuperSignal West DURA or SuperSignal West pico PLUS kits (Pierce) were used for developing. Western quantifications were done with under-saturated film exposures and Fiji software (Schindelin et al, 2012). A line was placed at the upper limit of the faster-mobility EGFP-Sfr1 band at an early time point of the *pat1-114* kinetics and signal above this line considered as reduced-mobility (phosphorylated) species. This signal was divided by the total signal to calculate the phosphorylation ratio.

## Recombination assays

Intergenic (crossovers in *h⁻ leu1-32 × h⁺ his5-303* crosses) and intragenic (gene conversion in *h⁻ ade6-M26 × h⁺ ade6-3049* crosses) recombination assays were performed as described in (Bustamante-Jaramillo et al, 2019). Intersister recombination levels using the VL1 system (Latypov et al, 2010) were assayed in *h⁻ intg::ade6D5′-hphR-ade6D3′ ade6-D20 × h⁺ ade6-D20* crosses. In total, $2 \times 10^4–5 \times 10^4$ viable spores were plated on 9 YE+Guanine plates (Guanine inhibits Adenine uptake and kills Ade⁻ cells, (Cummins and Mitchison, 1967; Pourquie, 1970)). Dilutions of the plating mixes were used to estimate the actual plated viable spores by

plating them on YES media. All Ade⁺ colonies in the YE+Guanine plates were grown as patches in YES plates, and replicated to YESHygro and MM to confirm proper markers (Hygromycin resistance and prototrophy).

## Immunoprecipitation, phosphatase treatment, and kinase assays

Overall, $10^9$ cells (100 ml O.D. 1) were collected at 4 °C from diploid synchronous *pat1-114* meiosis at the appropriate times, washed with 1 ml cold STOP buffer (0.9% NaCl, 1 mM $NaN_3$, 10 mM EDTA, 50 mM NaF), and frozen until processing. Cells for EGFP-Sfr1 immunoprecipitation (IP) were collected at prophase (3 h or 4 h after meiotic induction) and cells for Cdc13 IP were collected at meiosis I (5 h after meiosis induction). Frozen pellets were disrupted in a freezer mill (Spex SamplePrep 6770; 5 cycles 1 min run/1 min cool, 12 CPS) and resuspended in 1 ml cold HB buffer (25 mM MOPS, 60 mM β-Glycerophosphate, 15 mM $MgCl_2$, 1 mM DTT, 5 mM p-Nitrophenyl Phosphate, 15 mM EGTA, 1% Triton X-100, pH 7.2) supplemented with 1 mM Sodium Orthovanadate, 2 mM PMSF and 4× complete EDTA-free protease inhibitor cocktail (Roche). After centrifugation for 15 min (4 °C) at $17,000 \times g$ a sample of total protein extract (input) was saved; the rest (2–3 mg of protein extract quantified by BCA) was immuno-precipitated for 1 h at 4 °C with 25 µl GFP-Trap agarose (Chromotek) or 30 µl (50% v/v in HB buffer) of Protein A Sepharose (CL-4B, GE Healthcare) previously bound to 2 µl monoclonal anti-Cdc13 antibodies (6F11/2, Abcam) (or without antibody for negative control). The IPs were washed 3 times with cold HB buffer, resuspended in SDS-PAGE sample buffer, boiled and stored at −20 °C, or alternatively after the washes they were subjected to phosphatase and kinase assays (GFP-Trap IPs at 3 h). For the phosphatase treatment of EGFP-Sfr1 IPs, two additional final washes with phosphatase buffer (NEbuffer for Protein MetalloPhosphatases) were performed to eliminate any phosphatase inhibitor present in the IP. The IP was halved, 3 µl of λ-phosphatase ($4 \times 10^5$ U/ml) (New England Biolabs) were added to half of the IP (or none to the other half, negative control), and the reactions were carried out at 30 °C for 35 min (45 µl final volume). For kinase assays, the total EGFP-Sfr1 IPs were phosphatase treated, washed twice with HB buffer, and divided in two. One half was mixed with Cdc13 IPs, and the other half with negative control IPs (Protein A Sepharose without anti-Cdc13 antibody). The in vitro phosphorylation reaction was performed at 32 °C for 45 min in HB buffer supplemented with 200 µM ATP (60 µl final reaction volume). Inputs (100 µg) and IPs (25–50% of the initial GFP-traps) were examined by western blot with monoclonal anti-GFP JL-8 antibodies (1:3000, Living Colors, Clontech), rabbit monoclonal mix anti-P(S)CDK Substrate antibodies (1:1000, 9477 Cell Signaling), monoclonal anti-Cdc2 Y100.4 (1:3000, Santa Cruz), anti-mouse HRP-conjugated antibodies (Light Chain Specific, 1:3000, Jackson ImmunoResearch) and anti-rabbit HRP-conjugated antibodies (Light Chain Specific, 1:3000, Jackson ImmunoResearch).

## In vitro expression and pulldown assay

Expression plasmids were ordered to GeneUniversal or GeneCust (GST-Sfr1 and variants in pGEX-4T-1 plasmid, and Rad51 in pET-11d plasmid). After transformation in *Escherichia coli* (BL21 DE3),

recombinant protein expression was induced at O.D. 1 with 0.1 mM IPTG in 2×YT medium supplemented with 100 µg/ml Ampicillin for 2 h at 25 °C. In all, 5–15 ml cultures (15 ml for GST-Sfr1 and variants, and 5 ml for Rad51) were collected by centrifugation and resuspended in 0.5–1 ml 0.5× CelLytic reagent (Sigma-Aldrich) diluted in pulldown buffer (25 mM HEPES-KOH pH 7.5, 150 mM NaCl, 3.5 mM MgCl₂, 0.5 mM DTT, 0.05 mg/ml BSA, 10% glycerol, 0.05% Igepal) supplemented with 0.2 mg/ml lysozyme, 50 U/ml Benzonase (Sigma-Aldrich), 2 mM PMSF and 4× complete EDTA-free protease inhibitor cocktail (Roche), and briefly incubated at room temperature (RT) for 1–2 min to minimize GST-Sfr1 degradation. After centrifugation at 4 °C for 4 min at 17,000× g, cleared lysates were obtained and quantified by BCA. GST-Sfr1 lysates (~2 mg) and Rad51 extract (75 µg) were mixed and incubated with 30 µl Glutathione Sepharose 4B (50% slurry, Cytiva) for 1 h at 4 °C. Pulldowns were washed three times with cold pulldown buffer, resuspended in SDS-PAGE sample buffer, boiled and analyzed by western blot. Even in this condition, degradation was produced and a smaller GST-Sfr1 band containing the N-terminal was always observed in the input extracts. Since the Sfr1 N-terminal domain binds to Rad51, pulldown quantification (Rad51/Sfr1 ratio) was calculated taking into account both bands. High sensitivity to proteolysis at the boundary between the N-terminal disordered and the C-terminal structured domains has been reported (Kuwabara et al, 2012). Antibodies used were: monoclonal anti-GST (8–326) HRP-conjugated (1:3000, Invitrogen), monoclonal anti-Rad51 (3C10) (1:1000, Invitrogen), and anti-mouse HRP-conjugated (Light Chain Specific, 1:3000, Jackson ImmunoResearch).

## Yeast two-hybrid assay

Full-length regions for *sfr1⁺*, *sfr1-7A*, and *sfr1-7D* amplified from respective *S. pombe* strains were cloned into plasmid pGADT7 (Clontech) for GAL4 activation-domain prey constructs using NEBuilder HiFi Assembly kit. Primers used for amplification were 5'-gaggccagt-gaattccacccgATGTCGCAAACAATTAACTC-3' and 5'-tcccgtatcgatgcc-cacccTTACGATTTCCAATCACC-3'. All inserts were verified by sequencing with primer 5'-TAATACGACTCACTATAGGGC-3'. pGBKT7-Rad51 bait construct (GAL4 DNA-binding domain) is described in Polakova et al, 2016. Bait and prey plasmids were introduced into *Saccharomyces cerevisiae* strain PJ69-4a (Clontech) by lithium acetate transformation. Transformants were tested for protein-protein interactions by spotting 10-fold serial dilutions onto SD minimal medium lacking Leu, Trp (SD-L,W) and SD plates lacking Leu, Trp, and His (SD-L,W,H) (with or without different concentrations of 3-amino-1,2,4-triazol) or Leu, Trp, and Ade (SD-L,W,A).

## Time lapse in vivo microscopy

For EGFP-Sfr1 and Rad51-ECFP live imaging experiments, cultures of *h⁺* and *h⁻* cells were prepared and processed as described in (Bustamante-Jaramillo et al, 2021). For meiotic induction in this case, 5 ml O.D. 0.75 of each culture was mixed together prior to the washes with sterile water and transferred to 5 ml MM without nitrogen (total O.D. 1.5) for overnight incubation (14–15 h) at 25 °C with gentle agitation (60 rpm). Time-lapse imaging of zygotes was carried out at 25 °C for 4 h under Olympus IX81 spinning disk microscope equipped with a confocal CSUX1-A1 module

(Yokogawa), a 100×/1.4 Oil Plan APO lens, an Evolve camera (Teledyne Photometrics) and Metamorph software (Molecular Devices LLC). EGFP-Sfr1 images were acquired every 5 min and Rad51-ECFP images every 10 min. In both cases, 9-11 Z sections were collected to cover 4–5 µm of total thickness (0.5 µm step size). Images were processed and analyzed using Fiji software (Schindelin et al, 2012). For both proteins, the intensity (integrated density) and skewness of the nuclear signal were measured along prophase using the sum or maximum Z projection, respectively. For this analysis, the time stack of a selected zygote was processed with the maximum Z projection followed by a median filter (sigma 2); then, proper threshold was applied (Otsu method) to segment the nuclear signal. The resultant binary mask was applied to the corresponding original time stack (Z sum/max projection) to measure the parameters in each zygote. Timepoint 0 was individually set at the end of the *horsetail* movement to compare the asynchronous cells.

## Chromosome spreads preparation and immunostaining

Overall, $5 \times 10^7$ diploid cells (5 ml O.D. 1) were collected at different time points during *pat1-114* synchronous meiosis. Spreads were prepared essentially as described in (Loidl and Lorenz, 2009) and stored at −20 °C until use. Immunostaining was performed as described in (Loidl and Lorenz, 2009). Primary and secondary antibodies were incubated at RT, overnight and 4 h, respectively. Antibodies used were: rabbit IgG fraction anti-GFP (1:800; A11122 Molecular Probes), monoclonal anti-Rad51 (Clone 51RAD01/3C10) (1:50, Invitrogen), anti-mouse Alexa 488 (1:1000; A11001 Molecular Probes) and anti-rabbit Alexa 568 (1:1000; A11011 Molecular Probes). Specificity of the primary antibodies was tested by staining *sfr1⁺ rec12* deleted cells (Appendix Fig. S7). Images were acquired under Nikon Ti2-E spinning disk microscope equipped with a confocal Dragonfly module (ANDOR), a 100×/1.45 Oil Plan APO lens, a sCMOS Sona 4.2B-11 camera (ANDOR) and Fusion 2.2 software (ANDOR). In all, ×1.5 magnification and deconvolution were applied during acquisition. 16 Z sections were collected to cover 3 µm of total thickness (0.2 µm step size). Rad51 and EGFP-Sfr1 foci analysis was performed with Fiji software (Schindelin et al, 2012) using the sum Z projection. Mean intensity and number of foci were measured using Trackmate plug-in with an object diameter of 0.3 µm. A constant threshold among samples was set to count foci with significant intensity, but a weaker signal was always observed and uniformly distributed along chromatin. Colocalization percentage was calculated by counting overlapping Rad51 and EGFP-Sfr1 foci. For further colocalization analysis, Pearson's coefficient was determined using JACoP plug-in, and Costes randomization was applied in all cases to discard spurious colocalization (Costes *P* value = 100 in all cases).

## Whole-cell immunostaining and chromosome segregation assay

Homothallic *h⁹⁰* strains were grown in 50 ml YES until exponential phase (O.D. 0.8), washed three times with water, transferred to PMG-NH₄Cl plates, and incubated at 25 °C for 12–15 h. Immunostaining were performed as described in (Rabitsch et al, 2004). Primary and secondary antibodies were incubated at RT on a wheel, overnight and 6 h respectively. Antibodies used were: polyclonal

anti-Rad51 (63-001) (1:500, Bio Academia), monoclonal anti-tubulin TAT-1 (1:400, (Woods et al, 1989)), anti-mouse Alexa Fluor 568 (1:500, A110031 Invitrogen) and anti-rabbit Alexa Fluor 488 (1:500, A11034 Invitrogen). Samples were observed under Zeiss Imager.Z2 microscope equipped with a 60×/1.4 Oil lens, an Axiocam 506 camera and ZenPro software. Tubulin staining was used to select zygotes in anaphase I and II, and the persistence of Rad51 foci was examined. DNA segregation abnormalities were analyzed by DAPI staining, exploring for lagging or fragmented DNA, unequal segregation, or non-segregation (single DNA mass).

## Statistical analysis

Data were analyzed with Excel and Prism GraphPad. Normality was assessed with the Shapiro–Wilk test. Student's *t* test (unpaired, two tails) was used for statistical analysis in recombination assays, spore viability, pulldown quantifications and in vivo time-lapse microscopy. This test is robust for very small *n*, where the normality of the data is less reliable to evaluate, and it also accepts small deviations from normal distribution. For the statistical analysis of foci in chromosome spreads microscopy, we used nonparametric Mann–Whitney test to compare results at 3 h (two groups) and Kruskal–Wallis test (one-way nonparametric ANOVA) with Dunn's correction for multiple comparison (three groups) at the rest of the time points.

# Data availability

This study includes no data deposited in external repositories.

The source data of this paper are collected in the following database record: biostudies:S-SCDT-10_1038-S44318-024-00205-2.

# Peer review information

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

## Acknowledgements

The authors thank Hiroshi Iwasaki for EGFP-Sfr1 version, Dorota Dziadkowiec for Rad51-ECFP version, Alexander Lorenz for *sfr1-11::hphMX4* deletion, Jürg Kohli for VL1 system, Keith Gull for TAT-1 anti-tubulin antibodies, Carmen Castro for microscopy assistance (IBFG Microscopy Service), Jesús Pinto for image analysis assistance (IBFG Bioinformatic Service), and Sonia Martín for technical support. IP-B is a PhD student funded by the "Ministerio de Universidades" with an FPU19/03456 grant; LG was part of the project personnel in training funded by "Plan Operativo de Empleo Juvenil (Fondo Social Europeo e Iniciativa de Empleo Juvenil)" and Junta de Castilla y León; MB was a student of the master program "Biología Celular y Molecular" at the University of Salamanca. Work in CM-C laboratory was funded by grants from the MCIN/AEI/10.13039/501100011033/ & FEDER "*Una manera de hacer Europa*", grant number FEDER-PGC2018-101908-B-I00; and from Junta de Castilla y León, grant numbers CSI259P20 and CSI010P23 which are co-funded by FEDER program. IBFG was funded by the Program "Escalera de Excelencia" of the Junta de Castilla y León Ref. CLU-2017-03 co-funded by the P.O. FEDER of Castilla y León 14-20; and by the Internationalization Project "CL-EI-2021-08 - IBFG Unit of Excellence" of the CSIC, funded by the Junta de Castilla y León and co-financed by the European Union (ERDF "Europe drives our growth"). Work in the SBP laboratory was funded by the Slovak Research and Development Agency (APVV-21-0210).

## Author contributions

**Inés Palacios-Blanco**: Conceptualization; Formal analysis; Investigation; Visualization; Methodology; Writing—original draft; Writing—review and editing. **Lucía Gómez**: Formal analysis; Investigation; Methodology. **María Bort**: Formal analysis; Investigation; Methodology. **Nina Mayerová**: Formal analysis; Investigation; Visualization; Methodology. **Silvia Bágeľová Poláková**: Supervision; Funding acquisition; Investigation. **Cristina Martín-Castellanos**: Conceptualization; Formal analysis; Supervision; Funding acquisition; Investigation; Visualization; Methodology; Writing—original draft; Project administration; Writing—review and editing.

Source data underlying figure panels in this paper may have individual authorship assigned. Where available, figure panel/source data authorship is listed in the following database record: biostudies:S-SCDT-10_1038-S44318-024-00205-2.

## Disclosure and competing interests statement

The authors declare no competing interests.

# Expanded View Figures

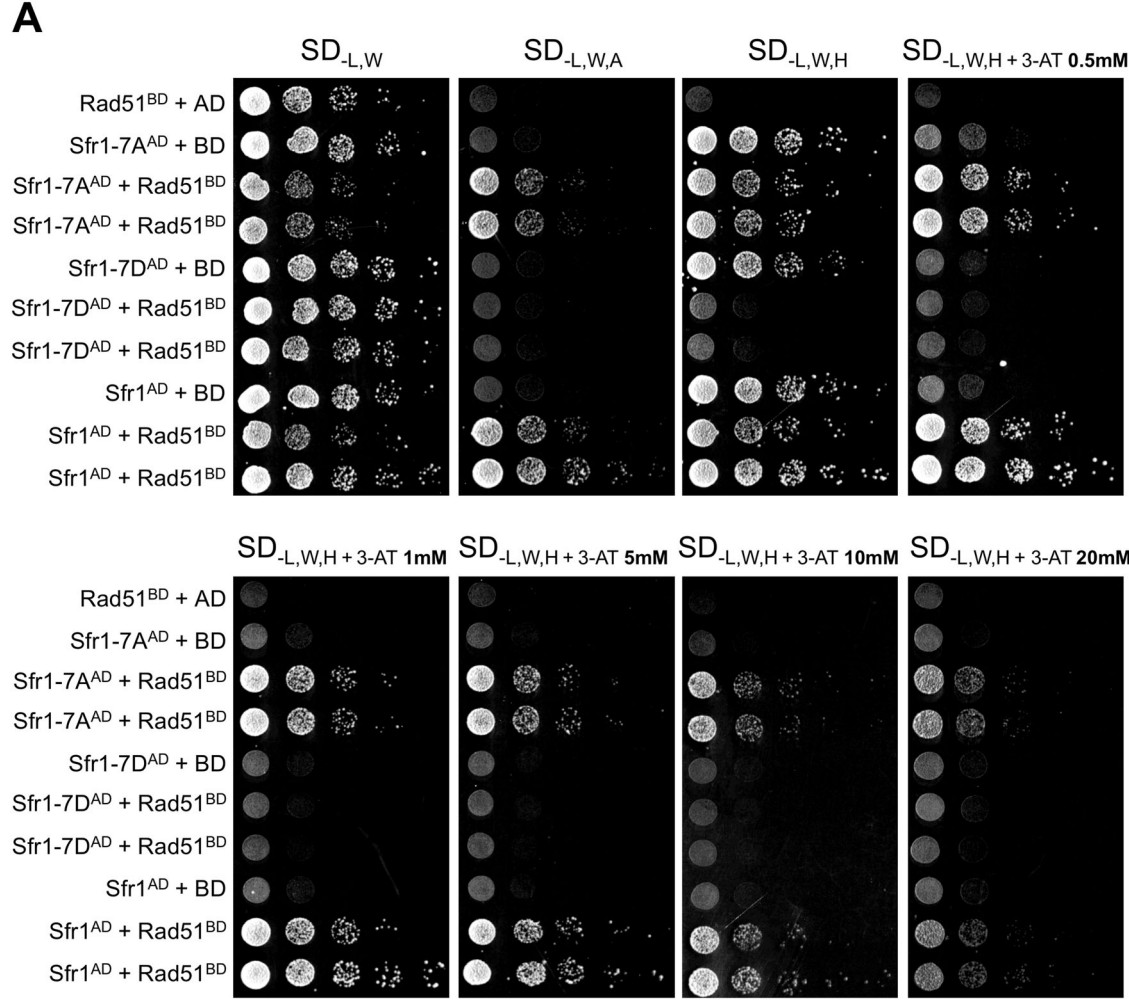

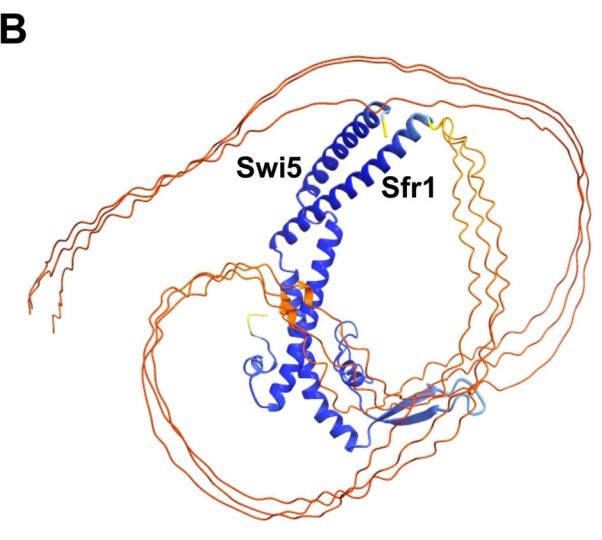

◀ **Figure EV1. Sfr1-7A but not Sfr1-7D interacts with Rad51 in Yeast Two-Hybrid assays.**

(A) Strains expressing Sfr1, Sfr1-7A, and Sfr1-7D fused to the GAL4 transcription-activation domain and Rad51 fused to the GAL4 DNA-binding domain were grown on SD plates lacking tryptophan and leucine (SD-L,W), grown in SD-L,W liquid O/N at 30 °C and then spotted at 10-fold serial dilutions on SD plates lacking leucine and tryptophan (SD-L,W) and SD plates lacking leucine, tryptophan and adenine (SD-L,W,A) or SD plates lacking leucine, tryptophan and histidine (SD-L,W,H) (with or without different concentrations of 3-amino-1,2,4-triazol, 3-AT). The known interaction between Sfr1 and Rad51 was used as a positive control. Growth on plates without histidine or without adenine indicates interaction between the fusion proteins. 3-AT is a competitive inhibitor of *HIS3* gene product which inhibits non-specific interactions and helps to discriminate differences in the interaction efficiency. Two independent transformants are shown for each interaction. Related to Fig. 3A. (B) Swi5–Sfr1 complexes with wild-type, Sfr1-7A and Sfr1-7D proteins modeled by ColabFold are shown (color bfactor palette) (https://colab.research.google.com/github/sokrypton/ColabFold/blob/main/AlphaFold2.ipynb). The large disordered N-terminal part of the proteins are shown in orange.

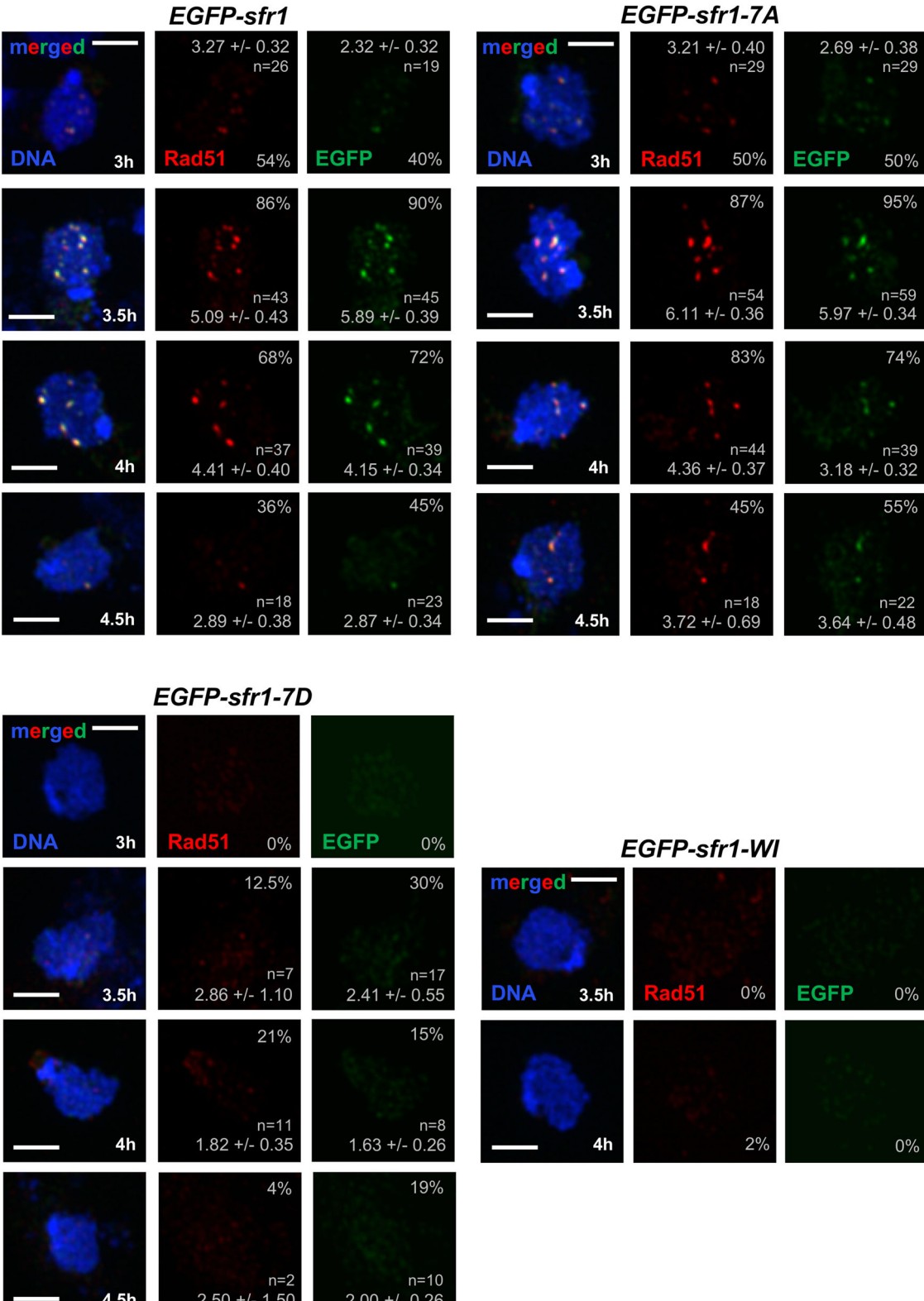

◀ **Figure EV2. Localization of mutant Sfr1 proteins and impact on Rad51 chromosome loading.**

*pat1-114 EGFP-sfr1* (CMC1649), *EGFP-sfr1-7A* (CMC1733), *EGFP-sfr1-7D* (CMC1756) and *EGFP-sfr1-WI* (CMC1769) diploid cells were induced to enter meiosis and collected at different time points during prophase for nuclear spread preparation. Spreads were stained with anti-GFP antibodies for the visualization of EGFP-Sfr1 proteins (in green) and with anti-Rad51 antibodies (in red). DAPI staining to visualize DNA is shown in blue. Representative images of nuclei at each time point after meiotic induction are shown (maximum Z projections). Notice that the images at 3.5 h are also presented as an introductory summary of the experiment in Fig. 4A. Scale bars correspond to 2 μm. The percentages of nuclei showing foci for the corresponding proteins are indicated. The mean of the number of foci per nucleus $+/-$ SEM (in n analyzed nuclei) is also indicated. Related to Fig. 4.

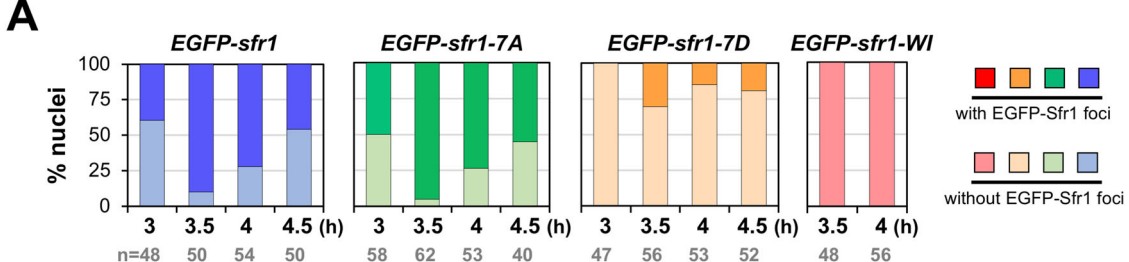

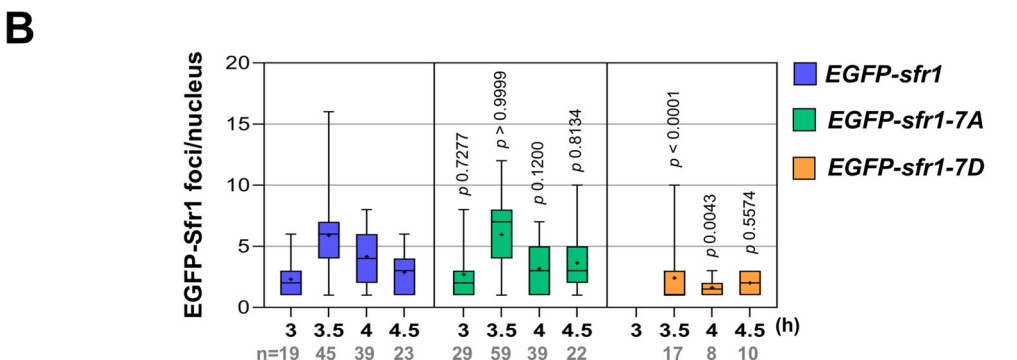

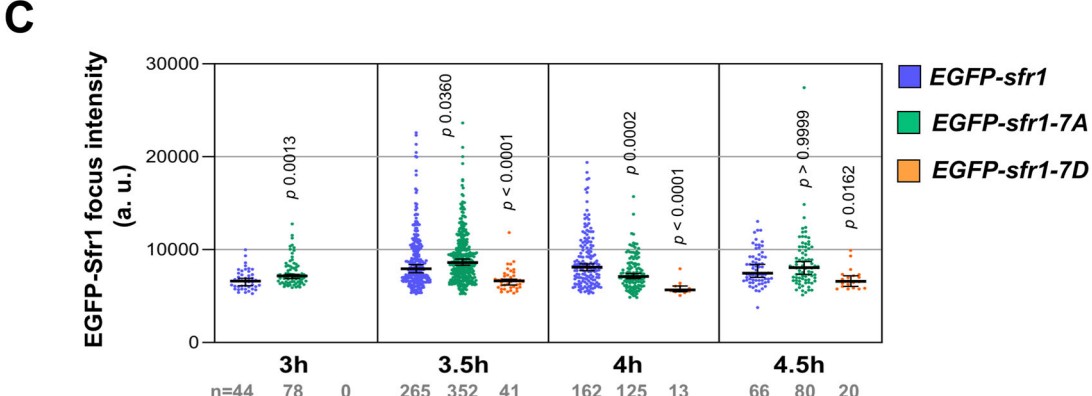

**Figure EV3. Impact of Sfr1 phosphorylation on its chromatin loading.**

*pat1-114 EGFP-sfr1* (CMC1649), *EGFP-sfr1-7A* (CMC1733), *EGFP-sfr1-7D* (CMC1756) and *EGFP-sfr1-WI* (CMC1769) diploid cells were induced to enter meiosis and collected at different time points during prophase for nuclear spread preparation. Spreads were stained with anti-GFP antibodies for the visualization of EGFP-Sfr1 proteins. (**A**) Dynamics of EGFP-Sfr1 (or mutant versions) signals during prophase. The percentage of nuclei with (dark colors) and without (light colors) EGFP-Sfr1 foci is represented. (**B**) Quantification of the number of EGFP-Sfr1 (or mutant versions) foci per nucleus in the nuclei with signal. Data are represented by box-and-whisker plots where boxes extend from the 25th to 75th percentiles, and bars within the boxes represent the medians and black crosses the means; the whiskers represent the minimum and the maximal range. (**C**) Representation of the intensity of individual EGFP-Sfr1 (or mutant versions) foci. The median $+/-$ 95% confidence interval is indicated. For all the graphs in the figure, the number of analyzed nuclei or foci is indicated (*n*). Comparisons were done with the *EGFP-sfr1* control experiment, and *P* values were calculated based on Mann–Whitney test to compare results at 3 h (2 groups) and Kruskal–Wallis test (one-way nonparametric ANOVA) with Dunn's correction at the rest of the time points. Related to Fig. 4.

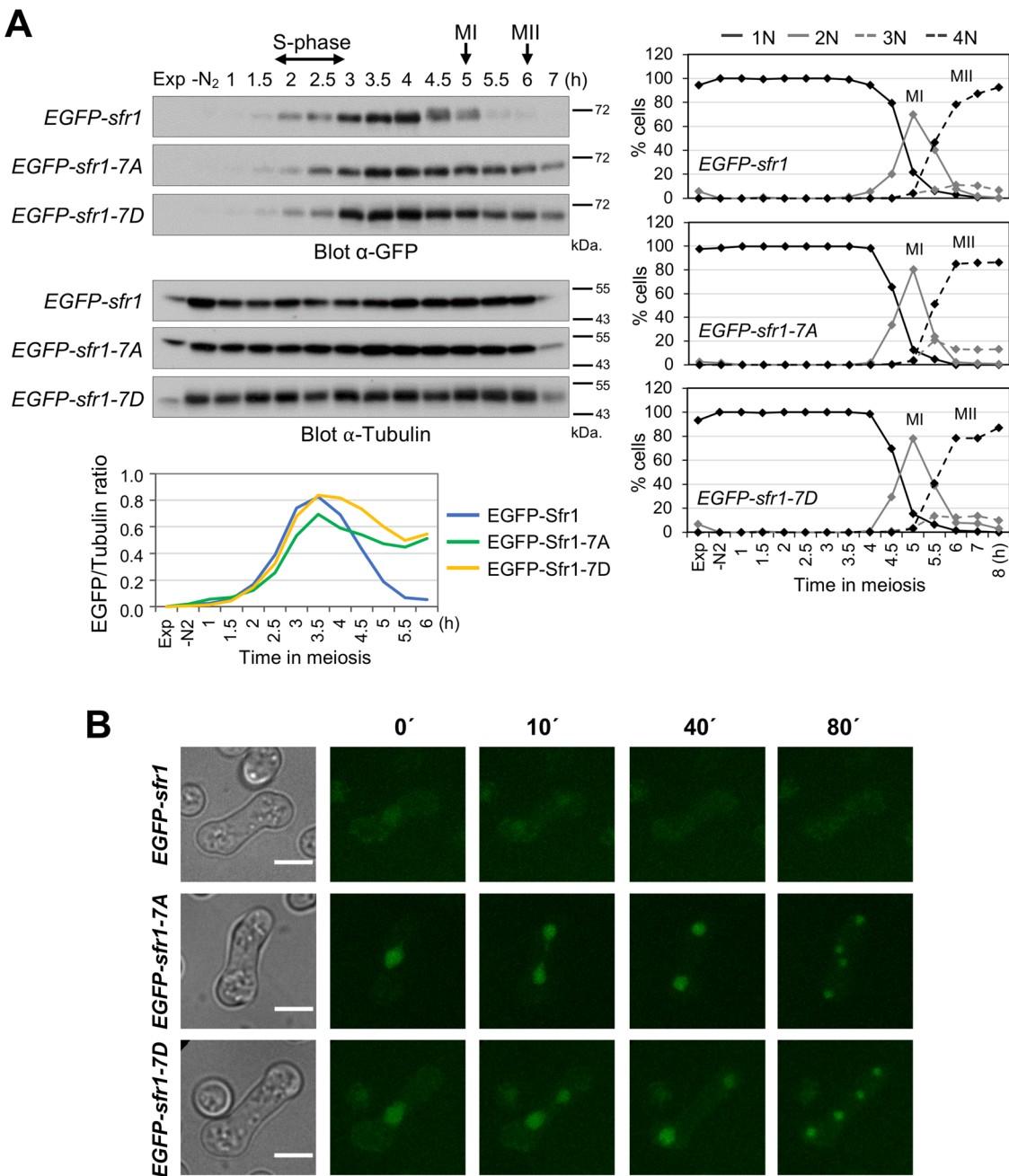

**Figure EV4. Expression of EGFP-Sfr1 and phospho-mutant proteins.**

(**A**) *pat1-114 EGFP-sfr1* (CMC1649), *EGFP-sfr1-7A* (CMC1733) and *EGFP-sfr1-7D* (CMC1756) diploid cells were induced to enter meiosis and collected at the indicated time points. Left, western blot detection of EGFP-Sfr1, EGFP-Sfr1-7A and EGFP-Sfr1-7D proteins (upper blots); tubulin detection was used as loading control (lower blots). Quantification of EGFP levels is shown at the bottom (EGFP/tubulin ratio, mean of two independent kinetics). Right, meiotic progression measured as the number of nuclei per cell; timing of meiosis I (MI) and meiosis II (MII) is indicated. (**B**) Same time lapse experiments as in Fig. 5 showing EGFP-Sfr1-7A and EGFP-Sfr1-7D expression after prophase. In this representation, time point 0 was set as the frame just prior to the first meiotic division (meiosis I) detected by the EGFP signal in the segregating nucleus. Scale bars correspond to 5 μm. Related to Fig. 5.

