## [Peer Review File · The EMBO Journal]

CDK phosphorylation of Sfr1 downregulates Rad51 function in late-meiotic homolog invasions

Inés Palacios-Blanco, Lucía Gómez, María Bort, Nina Mayerova, Silvia Bágelová Poláková, and Cristina Martín-Castellanos

Corresponding author(s): Cristina Martín-Castellanos (cmartin@usal.es)

Review Timeline:

Submission Date:	9th Jan 24
Editorial Decision:	16th Feb 24
Revision Received:	27th Jun 24
Editorial Decision:	31st Jul 24
Revision Received:	5th Aug 24
Accepted:	8th Aug 24

Editor: Hartmut Vodermaier

Transaction Report:

Dr. Cristina Martín-Castellanos
Instituto de Biología Funcional y Genómica (IBFG); CSIC-USAL
Zacarías González, 2
Salamanca, Salamanca 37007
Spain

16th Feb 2024

Re: EMBOJ-2023-116463
CDK phosphorylation of Sfr1 downregulates Rad51 function in late meiotic recombination intermediates

Dear Dr. Martín-Castellanos,

Thank you for submitting your manuscript on meiotic functions of phosphorylated Sfr1 to our journal. It has now been assessed by three expert referees, whose reports are copied below. As you will see, the referees find the work overall interesting and potentially important, but the comments (esp. those of referee 2) also bring up a number of substantive experimental and presentational issues that would need to be dealt with prior to publication. Should you be able to adequately answer these, we would be happy to consider a revised version further for The EMBO Journal. Since it is our policy to consider only a single round of major revision, I would in this case encourage you to contact me with a revision plan and preliminary point-by-point response already during the early stages of your revision work, in order to clarify how key concerns might best be addressed and to discuss which points would be the most relevant ones within the scope of the present study.

I should add that we would also be open to extension of the default three-months revision period if needed; with our 'scooping protection' (meaning that competing work appearing elsewhere in the meantime will not affect our considerations of your study) of course remaining valid also throughout such an extended revision.

Further information on preparing, formatting and uploading a revised manuscript can be found below and in our Guide to Authors. Thank you again for the opportunity to consider this work for The EMBO Journal, and I look forward to hearing from you in due time.

Yours sincerely,

Hartmut Vodermaier

9) Digital image enhancement is acceptable practice, as long as it accurately represents the original data and conforms to community standards. If a figure has been subjected to significant electronic manipulation, this must be clearly noted in the figure legend and/or the 'Materials and Methods' section. The editors reserve the right to request original versions of figures and the original images that were used to assemble the figure. Finally, we generally encourage uploading of numerical as well as gel/blot image source data; for details see: embopress.org/page/journal/14602075/authorguide#sourcedata

At EMBO Press, we ask authors to provide source data for the main manuscript figures. Our source data coordinator will contact you to discuss which figure panels we would need source data for and will also provide you with helpful tips on how to upload and organize the files.

In the interest of ensuring the conceptual advance provided by the work, we recommend submitting a revision within 3 months (16th May 2024). Please discuss the revision progress ahead of this time with the editor if you require more time to complete the revisions. Use the link below to submit your revision:

Link Not Available

Referee #1:

Phosphorylation of Sfr1 during meiosis has already been described but the role of this phosphorylation was not known. Now Palacios-Blanco et al. show that phosphorylation of Sfr1 by Cdc2 downregulates Rad51 function during meiosis. This finding is novel, important for the fields of meiosis and DNA repair and it is reasonably well supported by experimental data. I have the following suggestions how to improve the manuscript.

- major comment:

1) In order to get a better picture of how important is Sfr1 phosphorylation for meiosis, the authors should show spore viability of sfr1-delta, sfr1-7A and sfr1-7D mutants compared to wild-type. The authors mention in their manuscript that spore viability of sfr1-delta is only slightly reduced but, in my opinion, this would be a useful addition.

- minor comments:

1) Although the observed synthetic phenotype is very interesting, it is not clear why the authors decided to combine sfr1-7A mutant with dbl2 deletion. Few sentences explaining why dbl2 was chosen should be added.

2) It would be useful to show spore viability for single and double mutants described in Figure 7A.

3) Do 7A and 7D mutations affect Sfr1 protein levels? Protein levels of Sfr1, Sfr1-7A and Sfr1-7D during meiosis should be shown on one Western blot.

4) When indicating number of repeats, please write "n=3" instead of "n 3".

5) Molecular marker (kDa) is not indicated on all Western blots. Please add them.

6) Page 6, line 194; I suggest to change "... different massive proteomic approaches ..." to "... different proteomic approaches

...".

7) This manuscript is focused on Sfr1 phosphorylation but Swi5 is also phosphorylated. Are there predicted CDK phosphorylation sites on Swi5? This should be mentioned in the discussion.

8) It should be mentioned that pat1-114 induced meiosis used in this manuscript differs from wild-type meiosis (e.g., PMID: 12727894, PMID: 22487684).

9) I suggest not to use the term "cross-spindles" (Figure 7C). I prefer to call these: mononucleate zygotes containing two spindles.

Referee #2:

In this study, Palacios-Blanco et al investigate the role of CDK in controlling Swi5-Sfr1, an auxiliary factor in recombinase-mediated homolog invasion during recombination. The authors found that Sfr1 is phosphorylated by CDK, which downregulates its function during meiotic prophase I. A mutant mimicking this phosphorylation inhibits Rad51 binding and interhomolog recombination. Conversely, a non-phosphorylatable mutant exacerbates chromosome segregation issues in the presence of other mutations (dbl2), leading to unsegregated chromatin and Rad51 retention. The N-terminal disordered region of Sfr1, conserved across evolution, plays a role in regulating this phosphorylation. Overall, the authors propose that this newly discovered cell-cycle dependent mechanism ensures timely resolution of recombination intermediates, facilitating successful chromosome disjunction during gamete formation.

The work is overall interesting and merits publication. However, it would benefit from considerable revisions to address the points below.

i. Text/narrative and Figure organisation: the text and figures should be majorly revised.

1. The text is unnecessarily complicated and would benefit from being streamlined. All parts are at least 30% longer than would be needed to say everything that is relevant. This makes the reading tedious and the key messages difficult to grasp. For example, the introduction provides a lot of background, but there is no clear structure and the introduction should lead to the description of the hypothesis (why was Sfr1 studied?). Instead, the entire first page of the Results section (179-202) explains the gap in knowledge about the potential functions of CDK other than DSB formation and how this might be related to the regulation of strand invasion. This should be in the introduction. The authors mention in the first page of the results that 'data generated during the course of the (previous) work' prompted the initial experiments. Which data? how did they choose to work on Sfr1? The storyline could be easily and significantly improved.

2. The narrative is confusing with the authors jumping between figures all the time. e.g. the authors jump from Figure 3A to Figure 4A, without all the other data in Figure 3. See lines 285-291. After Figure 6 they jump back to Figure 4D (line 462). There are several other instances where this happens. This is in part because the way data is put into figures appears to be following the type of experiment (or chronology), rather than the scientific points made with it.

3. Some of the scientific considerations are not presented in an easy to understand way for the reader. For example: line 26-27 "This is crucial before the first meiotic division, hence, the non-phosphorylatable sfr1-7A mutant aggravates the segregation problems of the dbl2 mutant, displaying extensive unsegregated chromatin and Rad51 retention." - What is the dbl2 mutant? It comes as if everyone knows this.

4. The discussion is extremely long and contains a lot of speculations. Why speculate so much about the stability of the phosphomutants? Is this phenotype relevant to anything? In addition, new results/figures (Figures S7 and S8) are described here when they should be presented in the results.

5. The authors often use expressions that are not scientific (e.g. line 270 "interestingly enough,...")

6. From the 7 full figures, there are only 3-4 'key' points that can be made. Data could be re-arranged in a more logical way to guide the narrative. Figures 5 and 6 are almost entirely redundant with the chromosome spreads in Figure 3B. There is new information, but none of it needs to feature in a main figure. Figures 1 and 2 could also be easily combined.

7. I find the description of the recombination data and the explanation of the assays insufficient. The legends are complete (perhaps even too long), but from the text it is difficult to grasp what was measured. Why are there graphs and tables for each panel? What should be looked at in the graph vs the table?

8. The abstract could be improved a lot and should be rewritten using clearer and more precise language.

9. The title of the manuscript could be improved. "Late meiotic recombination intermediates" could be considered as double Holliday junctions, for which the authors' results do not show that Rad51 is somehow required.

ii. Data quality/data interpretation

10. The authors propose that CDK-mediated phosphorylation of Sfr1 reduces its ability to function with Rad51 to regulate recombination. As such, the key mutant to be analysed is the *sfr1-7A*. The 7D is interesting in that it can help understand what phosphorylation does, but the 7A is what should be used to understand the biological relevance of Sfr1 phosphorylation. Unfortunately, one has to wait until Figure 7 to finally understand that the phosphorylation of Sfr1 is needed for something (in a *dbl2* mutant background) that is most likely linked to recombination. This data is interesting, but under-explored. What exactly fails in the 7A mutant? Why is the phenotype only seen in a *dbl2* mutant background? The only phenotype of the 7A mutant in recombination is that Rad51 foci seem brighter. Does CDK inhibition (post DSB formation) also cause that?
11. The first paragraph of the Results section (179-202) discusses the five putative CDK sites previously described. Why do the authors mutate seven putative CDK sites? How were the additional two identified?
12. In Figure 1D, the shift in electrophoretic mobility is there from $t=2.5$ hours. There is then a second shift around 4-4.5 hours. Both are eliminated in the 7A mutant. Can the authors blot for cyclin in the same extracts? Does cyclin accumulation explain the shifts? Both, or only the second? Is the second shift also direct phosphorylation by CDK? Could it be PLK? Which of the two phosphorylation events is linked to the phenotypes described?
13. There is always a pool of unphosphorylated Sfr1. How do the authors incorporate this in the model? wouldn't this pool of Sfr1 interfere with the processing of recombination intermediates?
14. Can the authors IP Sfr1 from cells and show that Rad51 binds the phosphorylated form, but not the unmodified form? Alternatively, does in vitro phosphorylation of Sfr1 by Cdc2 (e.g. as in Figure 2C) reduce/abolish the interaction with Rad51?
15. In Figure 1D, it is already clear that the stability of the 7A mutant is higher, in particular at later time points in meiosis. This is not mentioned in the description of the data.
16. Figure 3B and C - Where are the meiotic progressions, Western blots, etc. of these mutants? The protein levels of *sfr1-7A* and *sfr1-7D* are only shown in Figure 6B, but should be included here. Moreover, FACS analyses of DNA content would help assess if the kinetics of meiotic progression in the mutants are similar to the WT.
17. Figure 3B and D - Why are there so few Rad51 foci at these specific times in meiosis when DSB formation should be highest? Would one not expect to see more Rad51 foci? How was the specificity of the Rad51 and Sfr1 signals tested? The authors could stain nuclear spreads for EGFP-Sfr1 and Rad51 in a *rec12* mutant in which no DSBs are formed.
18. Figure 4A - What explains the slight reduction in intergenic recombination in the *sfr1-7A* mutant? Shouldn't interhomolog recombination be increased since Sfr1 is proposed to be a target of CDK-dependent phosphorylation? Is intragenic recombination altered in the *sfr1-7A* mutant?
19. Figure 4B and C - Data for the *sfr1Δ* and *sfr1-7A* mutants are missing as an important comparison.
20. In Figures 6/7, the authors show that the 7A mutant is more stable. What if this is the reason for the anaphase phenotype observed? Can the authors exclude that the increased stability leads to some aberrant function that interferes with chromosome segregation?
21. At the very end of the paper the authors analyze the Sfr1-WI phenotype. Why at the end? I can't follow the logic.
22. Have the authors looked at spore viability in the mutants generated? This would normally be one of the first things to test.
23. How does the phenotype of 7A compare to the 13A phenotype described in the 13A in the Sevcovicova 2021 paper?
24. Figures 5-6 are redundant with the spreads in Figure 3B
25. Figure 4D: does cyclin or *cdc2* o/E change the proportion of Sfr1 phosphorylation? I am not convinced that the assay shows what the authors claim. If the *sfr1-7D* mutant shows a phenotype in the same assay as used in Figure S5, why wouldn't the cyclin/Cdc2 o/E have a phenotype? Could it be that the phosphomimetic *sfr1* mutant is only partially functional, as proposed in Sevcovicova et al. 2021?
26. Videos 1-4 and 4-8 - There is no indication of the time. For the sake of clarity and better comparability with the figures, the time should be included.
27. Figure S4A - No granular/accumulated signal can be distinguished in either wild type or mutants. To clarify whether the putative accumulated signals are indeed Rad51 foci, the analysis should also be performed in *rec12Δ* mutants. If ECFP-Rad51 does not indeed form a detectable granular/accumulated signal, potentially representing foci, this would also help to explain why

the putative Rad51 foci are detected in the time-lapse movies but not on the nuclear spreads (discussed in 442-445).

28. Lines 416 - 417 "Importantly, EGFP-Sfr1-7A and EGFP-Sfr1-7D proteins were similarly expressed during meiotic prophase analyzed by signal intensity in time lapse experiments as well as by western blot in pat1-114 synchronous meiosis (Figure 6A and B, and Table S1)." Analysis of the protein levels of either Sfr-7A or Sfr-7D showed that they are expressed or stabilised longer than the wild type, well beyond the putative M1 and MII event. Could there be a defect in meiotic progression in these mutants? These data should be combined with a comparison of the kinetics of meiotic progression, such as FACS or MI and MII analysis. In addition, analysis of Rad51 protein levels showed normal kinetics of Rad51 levels during meiosis (Figure S4C).

29. Figure 7A - The immunofluorescence images lack any description of the phenotypes observed and the time of sampling. Furthermore, do all sfr1-7A and sfr1-7D mutants behave in the same way and enter MI and II with similar kinetics?

30. Figure 7B - Where do the Rad51 foci predominantly localize to?

31. 536-538 - "This result was surprising given that Sfr1-WI protein retains some ability to bind to Rad51 when compared to Sfr1-7D protein (Figure 3A) (Argunhan et al., 2020)". The interaction between Sfr1-WI and Rad51 seems to be severely affected and could cause the observed recombination defects.

32. Figure 8 - The predicted structure by AlphaFold, and the structure published in Kuwabara et al. 2012, are presented without any description or indication of specific domains or the disordered regions. As such, they do not add anything significant in the context of this study and could be omitted.

Referee #3:

Palacios-Blanco et al.

This manuscript will be of interest for researchers studying homologous recombination, especially during meiosis, with an interest in molecular mechanisms and how meiotic recombination is regulated. The authors report the function of phosphorylation sites in the N-terminus of the recombination mediator protein Sfr1 in the fission yeast *Schizosaccharomyces pombe*. Sfr1 forms a complex with Swi5 which functions as a cofactor for the central recombination proteins Rad51 and Dmc1. The manuscript describes the identification of 7 CDK phosphorylation sites that negatively regulate the interaction of Sfr1-Swi5 with Rad51 and lead to disruption of Rad51 function in late meiotic prophase before meiosis I chromosome segregation. The authors convincingly show that Sfr1 is phosphorylated during meiotic prophase by CDK and that the identified sites are involved. Sfr1 phosphorylation disrupts interaction with Rad51, is critical for removal of Rad51 from meiotic chromosomes at the end of prophase, and required for accurate chromosome segregation during meiosis. The authors convincingly identified a novel regulatory mechanism restricting recombination to meiotic prophase.

Comments:

1) It is not clear how the individual phosphorylation sites were identified, what the evidence is that they are phosphorylated and what their relationship is to the sites studied by Liang P, Lister K, Yates L, Argunhan B, Zhang X (2023 Phosphoregulation of DNA repair via the Rad51 auxiliary factor Swi5-Sfr1. *J Biol Chem* 299: 104929). This requires more information and discussion.

2) It is not clear, if all 7 sites control the function or a subset, as only the 7A/D mutants are analyzed. I am not suggesting conducting more experiments on subsets of sites. Maybe some data are available. Some discussion would be helpful.

3) Figure 1B: What is the n?

4) Figure 2b, C: What is the input amount. Define C- in part C. What is the negative control?

5) Line 284-6, Figure 3A: The statement needs quantitation of the interaction defects.

6) Figure 4: The recombination data are good, but B and C lack the sfr1 deletion and 7A for comparison? Are these data available or published for the sfr1 deletion? I do not think that these data will add much to the papers. Some discussion would be helpful.

7) Figure 4C. I am not sure what the point of this experiment is and what it adds to the manuscript. There are no sfr1 data. I suggest eliminating this part or relegating this to the supplement.

8) What is the effect of sfr1-7A and -7D on spore viability? These data should be added.

9) Figure 5/6: Why is the quantitation of Figure 5 presented in Figure 6?

10) Figure 6B requires quantitation.

Additional points:

-) Line 81: established not stablished
-) L145: SSE not SEE
-) Line 427: though not thought
-) Genotype labeling is inconsistent with regards to capitalization and use of italcs.
-) Figure 3A, 4A-C, 7A-C: Define the controls and provide detail what they are.

Response to reviewers' comments

We thank the reviews for the overall positive view of our work and useful comments to improve it.

Our response point by point to their comments is highlighted in red (including text and figure editing).

The new data and new experimental additions are highlighted in green.

Figures have been reorganized.

Previous Figure	New Figure
Figure 1	Figure 1 (order of panels is reorganized)
Figure 2	Figure 2
Figure 3	Figure 4 (except panel 3A)
Figure 4	Figure 3 (included previous panel 3A and new data)
Figure 5	Figure 5 (included previous panel 6A)
Figure 6	Figure EV4 (except panel 6A)
Figure 4D and S5	Figure 6
Figure 7	Figure 7 (new panel D)
Figure 8	Figure 8
Figure S1	Appendix Figure S1
	Appendix Figure S2 (new)
	Figure EV1 (new)
Figure S2	Figure EV2
Figure S3	Figure EV3
Figure S4	Appendix Figure S3
Figure S5	Figure 6B
Figure S6	Appendix Figure S4
Figure S7	Appendix Figure S5
Figure S8	Appendix Figure S6
	Appendix Figure S7 (new)

Referee #1:

Phosphorylation of Sfr1 during meiosis has already been described but the role of this phosphorylation was not known. Now Palacios-Blanco et al. show that phosphorylation of Sfr1 by Cdc2 downregulates Rad51 function during meiosis. This finding is novel, important for the fields of meiosis and DNA repair and it is reasonably well supported by experimental data.

I have the following suggestions how to improve the manuscript.

- major comment:

1) In order to get a better picture of how important is Sfr1 phosphorylation for meiosis, the authors should show spore viability of sfr1-delta, sfr1-7A and sfr1-7D mutants compared to wild-type. The authors mention in their manuscript that spore viability of sfr1-delta is only slightly reduced but, in my opinion, this would be a useful addition.

We have added this information to the Result section and Figure 7. We have quantified spore viability in wild-type, *sfr1*-delta, *sfr1*-7A and *sfr1*-7D mutants, and double mutants in combination with *dbl2*-delta (Figure 7D).

A new paragraph describing these experiments has been introduced in the main text and the corresponding figure legend (highlighted in yellow in the word document).

This is a shared comment with referee 2 and 3.

- minor comments:

1) Although the observed synthetic phenotype is very interesting, it is not clear why the authors decided to combine *sfr1*-7A mutant with *dbl2* deletion. Few sentences explaining why *dbl2* was chosen should be added.

Dbl2 role in Rad51 chromosomal loading is mentioned in the Introduction. We have clarified (refreshed) this point when the experiment is described in the Results section (highlighted in yellow in the word document).

“Given the defect of the *sfr1*-7A mutant in Rad51 dynamics at the end of prophase (Figure 4D), we decided to address segregation defects in combination with the *dbl2* deletion mutant, which represents a compromised situation with Rad51 retention and segregation defects at meiosis I (Polakova *et al*, 2016).”

2) It would be useful to show spore viability for single and double mutants described in Figure 7A.

This is related to point 1 (major comment).

3) Do 7A and 7D mutations affect Sfr1 protein levels? Protein levels of Sfr1, Sfr1-7A and Sfr1-7D during meiosis should be shown on one Western blot.

This was already shown in Figure 6 by signal intensity quantification (IntDen, a.u.) in time lapse experiments (Fig. 6A) and by western blot analysis in synchronous *pat1-114* meiosis (Fig. 6B). Time lapse experiments allow to address *in vivo* nuclear levels from early prophase to chromosome segregation. Synchronous *pat1-114* meiosis allows to address total protein levels by western blot.

This Figure has been arranged and Fig. 6A is now Fig. 5B, and Fig. 6B is now Figure EV4A.

4) When indicating number of repeats, please write "n=3" instead of "n 3".

We have corrected this in figures and main text.

5) Molecular marker (kDa) is not indicated on all Western blots. Please add them.

Molecular marker does not run properly in Phos-tag gels. They contain EDTA which interferes with components of the gels (divalent cations, Mn²⁺) and affects the running of samples close to them (up to several wells). Molecular markers were not loaded in this type of gels.

We think these are the only gels where molecular markers were not indicated. In addition, in figure panels where several blots were presented, molecular markers were indicated for simplicity only in the top blot of the series (Fig. 6B (now 5B), S4C (now S3C) and S6C (now S4C)). We have now added the markers to all the individual blots.

6) Page 6, line 194; I suggest to change "... different massive proteomic approaches ..." to: "... different proteomic approaches ...".

We have removed the word "massive" as suggested.

7) This manuscript is focused on Sfr1 phosphorylation but Swi5 is also phosphorylated. Are there predicted CDK phosphorylation sites on Swi5? This should be mentioned in the discussion.

We have added a paragraph to the Discussion section where this is discussed (highlighted in yellow in the word document).

"Finally, CDK phospho-regulation of the Swi5-Sfr1 complex is likely exerted exclusively through Sfr1, since only two phosphorylated sites have been reported in Swi5 (Ser72 and Ser84; Cipak et al, 2009; Sevcovicova et al, 2021) and these are not CDK sites. It is currently unknown the function of this phosphorylation. Furthermore, Swi5 protein does not harbor putative CDK sites."

8) It should be mentioned that *pat1-114* induced meiosis used in this manuscript differs from wild-type meiosis (e.g., PMID: 12727894, PMID: 22487684).

We have added this information in the Material and Methods section, where the used of *pat1-114* synchronous meiosis is introduced (highlighted in yellow in the word document). Notice that we used this type of meiosis only when high synchrony was required (protein level analysis by western blots and nuclear spreading) and they are regularly used in the field.

"Azygotic *pat1* meiosis differs in some aspects from wild-type zygotic meiosis (Yamamoto & Hiraoka, 2003; Cipak et al, 2012). They were induced at lower than standard temperature (32°C instead of 34°C) and used only when high synchrony of the culture was required."

9) I suggest not to use the term "cross-spindles" (Figure 7C). I prefer to call these: mononucleate zygotes containing two spindles.

As suggested, we have changed this in the main text, Figure 7 and figure legend (highlighted in yellow in the word document).

Referee #2:

In this study, Palacios-Blanco et al investigate the role of CDK in controlling Swi5-Sfr1, an auxiliary factor in recombinase-mediated homolog invasion during recombination. The authors found that Sfr1 is phosphorylated by CDK, which downregulates its function during meiotic prophase I. A mutant mimicking this phosphorylation inhibits Rad51 binding and interhomolog recombination. Conversely, a non-phosphorylatable mutant exacerbates chromosome segregation issues in the presence of other mutations (*dbl2*), leading to unsegregated chromatin and Rad51 retention. The N-terminal disordered region of Sfr1, conserved across evolution, plays a role in regulating this phosphorylation. Overall, the authors propose that this newly discovered cell-cycle dependent mechanism ensures timely resolution of recombination intermediates, facilitating successful chromosome disjunction during gamete formation.

The work is overall interesting and merits publication. However, it would benefit from considerable revisions to address the points below.

i. Text/narrative and Figure organisation: the text and figures should be majorly revised.

1. The text is unnecessarily complicated and would benefit from being streamlined. All parts are at least 30% longer than would be needed to say everything that is relevant. This makes the reading tedious and the key messages difficult to grasp. For example, the introduction provides a lot of background, but there is no clear structure and the introduction should lead to the description of the hypothesis (why was Sfr1 studied?). Instead, the entire first page of the Results section (179-202) explains the gap in knowledge about the potential functions of CDK other than DSB formation and how this might be related to the regulation of strand invasion. This should be in the introduction. The authors mention in the first page of the results that 'data generated during the course of the (previous) work' prompted the initial experiments. Which data? how did they choose to work on Sfr1? The storyline could be easily and significantly improved.

We have tried to accommodate this reviewer concerns in the new manuscript version but we also would like to call the author autonomy to decide how to present our work (obviously not affecting scientific quality and comprehension of the manuscript).

We have simplified the first paragraph of the Introduction, eliminated almost the entire fourth paragraph, removed few sentences, and rephrased to better link some paragraphs. We think it was well structured (first we introduce meiosis, then recombination, then the Swi5-Sfr1 complex, then the necessity to repair recombination intermediates on time, and finally the summary of the work in the context of the previous information).

We have also removed some sentences from the rest of the text and rephrased some paragraphs to shorten them. Particularly the last part of the Discussion related to the function of the N-terminal part of Sfr1 has been reduced. In addition, we have added new subheadings to better follow the manuscript.

In the first paragraph of the Results section, the starting point, hypothesis and Sfr1 selection was all stated. We have rephrased this to make clearer that the “data generated during the course of the work” are included in the publication Bustamante-Jaramillo et al 2019 (highlighted in yellow in the word document).

“In addition, based on the comparison of DSB and CO levels, we proposed CDK might have an additional function, acting downstream of break formation in CO inhibition (Bustamante-Jaramillo *et al*, 2019).”

However, we would like to keep the first paragraph of the Results section in place. We think it is too detailed to be in the Introduction.

2. The narrative is confusing with the authors jumping between figures all the time. e.g. the authors jump from Figure 3A to Figure 4A, without all the other data in Figure 3. See lines 285-291. After Figure 6 they jump back to Figure 4D (line 462). There are several other instances where this happens. This is in part because the way data is put into figures appears to be following the type of experiment (or chronology), rather than the scientific points made with it.

The manuscript was not written in chronology order nor by the type of experiments.

The figures were organized by topics (reflected in the figure titles). We thought this type of organization would allow figure economy and a clearer data presentation.

As suggested by the reviewer, we have organized the Figures to keep the text and figure temporal lines as matched as possible.

3. Some of the scientific considerations are not presented in an easy to understand way for the reader. For example: line 26-27 "This is crucial before the first meiotic division, hence, the non-phosphorylatable sfr1-7A mutant aggravates the segregation problems of the dbl2 mutant, displaying extensive unsegregated chromatin and Rad51 retention." - What is the dbl2 mutant? It comes as if everyone knows this.

This sentence is in the abstract. We have re-phrased the abstract (highlighted in yellow in the word document).

This is related to point 8.

4. The discussion is extremely long and contains a lot of speculations. Why speculate so much about the stability of the phosphomutants? Is this phenotype relevant to anything? In addition, new results/figures (Figures S7 and S8) are described here when they should be presented in the results.

As mentioned above (point 1), we have removed some sentences from the text and rephrased some paragraphs to shorten them. Particularly the last part of the Discussion related to the function of the N-terminal part of Sfr1 has been reduced.

We consider important to describe and speculate about the new functions of the N-terminal part of Sfr1 and the conservation of the putative sequence determinants. Since this is not the main part of our results but interesting unpredicted observations that expands our knowledge of Sfr1 regulation, we added Figure S7 and S8 as supplemental figures to show conservation, and we would like to keep them like this. Please, notice these Figures are now Appendix Figure S5 and Figures S6.

We do not know the function of Sfr1 protein downregulation. Please, notice that in the first version of the manuscript we already mentioned in the Discussion:

“Regulation of Sfr1 protein stability at the end of meiotic prophase might represent an additional mechanism to inhibit late IH invasions”.

5. The authors often use expressions that are not scientific (e.g. line 270 "interestingly enough,...")

We have removed this unfortunate expression, and “interestingly, remarkably or notably” from the text.

6. From the 7 full figures, there are only 3-4 'key' points that can be made. Data could be re-arranged in a more logical way to guide the narrative. Figures 5 and 6 are almost entirely redundant with the chromosome spreads in Figure 3B. There is new information, but none of it needs to feature in a main figure. Figures 1 and 2 could also be easily combined.

As mentioned above (point 2), Figures were organized by topics.

As suggested by the reviewer, in the revised version we have organized the Figures to keep the text and figure temporal lines as matched as possible.

We considered joining Figures 1 and 2, but the inclusion of the FACS representation, nuclear counting, and a significant number of blots resulted into a crowded figure. We consider important to show the FACS and nuclear counting data in these experiments since they are the first shown in the manuscript. In our experience, they are requested when not included. In addition, we have added new blots to Figure 1 (requested by the reviewer in point 12).

Figure 5 is focused on the “*In vivo* localization of EGFP-Sfr1 and phospho-mutant proteins”. It describes the accumulation of EGFP-Sfr1 in aggregates that depends on DSB formation, and the pattern of the phospho-mutant proteins. This is the first report of Sfr1 meiotic localization *in vivo*. We did not want to put it as a supplementary figure.

Figure 6 is focused on the “Expression of EGFP-Sfr1 and phospho-mutant proteins”. It describes the protein levels of EGFP-Sfr1 and phospho-mutant.

Initially we thought of putting all these data in a single figure but we wanted to have good size images to appreciate the microscopy data (39 single images). Reducing all the panels to half size to accommodate all of them in a single figure would make the figure too crowded.

These figures (5 and 6) are not redundant with Figure 3B (now Figure 4A). They are complementary. Notice that Figure 4 focused on “*Sfr1 phosphorylation impairs Rad51 recombinase loading onto chromatin*”. Panel 4A shows Rad51 (and Sfr1) chromatin binding during **prophase** using spreads of chromosomes.

As mention in the text, Figure 5 and 6 complement this (only for Sfr1) using time lapse experiments with a **much wider meiotic temporal framework**, and in **live cells** to study protein localization.

Notice that these Figures are reorganized in the new version. Figure 6A is now in Figure 5B. The rest of Figure 6 is now Figure EV4.

7. I find the description of the recombination data and the explanation of the assays insufficient. The legends are complete (perhaps even too long), but from

the text it is difficult to grasp what was measured. Why are there graphs and tables for each panel? What should be looked at in the graph vs the table?

We have rephrased the introduction to the recombination assays (highlighted in yellow in the word document):

“We analyzed the IH invasion efficiency of the *sfr1-7A* and *sfr1-7D* alleles addressing both intergenic and intragenic recombination (see schemes of the assays in Figure 3B-C).”

These recombination assays (now in Figure 3B and 3C) are a measurement of homologous invasion during recombination since recombinants are not recovered when invasion is impaired.

Please, notice that original panel C has been removed as suggested by reviewer 3.

Regarding data presentation, tables show the actual data and graphs the percentage relative to control crosses. The actual data give an idea of the number of recombinants to analyze in order to have solid results. If the actual frequency of recombination is very small a higher number of recombinants needs to be analyzed (a higher number of spores from the crosses needs to be analyzed). This, and the actual analyzed recombinants stated in the figure legend, tell us about the data quality.

In addition, the actual data in the table is useful for other researches.

We would like to maintain both.

8. The abstract could be improved a lot and should be rewritten using clearer and more precise language.

We have re-phrased the abstract (highlighted in yellow in the word document). This is related to point 3.

9. The title of the manuscript could be improved. "Late meiotic recombination intermediates" could be considered as double Holliday junctions, for which the authors' results do not show that Rad51 is somehow required.

We have re-phrased the title (highlighted in yellow in the word document).

ii. Data quality/data interpretation

10. The authors propose that CDK-mediated phosphorylation of Sfr1 reduces its ability to function with Rad51 to regulate recombination. As such, the key mutant to be analysed is the *sfr1-7A*. The 7D is interesting in that it can help understand what phosphorylation does, but the 7A is what should be used to understand the biological relevance of Sfr1 phosphorylation. Unfortunately, one has to wait until Figure 7 to finally understand that the phosphorylation of Sfr1 is needed for something (in a *dbl2* mutant background) that is most likely linked to recombination. This data is interesting, but under-explored. What exactly fails in the 7A mutant? Why is the phenotype only seen in a *dbl2* mutant background? The only phenotype of the 7A mutant in recombination is that Rad51 foci seem brighter. Does CDK inhibition (post DSB formation) also cause that?

We are confused with this comment.

7A mutant is analyzed at the first figure in the manuscript (Figure 1), and from there in Figure 3, 4, 5, and 7 (and related Figures EVs). In all these figures, it is analyzed in parallel to the 7D mutant.

The first *in vivo* phenotype of the 7A mutant was shown in previous Figure 3 (now Figure 4), where a defect in Rad51 dynamics at the end of meiotic prophase is observed by a specific increase in Rad51 intensity in this particular mutant (the foci do not seem brighter, they are brighter; Figure 4D).

The model is proposed after the experiments and not before. We did not know that phosphorylation of Sfr1 had an inhibitory effect until we analyzed 7A and 7D mutants for recombination and Rad51 function in prophase (where Sfr1 promotes Rad51 activity), and showed that they were specifically impaired in the 7D mutant. Based on this, we proposed the model and looked for defects at late prophase (where Rad51 function is downregulated) in the 7A mutant (always with 7D allele in parallel).

Please notice, we start the subheading “Sfr1 phosphorylation contributes to the timely removal of Rad51 nucleofilaments at the end of meiotic prophase” in Results section as follows:

“If our model is correct we would expect non-phosphorylatable Sfr1-7A protein to maintain more stable interactions with Rad51 at late prophase that would interfere with proper chromosome disjunction. To study this...”

7A mutant shows a defect in Rad51 chromosome dynamics. The defect is clearly observed in *dbl2*⁺ cells (Figure 4D); however, this does not have an impact on chromatin segregation unless another mechanism for Rad51 removal is defective.

Please, notice also that in the Discussion we mentioned:

“Finally, it is also possible that Sfr1 phospho-regulation plays a role in the normal repair process earlier in prophase as a necessary step to remove Rad51 from the recombination intermediates facilitating downstream events such as DNA polymerases, ligases, translocases or resolvases entry. This could explain the small reduction in CO formation of the *sfr1-7A* allele (Figure 3B).”

In addition, although not statistically significant, Rad51 foci are also brighter in the *sfr1-7A* mutant in early prophase (Figure 4D).

Please, notice also that a 12% of the *sfr1-7A* zygotes shows Rad51 foci in anaphase I as shown in Figure 7C (this does not have an impact on chromatin segregation). It is possible that *sfr1-7A* cells tolerate this Rad51 dynamics alteration without major problems in chromosome segregation.

The pointed situation by the referee is very common. For example, in mitotic cells the *sfr1* phospho-mutants generated in Argunhan *et al*, 2020 and Liang *et al*, 2023 do not show sensitivity to DNA damage at least the Rad55-Rad57 pathway is compromised. Another example is Yen1 SSE. The defects in recombination intermediate resolution and chromosome segregation are only observed in a double *yen1 mus81* (or *mms4*) deletion mutant (Matos *et al*, 2011; Zakharyevich *et al*, 2012; Alonso-Ramos *et al*, 2021); or Sgs1 helicase, whose role in joint molecule resolution is only observed in the triple *mms4 yen1 slx4* mutant (Zakharyevich *et al*, 2012). These are examples of redundancy and backup mechanisms to ensure the efficiency of the process: in this case, timely resolution of joint molecules to implement an efficient CO formation, and therefore, a faithful chromosome segregation.

Inhibition of CDK activity during prophase led to a complete blockage of chromosome segregation given its fundamental role in this point of the cell cycle (see for example meiotic progression by nuclei counting in Fig. 2A). Cells block at the end of prophase (please notice Sfr1 accumulation at 4.5 and 5 h after 1-NM-PP1 addition in the Western blot shown in Fig. 2A, reflecting the blockage of meiotic progression). Therefore, no Rad51 retention in anaphase I can be addressed. We could address Rad51 foci intensity but given the pleiotropic effects of CDK and the accumulation of Sfr1 protein, we think the result interpretation would be difficult. The experiment with the *sfr1-7A* mutant version is more informative (Fig. 4).

11. The first paragraph of the Results section (179-202) discusses the five putative CDK sites previously described. Why do the authors mutate seven putative CDK sites? How were the additional two identified?

We identified 7 sites looking for CDK signatures. Massive proteomic data had identified 4 of them. We mutated all of them because, as indicated in the main text, the proteomic data were from vegetative growing cells and it was possible that the other residues were also phosphorylated in meiosis. In addition, all sites showed very high scores in phosphorylation prediction servers.

(Please, notice that we mentioned in the text that only 4 of the 5 identified sites by proteomic analysis were CDK sites).

We have explained this more clearly in the revised version.

In the first paragraph of the Results section we have added (highlighted in yellow in the word document):

“Except for Cipack *et al*, 2009 where a Sfr1-TAP purification was analyzed, these studies were not focused on Sfr1. Furthermore, all of them were conducted in vegetative cells.”

In the second paragraph of the Results section we have added (highlighted in yellow in the word document):

“As mentioned above, 4 phosphorylated residues in putative CDK sites have been identified in vegetative cells; besides, 3 additional putative sites are present in the protein which could be phosphorylated in meiosis. All 7 sites show ≥ 0.833 score in NetPhosYeast 1.0 prediction server.”

12. In Figure 1D, the shift in electrophoretic mobility is there from $t=2.5$ hours. There is then a second shift around 4-4.5 hours. Both are eliminated in the 7A mutant. Can the authors blot for cyclin in the same extracts? Does cyclin accumulation explain the shifts? Both, or only the second? Is the second shift also direct phosphorylation by CDK? Could it be PLK? Which of the two phosphorylation events is linked to the phenotypes described?

We have added blots for Cdc13 cyclin and Cdc2 in Figure 1.

Cdc13 levels have been described in mitotic and meiotic cultures. In meiosis, Cdc13 protein levels gradually accumulate already from S-phase, peak at the first meiotic division, and then decrease. Cdc13-dependent kinase activity is already observed at early prophase but maintained at lower levels due to Y15 inhibitory phosphorylation, and it is rapidly increased when cells enter meiosis I (Murakami and Nurse 1999; Borgne *et al*, 2002).

Notice that Sfr1 is phosphorylated from early prophase (Figure 1B). However, phosphorylation is maximal at late prophase just prior to meiosis I entry when a new low migrating band is visible. Thus, it is possible that the increment in Cdc13-associated kinase activity at late prophase promotes Sfr1 heavily phosphorylation as suggested by the referee.

Indeed, we mentioned this in the text (Discussion):

“Since the phosphorylation is maximal at the end of prophase, and performed at least by Cdc2-Cdc13 complexes which activity is maximal just prior to the first meiotic chromosome segregation, we propose that CDK-phosphorylation downregulates late IH invasions.”

In meiosis, 13 residues have been found phosphorylated in Sfr1 protein during meiosis prophase (using Sfr1-TAP purification, Sevcovicova *et al*, 2021). We did expect our Sfr1-7A mutant protein (containing mutations in 6 of these sites) to retain some retardation in Phos-tag gels. However, this was not the case and mobility shift in Phos-tag gels was almost abolished. Thus, it is possible that CDK is priming other phosphorylations.

It is known that CDK primes for DDK (Dbf4-dependent kinase) and Polo kinase. This has been well established for proteins involved in meiotic DSB formation (Mer2; Wan *et al*, 2008; Sasanuma *et al*, 2008) and DNA damage repair (Mms4; Gallo-Fernandez *et al*, 2012).

Three of the seven residues mutated in *sfr1-7A* are Thr-phospho Thr/Ser-Pro sites (T-pT/pS-P), and therefore, a priming role for additional phosphorylation can be recalled. These sites are T89, S147 and T152; and adjacent T146 and T151 residues could be also *in vivo* phosphorylated (S147 (or T146) and T152 (or T151) could not be assigned to an individual amino acid, Sevcovicova *et al*, 2021).

The rest of the residues described in Sevcovicova *et al*, 2021 (7 residues) are not adjacent to the CDK sites.

In any case, as a priming kinase the role of CDK in Sfr1 phospho-regulation is fundamental.

We have mentioned this in the new version of the manuscript in a new paragraph in the Discussion (highlighted in yellow in the word document).

13. There is always a pool of unphosphorylated Sfr1. How do the authors incorporate this in the model? wouldn't this pool of Sfr1 interfere with the processing of recombination intermediates?

The phosphorylation of a pool of the total protein is very common in western blot analysis. It is possible that the phosphorylated and unphosphorylated proteins are in different cellular compartments or that not all the molecules need to be phosphorylated to regulate the process.

In our case, notice that Sfr1 is gradually phosphorylated from early prophase, and phosphorylation is maximal just prior to meiosis I entry (Figure 1B). Please, notice that at this time point all the bands are retarded compared to early prophase position.

Notice also that total Sfr1-protein levels are reduced when cells enter chromosome segregation. This reduction is observed by western blot and time lapse microscopy (Figure EV4). This will also diminish the available protein for new nucleofilament formation.

We think this does not need to be mentioned in the manuscript.

14. Can the authors IP Sfr1 from cells and show that Rad51 binds the phosphorylated form, but not the unmodified form? Alternatively, does *in vitro* phosphorylation of Sfr1 by Cdc2 (e.g. as in Figure 2C) reduce/abolish the interaction with Rad51?

Rad51 interaction with Swi5-Sfr1 complex is weak and proposed to reflect a dynamic binding to confer flexibility to the interaction during nucleofilament elongation (Kuwabara *et al*, 2012, Argunhan *et al*, 2020).

Unfortunately, Sfr1-Rad51 interaction is not observed by IP. This has been reported by other labs (Iwasaki's lab in vegetate growing cells, Amakatsu *et al*, 2003; and Gregan's lab using the Sfr1-TAP for proteomic analysis in meiotic prophase, Sevcovicova *et al*, 2021).

We did also try the coIP *in vivo* since we normally use frozen cells and a freezer mill for our extract preparation. However, we did not success and we did not see the interaction. Our intention was to check if Sfr1-7A and Sfr1-7D differently interact with Rad51 *in vivo* as suggested by the referee, and to see whether the interaction of the wild-type protein was stronger at early prophase and weaker at late prophase.

The experiment proposed by the referee is complicated since relays in a previous *in vitro* kinase assay and several protein extracts prepared in parallel.

We have done an alternative experiment that we thought was more feasible (since phosphatase treatments are more efficient than kinase assays):

- Enrich EGFP-Sfr1 by GFP-trap in cells at late prophase (already phosphorylated)
- Split the GFP-trap in two
- Phosphatase treatment of one half (parallel treatment of the other half without phosphatase as control)
- Incubation with Rad51-expressing bacterial extracts
- Washes
- Rad51 detection by western blot.

We expected to detect Rad51 in the phosphatase treated GFP-trap and a reduction in the control no treated with phosphatase. Thus, we would test if phosphorylation impacts on Rad51 binding. We also included EGFP-Sfr1-7A in the experiment. We expected this unphosphorylatable version to interact with Sfr1 independently of the phosphatase treatment.

However, **we have not detected the Sfr1-Rad51 interaction in the experiment** (see Figure below). We did the experiment twice with different buffers in the last step (incubation with Rad51-expressing bacterial extracts), HB and pulldown buffer (both of them detailed in Material and Methods). EGFP-Sfr1 and EGFP-Sfr1-7A proteins were detected in the GFP-traps but Rad51 was not detected. Please notice that, in contrast to Sfr1 GFP-traps, Sfr1-7A GFP-traps were not detected with the anti-phospho (Ser) CDK-substrate antibodies (independently of the phosphatase treatment).

Two buffers were tested for the interaction: pull-down buffer and HB buffer supplemented with 100 mM NaCl (to increase ionic strength to that of pull-down buffer).

Therefore, we have also performed an alternative experiment based on an *in vitro* kinase assay of the GST-Sfr1 protein expressed in bacteria.

- GST-Sfr1 expression in bacteria and GST pull-down
- IP of Cdc2-Cdc13 from cells in meiosis I (IP in parallel with Protein A-Sepharose but without anti-Cdc13 antibody as a negative control)
- Mix GST-Sfr1 pull-down and Cdc13 IPs
- *In vitro* kinase assay
- Washes with the pull-down buffer
- Incubation with Rad51-expressing bacterial extracts
- Separation by centrifugation of Sepharose beads (bound fraction) and supernatant (flowthrough).
- Rad51 detection by western blot.

Since this type of experiments relies in the efficiency of the kinase assay we have used two different antibodies to IP Cdc13 from cells in meiosis I. We have used our regular commercial monoclonal anti-Cdc13 antibody (from Abcam)

and a rabbit anti-Cdc13 serum (generous gift from Dr. Sergio Moreno). We expected to increase the kinase assay efficiency by a better immunoprecipitation with the serum. However, this was not the case and both antibodies IP similar amount of Cdc2 (proxy of Cdc13 immunoprecipitation since direct detection is masked by immunoglobulins) (see Figure below).

In this experimental set up the interaction of GST-Sfr1 and Rad51 was observed. However, **phosphorylation by Cdc2-Cdc13 complexes was almost undetected**, and indeed similar amount of Rad51 was detected in the “bound fraction” after the kinase assays compared to the negative controls (unphosphorylated GST-Sfr1). In the case of the anti-Cdc13 serum, the cross-reaction of the Light Chain Specific secondary antibody with the heavy chains of the immunoglobulins in the serum masks the detection of a possible phosphorylation. In the case of the commercial anti-Cdc13 antibody, phosphorylation was barely detected in long exposures.

Thus, in the first experiment the phosphatase treatment was efficient but the experimental set up (GFP-Sfr1 from fission yeast and Rad51 from bacterial extracts) does not recapitulate the Sfr1-Rad51 interaction; and in the second experiment the Sfr1-Rad51 interaction is observed (both proteins from bacterial extracts) but GST-Sfr1 from bacteria is extremely inefficiently phosphorylated by Cdc2-Cdc13 to address the impact on Rad51 binding.

15. In Figure 1D, it is already clear that the stability of the 7A mutant is higher, in particular at later time points in meiosis. This is not mentioned in the description of the data.

We have mentioned this in the description of the data (highlighted in yellow in the word document).

16. Figure 3B and C - Where are the meiotic progressions, Western blots, etc. of these mutants? The protein levels of *sfr1-7A* and *sfr1-7D* are only shown in Figure 6B, but should be included here. Moreover, FACS analyses of DNA content would help assess if the kinetics of meiotic progression in the mutants are similar to the WT.

We run FACS analysis and perform nuclei counting in all our synchronous meiosis. Kinetics of meiotic progression in the mutants are similar to the wild-type control.

When introducing the experiment in the text, we have added a new sentence and a new supplementary Figure mentioning this (highlighted in yellow in the word document).

"sfr1-7D and *sfr1-7A* alleles did not alter meiotic progression and the strains showed the peak of meiosis I at 5 h after meiotic induction as *sfr1*⁺ cells (Appendix Figure S2)."

In addition, we have added these data to the experiment now in Figure 4 (previous Figure 3B-C).

In this particular kinetics we did not perform western analysis.

This experiment was done much later that the westerns in synchronous meiosis and the time lapse microscopy, and we firmly knew that the mutant proteins were normally expressed in prophase (where spreads were prepared) but extend their expression beyond chromosome segregation.

We do not find fundamental to add western blots to this particular experiment.

Please notice that western blots in previous Figure 6B (now Figure EV4A) have been quantified.

17. Figure 3B and D - Why are there so few Rad51 foci at these specific times in meiosis when DSB formation should be highest? Would one not expect to see more Rad51 foci? How was the specificity of the Rad51 and Sfr1 signals tested? The authors could stain nuclear spreads for EGFP-Sfr1 and Rad51 in a *rec12* mutant in which no DSBs are formed.

Rad51 foci are in the range of published data in *pat-114* synchronous meiosis (Sun *et al*, 2011).

The number is smaller than the reported number in zygotic meiosis (Lorenz *et al*, 2006), that in some aspects differs from *pat1-114* azygotic meiosis (Yamamoto & Hiraoka, 2003; Cipak *et al*, 2012).

Specificity of the Rad51 antibody (Clone 51RAD01) was already tested (Lorenz *et al*, 2006). In our case, the spreads of the *sfr1-WI* mutant serve as a control. Notice that EGFP-Sfr1-WI protein is normally expressed but it does not enter the nucleus and, therefore, is lost during spread preparation. No Rad51 signal is detected in these nuclei.

We have used previously this anti-GFP antibody (rabbit IgG fraction, A11122 Molecular Probes) in the laboratory and tested it for specificity. In this particular

case, the spreads of the *sfr1-WI* mutant serve as a control. No GFP signal is detected in these nuclei. Notice that EGFP-Sfr1-WI protein is normally expressed but it does not enter the nucleus and, therefore, is lost during spread preparation.

We have added a new supplementary figure (Appendix Figure S7) where spreads from *EGFP-sfr1 rec12⁺* and *sfr1⁺ delta-rec12* cells were stained with anti-GFP (rabbit IgG fraction, A11122 Molecular Probes) and anti-Rad51 antibodies (monoclonal 51RAD01/3C10, Invitrogen). Foci were detected for GFP and Rad51 in *EGFP-sfr1 rec12⁺* and not detected in *sfr1⁺ delta-rec12* spreads.

We have added the calling to this new figure in Material and Methods section (highlighted in yellow in the word document).

18. Figure 4A - What explains the slight reduction in intergenic recombination in the *sfr1-7A* mutant? Shouldn't interhomolog recombination be increased since Sfr1 is proposed to be a target of CDK-dependent phosphorylation? Is intragenic recombination altered in the *sfr1-7A* mutant?

Please, notice this Figure 4A is now Figure 3B.

7A mutant shows a defect in Rad51 chromosome dynamics. The defect is clearly observed in *dbl2⁺* cells (Figure 4D).

Please, notice that we mentioned (Discussion):

“Finally, it is also possible that Sfr1 phospho-regulation plays a role in the normal repair process earlier in prophase as a necessary step to remove Rad51 from the recombination intermediates facilitating downstream events such as DNA polymerases, ligases, translocases or resolvases entry. This could explain the small reduction in CO formation of the *sfr1-7A* allele (Figure 3B).”

In addition, Rad51 interaction with Swi5-Sfr1 complex is weak and proposed to reflect a dynamic binding to confer flexibility to the interaction during nucleofilament elongation (Kuwabara *et al*, 2012; Argunhan *et al*, 2020). We think that this dynamic might be also altered in the *sfr1-7A* mutant, and it can also contribute to the small reduction in the interaction with the homologous chromosome.

We have added few sentences in the Discussion to mention this (highlighted in yellow in the word document), just after the paragraph pointed above.

We have measured intragenic recombination in the *sfr1-7A* mutant. This was also suggested by referee 3.

Data are presented in Figure 3C and highlighted in yellow in the word document.

sfr1-7A mutant does not affect intragenic recombination. The difference between GC (not reduction) and COs (20% reduction) in the mutant could be explained by the proximity of the alleles in the GC assay and the higher requirement of a stable invasion for COs formation.

19. Figure 4B and C - Data for the *sfr1Δ* and *sfr1-7A* mutants are missing as an important comparison.

This is related to the previous comment 18.

We have added *sfr1* deletion and 7A mutants in panel B (gene conversion analysis; now Figure 3C). As for COs, delta *sfr1* reduces GC and the reduction is statistically different (stronger) from the observed one in *sfr1-7D* mutants.

This is a shared comment with referee 3.

We have eliminated part C in this figure as suggested by referee 3.

20. In Figures 6/7, the authors show that the 7A mutant is more stable. What if this is the reason for the anaphase phenotype observed? Can the authors exclude that the increased stability leads to some aberrant function that interferes with chromosome segregation?

We mentioned in the Discussion when describing the presence of PEST sequences:

“Regulation of Sfr1 protein stability at the end of meiotic prophase might represent an additional mechanism to inhibit late IH invasions.”

Please notice that 7D mutant protein is also more stable but it does not impair chromosome segregation at anaphase I. Thus, it is not just the presence of the protein but the presence of a reluctant protein to phosphorylation (and inhibition).

21. At the very end of the paper the authors analyze the Sfr1-WI phenotype. Why at the end? I can't follow the logic.

sfr1-WI phenotype is analyzed early in the manuscript, in Figure 3A, Figure 4A-B and related supplementary Figures (Figure EV2 and Figure EV3A). We used this allele as a control since it was reported to affect Rad51 binding in *in vitro* assays. This is stated when we introduced the mutant in the Results section.

During the time lapse experiments we realized that the protein does not enter the nucleus and checked for protein levels and recombination proficiency (please, notice that this allele was reported to be proficient for DNA repair in vegetative cells (Argunhan *et al*, 2020)).

As this is not the central part of the manuscript, all this information was gathered in a supplementary figure (now Appendix Figure S4).

22. Have the authors looked at spore viability in the mutants generated? This would normally be one of the first things to test.

We have plated many thousands of spores for our different recombination assays and we have not noticed a reduction in spore viability in the *sfr1-7A* and *sfr1-7D* mutants.

We have added this information to the Result section and Figure 7 (highlighted in yellow in the word document). We have quantified the spore viability in wild-type, *sfr1* delta, *sfr1-7A* and *sfr1-7D* mutants, and double mutants in combination with *dbl2* delta (Figure 7D).

sfr1-7A and *sfr1-7D* single mutants show wild-type levels of spore viability. However, *sfr1-7A* (and not *sfr1-7D*) mutants worsen the defects of the *dbl2* delta mutant.

This is a shared comment with referee 1 and 3.

23. How does the phenotype of 7A compare to the 13A phenotype described in the 13A in the Sevcovicova 2021 paper?

In Sevcovicova et al, 2021 the only meiotic phenotype addressed is homolog disjunction in meiosis I. 13A protein expression rescues the homolog disjunction defect of a *sfr1* deletion mutant. We interpret this as 13A protein being proficient for recombination and recovering the recombination defect of the *sfr1* deletion mutant.

Thus, this is consistent with our *sfr1-7A* mutant showing quite normal rates of COs (81%).

We have not included this comparison in the revised version.

24. Figures 5-6 are redundant with the spreads in Figure 3B

This point is raised above in comment 6.

25. Figure 4D: does cyclin or *cdc2* o/E change the proportion of Sfr1 phosphorylation? I am not convinced that the assay shows what the authors claim. If the *sfr1-7D* mutant shows a phenotype in the same assay as used in Figure S5, why wouldn't the cyclin/Cdc2 o/E have a phenotype? Could it be that the phosphomimetic *sfr1* mutant is only partially functional, as proposed in Sevcovicova et al. 2021?

VL1 system (and similar PS1 system) has been regularly used to address USCR (Latypov et al, 2010; Mallela et al, 2011); we do not understand why the result with this assay does not convince the referee about the increment in USCR when CDK activity is modulated.

The number of Sfr1 molecules that are phosphorylated when CDK is modestly increased, would impair interhomolog Rad51 nucleofilament function, promoting its disassembly and facilitating intersister recombination. Please notice in the table the primary data in this experiment. Basal USCR is very low, 73-80 Ade⁺ per 10⁶ viable spores, and moderately raises to 110-135 in the integrants.

When copy number of *cdc2* or *cdc13* is increased (1 or 2 extra genomic copies; thus, no high overexpression) many CDK substrates will be phospho-modulated and, in addition, these substrates will continue to have phosphorylation-dephosphorylation cycles. In the case of the *sfr1-7D* mutant, the protein cannot be phospho-modulated. We expect this to have a stronger impact on homolog invasion, and therefore, on the recombination assays pointed by the referee. We think these experimental situations are not fully comparable.

Since the Sfr1-7D protein confers a loss of function phenotype, we cannot completely rule out that the mutant protein is only partially functional. However, several observations indicate this is unlikely. First, the mutations are in a disorganized region and, therefore, we expect they will not distort any structure. Indeed, the protein shows the same disorganized profile when run in AlphaFold (as the wild-type Sfr1 and Sfr1-7A proteins). Second, Liang et al, 2023 checked by biophysics means their Sfr1-5A and Sfr1-5D proteins, which share 4 of our modifications, and both proteins behave as the wild-type protein. Thus, we do not expect our mutant to alter general protein structure.

Please notice that in Figure EV1B we have added the predicted structures of the Sfr1, Sfr1-7A and Sfr1-7D proteins in complex with Swi5 (modelled by ColabFold). N-terminal domains of the three proteins are disorganized and they do not interfere with the core structured complex (Swi5/Sfr1-C-terminal).

To address comment 14, we have performed two experiments that would help to circumvent the concern about functionality of the *sfr1-7D* mutant (Sfr1

phospho-modulation and impact on Rad51 binding). However, in one of the experiments the Sfr1-Rad51 interaction was not detected in the experimental set up and the phosphorylation dependency was not possible to address (see the experiment above).

In the additional experiment, Sfr1-Rad51 interaction was observed (both proteins from bacterial extracts) but GST-Sfr1 from bacteria was very inefficiently phosphorylated by Cdc2-Cdc13 to address the impact on Rad51 binding.

26. Videos 1-4 and 4-8 - There is no indication of the time. For the sake of clarity and better comparability with the figures, the time should be included. Time indication have been added to the final version of these videos (presented now as Movie EV1-9).

27. Figure S4A - No granular/accumulated signal can be distinguished in either wild type or mutants. To clarify whether the putative accumulated signals are indeed Rad51 foci, the analysis should also be performed in *rec12Δ* mutants. If ECFP-Rad51 does not indeed form a detectable granular/accumulated signal, potentially representing foci, this would also help to explain why the putative Rad51 foci are detected in the time-lapse movies but not on the nuclear spreads (discussed in 442-445).

We are not sure about this comment. Foci are observed in the images and movies. They are not very prominent but they are distinguished.

The suggested control has been already done in Yang *et al*, 2015, 10.1083/jcb.201501035, (see Figure 3); foci were not detected in a *rec12* deletion mutant.

In that case the Rad51-ECFP:Rad51 ratio was 1:1 (*h⁹⁰ rad51-ECFP-ura4⁺-rad51* cells) and 1:2 in our experiments (*h⁺ rad51-ECFP-ura4⁺-rad51 (sfr1⁺ or mutant allele) X h⁻* (or *sfr1* mutant allele) crosses).

rad51-ECFP-ura4⁺-rad51 is an insertion of a genomic fragment containing the *rad51-ECFP* version; it is integrated at the *rad51* locus and maintains the wild-type copy (Akamatsu *et al*, 2007).

We have added this control to Appendix Figure S3 (previous Appendix Figure S4). The skewness of the Rad51-ECFP signal is statistically reduced in the *rec12* deletion mutant without affecting total intensity, though it is not completely eliminated as observed for the EGFP-Sfr1 signal. This indicates a different nature of the signals, with the Rad51-ECFP signal being more granular and the EGFP-Sfr1 signal more homogenous/flatter. We have added few sentences in the main text when the experiment is described (highlighted in yellow in the word document). We have also added a new EV movie.

28. Lines 416 - 417 "Importantly, EGFP-Sfr1-7A and EGFP-Sfr1-7D proteins were similarly expressed during meiotic prophase analyzed by signal intensity in time lapse experiments as well as by western blot in *pat1-114* synchronous meiosis (Figure 6A and B, and Table S1)." Analysis of the protein levels of either Sfr-7A or Sfr-7D showed that they are expressed or stabilised longer than the wild type, well beyond the putative M1 and MII event. Could there be a defect in meiotic progression in these mutants? These data should be combined with a comparison of the kinetics of meiotic progression, such as

FACS or MI and MII analysis. In addition, analysis of Rad51 protein levels showed normal kinetics of Rad51 levels during meiosis (Figure S4C).

Please notice that the pointed sentence refers to the expression **during meiotic prophase**, when spreads were prepared and analyzed.

This is related to comment 16.

We run FACS analysis and perform nuclei counting in all our synchronous meiosis. Kinetics of meiotic progression in the mutants are similar to the wild type control.

When introducing this experiment in the text, we have added a new sentence and a new supplementary Figure mentioning this (**highlighted in yellow in the word document**).

“*sfr1-7D* and *sfr1-7A* alleles did not alter meiotic progression and the strains showed the peak of meiosis I at 5 h after meiotic induction as *sfr1*⁺ cells (Appendix Figure S2).”

In addition, we have added the data to the experiment in previous Figure 6B (now Figure EV4A). Please notice also that western blots in this figure panel have been quantified.

29. Figure 7A - The immunofluorescence images lack any description of the phenotypes observed and the time of sampling. Furthermore, do all *sfr1-7A* and *sfr1-7D* mutants behave in the same way and enter MI and II with similar kinetics?

The images exemplify the scored phenotypes described in the legend of the graphs just below them (in the same panel, Figure 7A). We have stated this in the figure legend.

We have added labels to the images to make clear they are examples of the categories quantified in the graphs.

These are fixed zygotes from asynchronous meiosis as stated in Material and Methods (“...transferred to PMG-NH₄Cl plates and incubated at 25°C for 12-15 h.”). These cells were induced to enter meiosis in plates. These are not time lapse experiments. It is not possible to address any kinetics in this type of experiments.

We have stated in the figure legend that these cells are **asynchronously** induced to enter meiosis (**highlighted in yellow in the word document**).

Related to this, we have added the spore viability for all these mutants (single and double mutants) to Figure 7 (Figure 7D). The data correlates with the segregation problems. *sfr1-7A* and *sfr1-7D* single mutants show wild-type levels of spore viability. However, *sfr1-7A* (and not *sfr1-7D*) mutants worsen the defects of the *dbl2* delta mutant.

30. Figure 7B - Where do the Rad51 foci predominantly localize to?

Rad51 foci are associated to the missegregated DNA (in one of the unequal segregated DNA masses, the lagging DNA masses or in the unsegregated DNA).

We have added the comment “associated to the missegregated DNA” when describing Rad51 retention (**highlighted in yellow in the word document**).

31. 536-538 - "This result was surprising given that Sfr1-WI protein retains some ability to bind to Rad51 when compared to Sfr1-7D protein (Figure 3A)

(Argunhan et al., 2020)". The interaction between Sfr1-WI and Rad51 seems to be severely affected and could cause the observed recombination defects.

The result surprised us because, although strongly affected, some binding to Rad51 can be observed in the pulldown assays compared to Sfr1-7D protein where no Rad51 is pulled down. This anticorrelates with the recombination proficiency that is better in the *sfr1-7D* allele compared to the *sfr1-WI* allele and statistically different (Appendix Figure S4). Indeed, this anticorrelation is also observed when analyzing Rad51 chromatin binding in nuclear spreads (Figure 4). We would not expect a worse recombination rate and Rad51 chromatin binding for the *sfr1-WI* allele compared to the *sfr1-7D* mutant.

We have added this rationale to the pointed paragraph to better explain our argument (highlighted in yellow in the word document).

The interaction between Sfr1-WI and Rad51 is very weak but the Sfr1-WI protein barely enters the nucleus. It is the same that it binds or not to Rad51 *in vitro*. *In vivo*, Sfr1-WI and Rad51 proteins are in different cellular compartments (please notice that Rad51 enters the nucleus in the *sfr1-WI* mutant). It is the defect in nuclear localization which explains the defect in recombination of the *sfr1-WI* allele.

32. Figure 8 - The predicted structure by AlphaFold, and the structure published in Kuwabara et al. 2012, are presented without any description or indication of specific domains or the disordered regions. As such, they do not add anything significant in the context of this study and could be omitted.

We would like to keep these pictures. We have described them better in the new version.

Please notice that we have added the predicted structures of the Sfr1, Sfr1-7A and Sfr-7D proteins in complex with Swi5 (ColabFold; <https://colab.research.google.com/github/sokrypton/ColabFold/blob/main/AlphaFold2.ipynb>). N-terminal domains of the three proteins are disorganized and they do not interfere with the core structured complex (Swi5/Sfr1-C-terminal). This is presented in the new Figure EV1B.

Referee #3:

Palacios-Blanco et al.

This manuscript will be of interest for researchers studying homologous recombination, especially during meiosis, with an interest in molecular mechanisms and how meiotic recombination is regulated. The authors report the function of phosphorylation sites in the N-terminus of the recombination mediator protein Sfr1 in the fission yeast *Schizosaccharomyces pombe*. Sfr1 forms a complex with Swi5 which functions as a cofactor for the central recombination proteins Rad51 and Dmc1. The manuscript describes the identification of 7 CDK phosphorylation sites that negatively regulate the interaction of Sfr1-Swi5 with Rad51 and lead to disruption of Rad51 function in late meiotic prophase before meiosis I chromosome segregation. The authors convincingly show that Sfr1 is phosphorylated during meiotic prophase by CDK and that the identified sites are involved. Sfr1 phosphorylation disrupts interaction with Rad51, is critical for removal of Rad51 from meiotic chromosomes at the end of prophase, and required for accurate chromosome segregation during meiosis. The authors convincingly identified a novel regulatory mechanism restricting recombination to meiotic prophase.

Comments:

1) It is not clear how the individual phosphorylation sites were identified, what the evidence is that they are phosphorylated and what their relationship is to the sites studied by Liang P, Lister K, Yates L, Argunhan B, Zhang X (2023 Phosphoregulation of DNA repair via the Rad51 auxiliary factor Swi5-Sfr1. *J Biol Chem* 299: 104929). This requires more information and discussion.

Proteomic studies identified 5 phosphorylated residues in Sfr1, 4 of them CDK sites. However, **none of these studies were focused on Sfr1.**

In addition, Sevcovicova *et al*, 2021 described that 6 out the 7 predicted CDK sites in Sfr1 are indeed *in vivo* phosphorylated during meiotic prophase. **In this case the study was focused on Sfr1 (using a Sfr1-TAP purification).**

We mutated all 7 residues because, as indicated in the main text, the proteomic data were from vegetative growing cells and it was possible that the other residues were also phosphorylated in meiosis. In addition, all sites showed high scores in phosphorylation prediction servers.

We have explained this more clearly in the revised version.

In the first paragraph of the Results section we have added (highlighted in yellow in the word document):

“Except for Cipack *et al*, 2009 where a Sfr1-TAP purification was analyzed, these studies were not focused on Sfr1. Furthermore, all of them were conducted in vegetative cells.”

In the second paragraph of the Results section we have added (highlighted in yellow in the word document):

“As mentioned above, 4 phosphorylated residues in putative CDK sites have been identified in vegetative cells; besides, 3 additional putative sites are present in the protein which could be phosphorylated in meiosis. All 7 sites show ≥ 0.833 score in NetPhosYeast 1.0 prediction server.”

We have also added a sentence to Figure 1C legend to indicate that the highlighted phosphorylated CDK sites were identified in meiosis using a Sfr1-TAP purification (highlighted in yellow in the word document).

In the Discussion section we already describe the relationship with the residues studied in Liang *et al*, 2023. In this paragraph (starting in line 701), identified phosphorylated residues in proteomic studies (in vegetative growth), identified phosphorylated Sfr1 residues in meiosis (Sevcovicova *et al*, 2021) and the residues studied in Liang *et al*, are all collectively discussed.

As we mention in this paragraph, Liang *et al*, used mutants in “5 reported phosphorylated sites”. They do not identify them. They selected them based exclusively on massive phosphoproteomic studies in vegetative cells. They did not include the residues identified in meiotic prophase in the later study by Sevcovicova *et al*.

2) It is not clear, if all 7 sites control the function or a subset, as only the 7A/D mutants are analyzed. I am not suggesting conducting more experiments on subsets of sites. Maybe some data are available. Some discussion would be helpful.

We have just analyzed 7A/7D mutants. No information on subsets of sites was generated in our study.

We planned to use GFP-trap for EGFP-Sfr1 enrichment and phospho-peptide identification in meiotic prophase. However, Sevcovicova and colleagues work (using a Sfr1-TAP purification) was published during our project and they identified 6 out the 7 CDK sites altered in our 7A/7D mutants; we decided to continue our work with the mutant harbouring the 7 mutations (7A/7D).

We discuss in the paragraph starting in line 701 that 5A/5D mutants in Liang *et al*, 2023 contains 4 of the residues we have studied. Thus, it is formally possible that 4D/4A (on Thr73, Ser109, Ser116, and Ser165) mutants recapitulates 7D/7A mutants. In addition, Ser109, Ser116, Ser147, and Ser165 residues are well conserved among *Schizosaccharomyces* species (Appendix Figure S5); thus, this is another interesting mutant to analyze.

We have extended this paragraph of the Discussion section to mention this (highlighted in yellow in the word document).

3) Figure 1B: What is the n?

This is the number of different independent experimental repeats in the quantification. We have changed “n 3” to “n=3” in figures and main text as suggested by reviewer 1.

4) Figure 2b, C: What is the input amount. Define C- in part C. What is the negative control?

The input amount is now indicated in the figure legend and Material and Methods section (highlighted in yellow in the word document).

C- definition is stated in the figure legend and it is explained in more detail in Material and Methods section. It is a kinase assay with dephosphorylated EGFP-Sfr1 (GFP-trap) as a substrate and a negative control IP (Protein A Sepharose without anti-Cdc13 antibody); it is a negative control of a parallel IP with anti-Cdc13 antibodies.

We have re-designed the figure labeling to make clear the nature of the negative control: “-CDK (-Ab IP)” and “+CDK (Cdc13 IP)”.

5) Line 284-6, Figure 3A: The statement needs quantitation of the interaction defects.

We have quantified the interaction of Rad51 and Sfr1 proteins in pulldown assays (n=3 independent experiments). Quantification is shown in the same figure.

GST-Sfr1-7A protein significantly binds to Rad51 with an 81% efficiency compared to control GST-Sfr1 protein (n=3, *p value* 0.5709).

Description of the result is highlighted in yellow in the word document (main text and Material and Methods section).

We have reinforced this result by a “Yeast Two Hybrid” assay. Sfr1-7A and not Sfr1-7D interacts with Rad51 in this experiment. This is shown in a new Extended View (EV) Figure (Figure EV1A). The new experiment is mentioned in the Results section and in Materials and Methods section (highlighted in yellow in the word document).

6) Figure 4: The recombination data are good, but B and C lack the *sfr1* deletion and 7A for comparison? Are these data available or published for the *sfr1* deletion? I do not think that these data will add much to the papers. Some discussion would be helpful.

Yes, data for *sfr1* deletion in these genetic recombination assays have been published (Lorenz *et al*, 2014). Gene conversion is strongly affected and CO associated with GC (NCO/CO balance) modestly reduced.

We have measured intragenic recombination in the *sfr1-7A* mutant (this is a shared comment with referee 2) and delta *sfr1*.

Data are presented in Figure 3C and highlighted in yellow in the word document.

sfr1-7A mutant does not affect intragenic recombination. The difference between GC (not reduction) and COs (20% reduction) in the mutant could be explained by the proximity of the alleles in the GC assay and the higher requirement of a stable invasion for COs formation.

Delta *sfr1* reduces GC to 30% of the wild-type levels. This is statistically different from the 48% observed in the *sfr1-7D* mutant.

7) Figure 4C. I am not sure what the point of this experiment is and what it adds to the manuscript. There are no *sfr1* data. I suggest eliminating this part or relegating this to the supplement.

We have eliminated part C in this figure as suggested.

It is not a central part of our argument (role of Sfr1 phospho-regulation in homologous invasion) and it would need a more detailed explanation.

In fission yeast it has been proposed that Swi5-Sfr1 complex protects the D-loop from the unwinding action of Fml1 helicase, promoting in this way the formation of Holliday junctions and COs (Lorenz *et al*, 2014). This is the reason we explored a possible role of Sfr1 phospho-regulation in NCO/CO balance. We used only the 7D allele since it is the one with a strong reduction in CO levels, indicating the inhibition of Sfr1 function by phosphorylation. We used *fml1* deletion as a control with a clear unbalance to CO formation.

We did not see a phenotype with this 7D allele. This indicates that the intermediates capable to establish invasion with the homologous chromosome in this mutant are not affected in their fate to be repaired as NCO or COs. Thus, Sfr1 phosphorylation does not seem important for NCO/CO balance regulation. As *sfr1* deletion mutants shows a mild reduction of this balance (less COs associated to gene conversion event) (Lorenz *et al*, 2014), it seems that it is the presence and not the phosphorylation state of Sfr1 which protects the D-loop from Fml1 helicase.

We could add this argument to the description of this result, but as mention above, this is not a central part of our manuscript. In addition, we did not have the *sfr1*-delta strains when we did this experiment.

8) What is the effect of *sfr1*-7A and -7D on spore viability? These data should be added.

We have added this information to the Result section and Figure 7 (highlighted in yellow in the word document). We have quantified the spore viability (Figure 7D) in wild-type, *sfr1* delta, *sfr1*-7A and *sfr1*-7D mutants, and double mutants in combination with *dbl2* delta.

sfr1-7A and *sfr1*-7D single mutants show wild-type levels of spore viability. However, *sfr1*-7A (and not *sfr1*-7D) mutants worsen the defects of the *dbl2* delta mutant.

This is a shared comment with referee 1 and 2.

9) Figure 5/6: Why is the quantitation of Figure 5 presented in Figure 6?

We have reorganized the figures.

Please notice that previous Figure 6A (total intensity quantification) is now part of Figure 5 (5B).

10) Figure 6B requires quantitation.

Please notice that EGFP-Sfr1 and phospho-mutant protein levels are quantified in panel A (now Figure 5B) using the intensity of the proteins during time lapse microscopy.

The pointed western blots stress the point that the phospho-mutant proteins are present after chromosome segregation, which is obvious by comparison of the blots (and by microscopy shown in the next panel).

We have quantified these blots (from two independent experiments for each *sfr1* mutant).

Additional points

) Line 81: established not stablished

This has been corrected.

) L145: SSE not SEE

This has been corrected.

) Line 427: though not thought

This has been corrected.

) Genotype labeling is inconsistent with regards to capitalization and use of italics.

We are confused about this. Genotypes are in lowercase and italics.

In the case of fusion proteins, the strain genotype is indicated as *EGFP-sfr1*, *EGFP-sfr1-7A*.... since EGFP is an acronym that cannot be pronounced as a word and it is written in capital letter. We have always written this type of genotype in this manner.

) Figure 3A, 4A-C, 7A-C: Define the controls and provide detail what they are. The strains (and plasmids) used in these experiments are indicated in the Figure legends, and the strain genotype is detailed in Appendix Table S5. In each experiment, all the strains are exactly the same except for the particular mutation in the gene under study (*sfr1-7A*, *sfr1-7D*...).

We have added "*sfr1*⁺" at several places in the text to make clear that control cells were *sfr1*⁺ (highlighted in yellow along the word document). This has been also added to "control" labels in the figures.

In the case of Figure 3A, we have also added "empty vector" in the figure legend to make clear the nature of the negative control (C-). Please notice that in the scheme of the experiment, "GST (C-), GST-Sfr1 and mutant versions" also indicates the nature of the negative control.

Dr. Cristina Martín-Castellanos
Instituto de Biología Funcional y Genómica (IBFG); CSIC-USAL
Zacarías González, 2
Salamanca, Salamanca 37007
Spain

31st Jul 2024

Re: EMBOJ-2023-116463R
CDK phosphorylation of Sfr1 downregulates Rad51 function in late-meiotic homolog invasions

Dear Dr. Martín-Castellano,

Thank you again for submitting your revised manuscript to The EMBO Journal. It has now been seen once more by the original referees, whose comments are copied below. Since all three were generally satisfied with your revisions and responses to the initial comments, we should be happy to consider the study further for EMBO Journal publication, pending satisfactory addressing of a number of remaining editorial issues:

- Please include a dedicated "Data Availability" section at the end of the Material and Methods; should there no data deposition to public repositories linked to the study, this should still be stated as "This study includes no data deposited in external repositories." For details, please see <https://www.embopress.org/page/journal/14602075/authorguide#dataavailability>
- Please double-check all citations in the reference list, as many of them appear to be still incomplete (lacking page or eLocator numbers).
- Please rename the Conflict of Interest section into "Disclosure and Competing Interests Statement", in accordance with our updated Guide to Authors (<https://www.embopress.org/competing-interests>)
- Please double-check to make sure to all relevant funding information in the manuscript is congruent with the info entered into our submission system. Currently missing in the submission system are: the "Ministerio de Universidades" with an FPU19/03456 grant; "Plan Operativo de Empleo Juvenil (Fondo Social Europeo e Iniciativa de Empleo Juvenil)"
- I would propose that Appendix Tables S1-4 should be uploaded in directly accessible data format, i.e. as four individual DOCX or XLSX files. For this, they would need to be renamed as Expanded View tables (title and in-text call-out: "Table EV1-4"), with their respective title/legend included in a separate tab. Accordingly, the remaining Appendix Table (currently: Appendix Table S5) should be renamed into Appendix Table S1 (also its in-text reference).
- Please pre-face the Appendix with the header "Appendix" and the article title and author names. Please add each Appendix Figure legend directly underneath the respective figure, instead of in one single block. The references listed after the final Appendix Table should be collated in a dedicated list of "Appendix References" at the end of the Appendix, and should also be listed in the Appendix Table of Contents.
- In the figures/figure legends, please note that the exact p values need to be provided in the legends of Figures EV 3b-c; and that the box plots need to be defined in terms of minima, maxima, bounds of box and whiskers, and percentile in the legends of Figures 4c; EV 3b.
- Finally, during our routine pre-acceptance data checks, we noticed that several micrographs and blot panels appear to have been reused in different figures without sufficient indication/clarification. Please make sure to clearly indicate in all respective figure legends if and why certain panels are repeatedly shown. Please also explain and clarify in your final response letter to the editor. In particular, micrographs appear to be re-used between Figures 4 and EV2; and blots are reused in Figures EV4 and Appendix Figures S4 and S5. Although this is briefly mentioned in only one of the legends, it appears puzzling that certain tubulin controls would appear in different combinations between figures, when they should supposedly come from the same experiments.

I am therefore returning the manuscript to you for a final round of revision, to allow you to provide the necessary clarifications, and to upload revised files including the requested presentational and editorial modifications. After that, we should hopefully be able to proceed with formal acceptance and production of the manuscript.

Yours sincerely,

Hartmut Vodermaier

*** PLEASE NOTE: All revised manuscripts are subject to initial checks for completeness and adherence to our formatting guidelines. Revisions may be returned to the authors and delayed in their editorial re-evaluation if they fail to comply to the following requirements (see also our Guide to Authors for further information):

- 1) Every manuscript requires a Data Availability section (even if only stating that no deposited datasets are included). Primary datasets or computer code produced in the current study have to be deposited in appropriate public repositories prior to resubmission, and reviewer access details provided in case that public access is not yet allowed. Further information: embopress.org/page/journal/14602075/authorguide#dataavailability
- 2) Each figure legend must specify
 - size of the scale bars that are mandatory for all micrograph panels
 - the statistical test used to generate error bars and P-values
 - the type error bars (e.g., S.E.M., S.D.)
 - the number (n) and nature (biological or technical replicate) of independent experiments underlying each data point
 - Figures may not include error bars for experiments with $n < 3$; scatter plots showing individual data points should be used instead.
- 3) Revised manuscript text (including main tables, and figure legends for main and EV figures) has to be submitted as editable text file (e.g., .docx format). We encourage highlighting of changes (e.g., via text color) for the referees' reference.
- 4) Each main and each Expanded View (EV) figure should be uploaded as individual production-quality files (preferably in .eps, .tif, .jpg formats). For suggestions on figure preparation/layout, please refer to our Figure Preparation Guidelines: <http://bit.ly/EMBOPressFigurePreparationGuideline>
- 5) Point-by-point response letters should include the original referee comments in full together with your detailed responses to them (and to specific editor requests if applicable), and also be uploaded as editable (e.g., .docx) text files.
- 6) Please complete our Author Checklist, and make sure that information entered into the checklist is also reflected in the manuscript; the checklist will be available to readers as part of the Review Process File. A download link is found at the top of our Guide to Authors: embopress.org/page/journal/14602075/authorguide
- 7) All authors listed as (co-)corresponding need to deposit, in their respective author profiles in our submission system, a unique ORCID identifier linked to their name. Please see our Guide to Authors for detailed instructions.
- 8) Please note that supplementary information at EMBO Press has been superseded by the 'Expanded View' for inclusion of additional figures, tables, movies or datasets; with up to five EV Figures being typeset and directly accessible in the HTML version of the article. For details and guidance, please refer to: embopress.org/page/journal/14602075/authorguide#expandedview
- 9) To facilitate reproducibility and cross-laboratory adoption of methodologies, please structure the Materials & Methods section as outlined in our guide to authors, including a completed Reagents and Tools Table that can be downloaded from our author guidelines as well (<https://www.embopress.org/page/journal/14602075/authorguide#structuredmethods>).
- 10) Digital image enhancement is acceptable practice, as long as it accurately represents the original data and conforms to community standards. If a figure has been subjected to significant electronic manipulation, this must be clearly noted in the figure legend and/or the 'Materials and Methods' section. The editors reserve the right to request original versions of figures and the original images that were used to assemble the figure. Finally, we generally encourage uploading of numerical as well as gel/blot image source data; for details see: embopress.org/page/journal/14602075/authorguide#sourcedata

At EMBO Press, we ask authors to provide source data for the main manuscript figures. Our source data coordinator will contact you to discuss which figure panels we would need source data for and will also provide you with helpful tips on how to upload and organize the files.

In the interest of ensuring the conceptual advance provided by the work, we recommend submitting a revision within 3 months (29th Oct 2024). Please discuss the revision progress ahead of this time with the editor if you require more time to complete the revisions. Use the link below to submit your revision:

Link Not Available

Referee #1:

The authors addressed all of my comments. In my opinion, the manuscript has been improved and I think it is ready for publication.

Referee #2:

Overall, the authors have made a great effort to address the reviewers comments.

I fully agree with the authors that they should be the ones deciding what to include in the manuscript, and in which order. My comments did not intent to "force" them to change anything, but rather to encourage them to think about ways to make the work more accessible to non specialist readers (and scientifically even more solid).

Referee #3:

Palacios-Blanco et al. revised manuscript

This manuscript will be of interest for researchers studying homologous recombination, especially during meiosis, with an interest in molecular mechanisms and how meiotic recombination is regulated. The authors report the function of phosphorylation sites in the N-terminus of the recombination mediator protein Sfr1 in the fission yeast *Schizosaccharomyces pombe*. Sfr1 forms a complex with Swi5 which functions as a cofactor for the central recombination proteins Rad51 and Dmc1. The manuscript describes the identification of 7 CDK phosphorylation sites that negatively regulate the interaction of Sfr1-Swi5 with Rad51 and lead to disruption of Rad51 function in late meiotic prophase before meiosis I chromosome segregation. The authors convincingly show that Sfr1 is phosphorylated during meiotic prophase by CDK and that the identified sites are involved. Sfr1 phosphorylation disrupts interaction with Rad51, is critical for removal of Rad51 from meiotic chromosomes at the end of prophase, and required for accurate chromosome segregation during meiosis. The authors convincingly identified a novel regulatory mechanism restricting recombination to meiotic prophase.

Revision:

In the revision, the authors adequately address my comments by adding new experimental data (Figure EV1, Figure 3C, Figure 7), providing necessary quantitations (Figure 3A, 6B), adding more discussion about the phospho-site mutants, and making necessary text changes. In my opinion, this manuscript will make a nice contribution.

Comments:

1) It is not clear how the individual phosphorylation sites were identified, what the evidence is that they are phosphorylated and what their relationship is to the sites studied by Liang P, Lister K, Yates L, Argunhan B, Zhang X (2023 Phosphoregulation of DNA repair via the Rad51 auxiliary factor Swi5-Sfr1. *J Biol Chem* 299: 104929). This requires more information and discussion.

Revision:

The identification of the phosphorylation sites is now sufficiently discussed.

2) It is not clear, if all 7 sites control the function or a subset, as only the 7A/D mutants are analyzed. I am not suggesting conducting more experiments on subsets of sites. Maybe some data are available. Some discussion would be helpful.

Revision:

The discussion has been added.

3) Figure 1B: What is the n?

Revision:

OK

4) Figure 2b, C: What is the input amount. Define C- in part C. What is the negative control?

Revision:
OK

5) Line 284-6, Figure 3A: The statement needs quantitation of the interaction defects.

Revision:
The quantitation has been added. The authors also added a two-hybrid analysis that corroborates the conclusion that Sfr1-7A but not Sfr1-7D interacts with Rad51 In Figure EV1.

6) Figure 4: The recombination data are good, but B and C lack the sfr1 deletion and 7A for comparison? Are these data available or published for the sfr1 deletion? I do not think that these data will add much to the papers. Some discussion would be helpful.

Revision:
The data have been added.

7) Figure 4C. I am not sure what the point of this experiment is and what it adds to the manuscript. There are no sfr1 data. I suggest eliminating this part or relegating this to the supplement.

Revision:
This part was eliminated.

8) What is the effect of sfr1-7A and -7D on spore viability? These data should be added.

Revision:
The experimental data were added.

9) Figure 5/6: Why is the quantitation of Figure 5 presented in Figure 6?

Revision:
OK

10) Figure 6B requires quantitation.

Revision:
The blots were quantified.

Additional points:

-) Line 81: established not stablished
-) L145: SSE not SEE
-) Line 427: though not thought
-) Genotype labeling is inconsistent with regards to capitalization and use of italcs.
-) Figure 3A, 4A-C, 7A-C: Define the controls and provide detail what they are.

Revision:
OK

Editorial letter

Dr. Cristina Martín-Castellanos
Instituto de Biología Funcional y Genómica (IBFG); CSIC-USAL
Zacarías González, 2
Salamanca, Salamanca 37007
Spain

31st Jul 2024

Re: EMBOJ-2023-116463R
CDK phosphorylation of Sfr1 downregulates Rad51 function in late-meiotic homolog invasions

Dear Dr. Martín-Castellano,

Thank you again for submitting your revised manuscript to The EMBO Journal. It has now been seen once more by the original referees, whose comments are copied below. Since all three were generally satisfied with your revisions and responses to the initial comments, we should be happy to consider the study further for EMBO Journal publication, pending satisfactory addressing of a number of remaining editorial issues:

- Please include a dedicated "Data Availability" section at the end of the Material and Methods; should there no data deposition to public repositories linked to the study, this should still be stated as "This study includes no data deposited in external repositories." For details, please see <https://www.emboj.org/page/journal/14602075/authorguide#dataavailability>

- Please double-check all citations in the reference list, as many of them appear to be still incomplete (lacking page or eLocator numbers).

- Please rename the Conflict of Interest section into "Disclosure and Competing Interests Statement", in accordance with our updated Guide to Authors (<https://www.emboj.org/competing-interests>)

- Please double-check to make sure to all relevant funding information in the manuscript is congruent with the info entered into our submission system. Currently missing in the submission system are: the "Ministerio de Universidades" with an FPU19/03456 grant; "Plan Operativo de Empleo Juvenil (Fondo Social Europeo e Iniciativa de Empleo Juvenil)"

- I would propose that Appendix Tables S1-4 should be uploaded in directly accessible data format, i.e. as four individual DOCX or XLSX files. For this, they would need to be renamed as Expanded View tables (title and in-text call-out: "Table EV1-4"), with their respective title/legend included in a separate tab. Accordingly, the remaining Appendix Table (currently: Appendix Table S5) should be renamed into Appendix Table S1 (also its in-text reference).

- Please pre-face the Appendix with the header "Appendix" and the article title and author names. Please add each Appendix Figure legend directly underneath the respective figure, instead of in one single block. The references listed after the final Appendix Table should be collated in a dedicated list of "Appendix References" at the end of the Appendix, and should also be listed in the Appendix Table of Contents.

- In the figures/figure legends, please note that the exact p values need to be provided in the legends of Figures EV 3b-c; and that the box plots need to be defined in terms of minima, maxima, bounds of box and whiskers, and percentile in the legends of Figures 4c; EV 3b.

- Finally, during our routine pre-acceptance data checks, we noticed that several micrographs and blot panels appear to have been reused in different figures without sufficient indication/clarification. Please make sure to clearly indicate in all respective figure legends if and why certain panels are repeatedly shown. Please also explain and clarify in your final response letter to the editor. In particular, micrographs appear to be re-used between Figures 4 and EV2; and blots are reused in Figures EV4 and Appendix Figures S4 and S5. Although this is briefly mentioned in only one of the legends, it appears puzzling that certain tubulin controls would appear in different combinations between figures, when they should supposedly come from the same experiments.

I am therefore returning the manuscript to you for a final round of revision, to allow you to provide the necessary clarifications, and to upload revised files including the requested presentational and editorial modifications. After that, we should hopefully be able to proceed with formal acceptance and production of the manuscript.

Yours sincerely,

Hartmut Vodermaier

Response to the Editor

We would like to thank the reviews for the positive view of our revised manuscript, which has definitively been improved by their comments. We also thank EMBO Journal for the possibility of publishing our work.

Our response point by point to the Editor comments is highlighted **in red**.

- Please include a dedicated "Data Availability" section at the end of the Material and Methods; should there no data deposition to public repositories linked to the study, this should still be stated as "This study includes no data deposited in external repositories." For details, please

see <https://www.embopress.org/page/journal/14602075/authorguide#dataavailability>

We have added this statement (highlighted in blue in the word document).

- Please double-check all citations in the reference list, as many of them appear to be still incomplete (lacking page or eLocator numbers).

We have added this missing information (highlighted in blue in the word document).

- Please rename the Conflict of Interest section into "Disclosure and Competing Interests Statement", in accordance with our updated Guide to Authors (<https://www.embopress.org/competing-interests>).

We have changed this accordingly to the indications (highlighted in blue in the word document).

Related to this we have also renamed the Materials and Methods section into "Methods" (highlighted in blue in the word document).

- Please double-check to make sure to all relevant funding information in the manuscript is congruent with the info entered into our submission system. Currently missing in the submission system are: the "Ministerio de Universidades" with an FPU19/03456 grant; "Plan Operativo de Empleo Juvenil (Fondo Social Europeo e Iniciativa de Empleo Juvenil)".

We have entered this information into the submission system.

- I would propose that Appendix Tables S1-4 should be uploaded in directly accessible data format, i.e. as four individual DOCX or XLSX files. For this, they would need to be renamed as Expanded View tables (title and in-text call-out: "Table EV1-4"), with their respective title/legend included in a separate tab. Accordingly, the remaining Appendix Table (currently: Appendix Table S5) should be renamed into Appendix Table S1 (also its in-text reference).

We have renamed these Tables as suggested. They have been renamed in the main text and figure legends (highlighted in blue in the word document).

They have been also renamed in the legend of the Appendix Figure S3 (included in the new Appendix pdf file that has been uploaded to the submission system).

They have been uploaded to the submission system as Expanded View files (Table EV1-4), along with a corresponding title and brief legend.

- Please pre-face the Appendix with the header "Appendix" and the article title and author names. Please add each Appendix Figure legend directly underneath the respective figure, instead of in one single block. The references listed after the final Appendix Table should be collated in a dedicated list of

"Appendix References" at the end of the Appendix, and should also be listed in the Appendix Table of Contents.

Appendix information has been edited following the indications and a new Appendix pdf file has been uploaded to the submission system.

- In the figures/figure legends, please note that the exact p values need to be provided in the legends of Figures EV 3b-c; and that the box plots need to be defined in terms of minima, maxima, bounds of box and whiskers, and percentile in the legends of Figures 4c; EV 3b.

The missing p values have been added to Figure 4 and Figure EV3, and the new figures uploaded to the submission system.

The sentence "n.s. p value > 0.05" has been removed from the figure legends, and box plots are now defined in more detail (highlighted in blue in the word document).

"Data are represented by box-and-whisker plots where boxes extend from the 25th to 75th percentiles, and bars within the boxes represent the medians and black crosses the means; the whiskers represent the minimum and the maximal range."

- Finally, during our routine pre-acceptance data checks, we noticed that several micrographs and blot panels appear to have been reused in different figures without sufficient indication/clarification. Please make sure to clearly indicate in all respective figure legends if and why certain panels are repeatedly shown. Please also explain and clarify in your final response letter to the editor. In particular, micrographs appear to be re-used between Figures 4 and EV2; and blots are reused in Figures EV4 and Appendix Figures S4 and S5. Although this is briefly mentioned in only one of the legends, it appears puzzling that certain tubulin controls would appear in different combinations between figures, when they should supposedly come from the same experiments.

We have added a sentence in the legend of Figure EV2 to clarify the duplication of the 3.5 h images in Figure 4 and Figure EV2 (highlighted in blue in the word document):

"Notice that the images at 3.5 h are also presented as an introductory summary of the experiment in Figure 4A."

We have added the comment only in the legend of Figure EV2 since it is the place where the duplication appears.

We have rephrased the legend of Appendix Figure S3 (please notice that it is not S5) to better explain the duplication of the tubulin blots in Figure EV4A and Appendix Figures S3C and S4C: the same membranes were used to detect EGFP-Sfr1 proteins (Figure EV4A and Appendix Figure S4C), Rad51 (Appendix Figure S3C) and Tubulin (loading control); therefore, the loading control is the same. Please notice that EGFP-Sfr1 proteins and Rad51 run at quite different position in the gel (see molecular weight markers).

“The same membranes shown in Figure EV4A and Appendix Figure S4C were reused for this second western blot; therefore, tubulin detection as loading control is the same for this new WB and same blots are presented here (lower blots).”

The new legend of the Appendix Figure S3 is in the new Appendix pdf file that has been uploaded to the submission system.

We have added the comment only in the legend of Appendix Figure S3 since it is the place where the duplications appear.

Dr. Cristina Martín-Castellanos
Instituto de Biología Funcional y Genómica (IBFG); CSIC-USAL
Zacarías González, 2
Salamanca, Salamanca 37007
Spain

8th Aug 2024

Re: EMBOJ-2023-116463R1
CDK phosphorylation of Sfr1 downregulates Rad51 function in late-meiotic homolog invasions

Dear Dr. Martín-Castellanos,

Thank you for submitting your final revised manuscript for our consideration. I am pleased to inform you that we have now accepted it for publication in The EMBO Journal.

Yours sincerely,

Hartmut Vodermaier
